# LAPP: Large Language Model Feedback for Preference-Driven Reinforcement Learning

**Pingcheng Jian**                                                    *pingcheng.jian@duke.edu*
*Duke University*

**Xiao Wei**                                                                   *xiao.wei@duke.edu*
*Duke University*

**Yanbaihui Liu**                                                     *yanbaihui.liu@duke.edu*
*Duke University*

**Samuel A. Moore**                                                  *yanbaihui.liu@duke.edu*
*Duke University*

**Michael M. Zavlanos**                                        *michael.zavlanos@duke.edu*
*Duke University*

**Boyuan Chen**                                                      *boyuan.chen@duke.edu*
*Duke University*

**Project Website:** www.generalroboticslab.com/LAPP

**Reviewed on OpenReview:** `https://openreview.net/forum?id=cq76wx7T9F`

## Abstract

We introduce Large Language Model-Assisted Preference Prediction (LAPP), a novel framework for robot learning that enables efficient, customizable, and expressive behavior acquisition with minimum human effort. Unlike prior approaches that rely heavily on reward engineering, human demonstrations, motion capture, or expensive pairwise preference labels, LAPP leverages large language models (LLMs) to automatically generate preference labels from raw state-action trajectories collected *during* reinforcement learning (RL). These labels are used to train an online preference predictor, which in turn guides the policy optimization process toward satisfying high-level behavioral specifications provided by humans. Our key technical contribution is the integration of LLMs into the RL feedback loop through trajectory-level preference prediction, enabling robots to acquire complex skills including subtle control over gait patterns and rhythmic timing. We evaluate LAPP on a diverse set of quadruped locomotion and dexterous manipulation tasks and show that it achieves efficient learning, higher final performance, faster adaptation, and precise control of high-level behaviors. Notably, LAPP enables robots to master highly dynamic and expressive tasks such as quadruped backflips, which remain out of reach for standard LLM-generated or handcrafted rewards. Our results highlight LAPP as a promising direction for scalable preference-driven robot learning.

## 1 Introduction

Designing effective reward functions remains a fundamental challenge in training robots with reinforcement learning (RL) (Ratner et al., 2018; Dang et al., 2023; Eschmann, 2021; Sorg et al., 2010; Evans et al., 2021; Ng et al., 1999; Grzes & Kudenko, 2008; Devlin & Kudenko, 2012). Reward functions define the objectives and constraints of the learning process, but are often hand-crafted through trial and error, a process that is

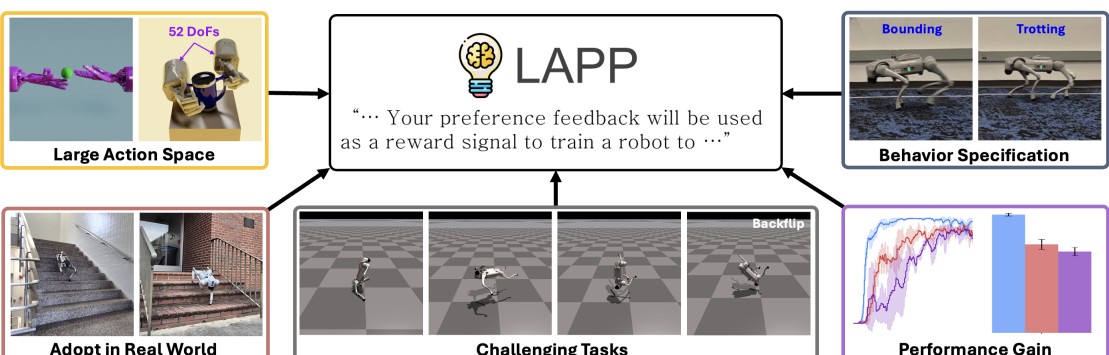

Figure 1: **Large Language Model-Assisted Preference Prediction (LAPP)** takes in language behavior instructions and generates preference feedback to guide reinforcement learning training from raw state-action robot trajectories.

labor-intensive, error-prone, and difficult to scale. Poorly designed rewards can lead to suboptimal or unsafe behaviors, making robust and expressive policy learning especially difficult in real-world robotic systems.

Several alternatives have been proposed to reduce this burden. Inverse RL infers reward functions from expert demonstrations (Arora & Doshi, 2021; Ng et al., 2000; Abbeel & Ng, 2004; Zhou & Small, 2021), but requires significant human effort and data collection. More recently, large language models (LLMs) and vision-language models (VLMs) have been used to automate aspects of reward design (Ma et al., 2023; Yu et al., 2023; Xie et al., 2024; Yu et al., 2024; Wang et al., 2024). These methods typically refine reward functions by analyzing task descriptions or environment code. While promising, they often fail to capture nuanced and high-level behavioral qualities, such as rhythmic locomotion or expressive timing, which are hard to specify with explicit reward terms.

Another line of work bypasses explicit reward engineering by learning from human or AI preferences over trajectory pairs (Christiano et al., 2017; Early et al., 2022; Kim et al., 2023; Wang et al., 2024; Venkataraman et al., 2024). By querying which of two behaviors is preferable, these methods convert preferences into supervision for training reward models. However, preference queries can be costly and cognitively demanding for humans, and recent VLM-based methods remain limited to relatively simple tasks with low-dimensional action spaces. These approaches also typically assume a Markovian decision process, which may not hold for long-horizon and high-dimensional control tasks.

In this work, we propose LLM-Assisted Preference Prediction (LAPP) (Fig. 1), a novel framework that enables robots to learn efficient, customizable, and expressive behaviors from human language specifications. The core idea of LAPP is to leverage LLMs to generate preference labels on full state-action trajectories, which are then used to train an online transformer-based reward predictor. This predictor produces dense, trajectory-informed reward estimates that guide policy optimization and are continually updated with new policy training and rollouts. Unlike prior approaches, LAPP integrates the LLM-generated feedback directly into the RL loop, enabling closed-loop refinement of learned behaviors.

We evaluate LAPP across a suite of challenging control tasks, including quadruped locomotion and dexterous manipulation with up to 52-dimensional action spaces. LAPP not only accelerates training and improves final performance compared to state-of-the-art baselines, but also enables nuanced control over high-level behavior attributes, such as gait symmetry, timing, and cadence, through simple languag inputs. LAPP also enables faster adaptation to unseen environmental conditions. Notably, LAPP successfully solves exploration-heavy tasks such as quadruped backflips, which were previously infeasible with human-designed, LLM-generated, or VLM-derived reward functions.

We summarize our key contributions as follows:

1. A novel learning framework (LAPP) that uses LLM-generated preference feedback over state-action trajectories to guide reinforcement learning with language behavior instructions.

2. A transformer-based online preference predictor that models trajectory-level feedback and integrates it as dense supervision into the RL policy learning loop through iterative policy & reward model improvements.

3. Empirical results showing the LAPP outperforms baselines in training speed, final performance, adaptation efficiency, and controllability of high-level behaviors across complex robotic tasks.

4. Ablation studies dissecting architectural and algorithm design choices to reveal how trajectory-level modeling and online reward model updates contribute to LAPP's success.

## 2 Related Work

**Foundation Models for Robotics.** Recent advances in foundation models have spurred applications in robotic action generation (Octo Model Team et al., 2024; Szot et al., 2023; Zitkovich et al., 2023; Tang et al., 2023), simulation (Authors, 2024), task planning (Lin et al., 2023; Ahn et al., 2022; Singh et al., 2023; Huang et al., 2024; Zhang et al., 2023; Hao et al., 2023; Liu et al., 2023; Ha et al., 2023; Wang et al., 2023e;a; Ding et al., 2023; Silver et al., 2024; Xie, 2020; Huang et al., 2023; Liang et al., 2023), and sim-to-real transfer (Ma et al., 2024). A growing line of work focuses on using language or vision-language models to automate aspects of reward engineering (Ma et al., 2023; Yu et al., 2023; Xie et al., 2024; Yu et al., 2024; Wang et al., 2024) or generate training environments (Liang et al., 2024; Wang et al., 2023b;d; Faldor et al., 2024; Wang et al., 2023c). However, these approaches remain limited in specifying high-level behaviors (Octo Model Team et al., 2024), handling hard exploration challenges (Ma et al., 2023), and scaling to high-dimensional action spaces (Wang et al., 2024). In contrast, LAPP fills in this gap and achieves superior performance in hard exploration tasks with high-dimensional action spaces given only high-level behavior specification.

**Reward Signal Design for Challenging Robotic Tasks.** Reward design is a crucial component of RL (Ratner et al., 2018; Dang et al., 2023; Eschmann, 2021; Sorg et al., 2010; Jian et al., 2021; Evans et al., 2021; Xia et al., 2024). To address sparse reward issues, prior works explore reward shaping (Grzes & Kudenko, 2008; Devlin & Kudenko, 2012; Devidze et al., 2022; Marom & Rosman, 2018; Hu et al., 2020; Gupta et al., 2022; Grześ, 2017; Zou et al., 2019; Goyal et al., 2019; Memarian et al., 2021; Hussein et al., 2017; Ng et al., 1999). However, complex agile motions, such as rapid locomotion (Margolis et al., 2024) and backflips (Tang et al., 2021; Kim et al., 2024a), remain difficult to learn with a single reward function.

Reward design remains a bottleneck for complex RL tasks, especially when rewards are sparse, brittle, or difficult to engineer. Prior efforts explore reward shaping (Grzes & Kudenko, 2008; Devlin & Kudenko, 2012; Devidze et al., 2022; Marom & Rosman, 2018; Hu et al., 2020; Gupta et al., 2022; Grześ, 2017; Zou et al., 2019; Goyal et al., 2019; Memarian et al., 2021; Hussein et al., 2017; Ng et al., 1999), curriculum learning (Tang et al., 2021; Margolis et al., 2024; Ryu et al., 2024), and multi-objective optimization (Kim et al., 2024a; Kyriakis & Deshmukh, 2022; Van Moffaert et al., 2013; Basaklar et al., 2022; Xu et al., 2020; Cai et al., 2024; Abdolmaleki et al., 2020; Yang et al., 2019; Hayes et al., 2022; Huang et al., 2022). Inverse RL methods aim to infer reward signals from demonstrations (Arora & Doshi, 2021; Ng et al., 2000; Hadfield-Menell et al., 2016; Zakka et al., 2022; Brown et al., 2018; Kumar et al., 2023; Das et al., 2021; Abbeel & Ng, 2004; Zhou & Small, 2021), but require curated expert data.

While recent works attempt to automate reward or curriculum generation using LLMs (Ma et al., 2023; Liang et al., 2024), they still depend on explicit reward decompositions or low-level state supervision which can be difficult to obtain for complex tasks and high-level behavior specifications. Our method complements these advances by using LLMs to generate implicit preference feedback. As our experiments show, our method achieves the best performance when combined with previous state-of-the-art reward designs.

**Human-Guided Machine Learning.** Integrating human guidance into machine learning has been widely explored to improve training efficiency and model performance (Amershi et al., 2014; Gil et al., 2019; Wu et al., 2022; Zhang et al., 2019). Various methods incorporate human demonstrations (Pomerleau, 1988; Schaal, 1996), instructions (Zhou & Small, 2021; Saran et al., 2021), and corrections (Chai & Li, 2020; Ji et al., 2024) to enhance imitation learning (Pomerleau, 1988; Schaal, 1996; Saran et al., 2021) or inverse RL (Abbeel & Ng, 2004; Zhou & Small, 2021). Other works model human feedback as reward functions (Knox &

Stone, 2008; Warnell et al., 2018) or advantage functions (MacGlashan et al., 2016; Arumugam et al., 2019) to guide RL. Recent advancements extend these algorithms to continuous action spaces (Sheidlower et al., 2022), multi-agent scenarios (Ji et al., 2024), multi-agent scenarios (Ji et al., 2024), and real-time human feedback (Zhang et al., 2024a;b).

The most relevant works to ours are those that learn from human preferences (Wirth et al., 2017; Akrour et al., 2011; Daniel et al., 2015; Fürnkranz et al., 2012; Ibarz et al., 2018; Wilson et al., 2012; Wirth et al., 2016; Kim et al., 2024b; Dong et al., 2023; Liu et al., 2024; Aroca-Ouellette et al., 2024; Akrour et al., 2012; Liu et al., 2020; Lee et al., 2021b; Knox et al., 2022; Ouyang et al., 2022; Park et al., 2022; Verma & Metcalf, 2022; Christiano et al., 2017), with applications in LLM fine-tuning (Brown et al., 2020), summarization (Wu et al., 2021), browser-assisted question answering (Nakano et al., 2021), robotic manipulation (Hejna III & Sadigh, 2023), and locomotion (Yuan et al., 2024). Yu et al. (2024) uses human preference to select the reward functions generated by a LLM, while the human preference is not directly predicted as a preference reward to guide the policy optimization.

Our work builds on reinforcement learning from human feedback (RLHF) (Christiano et al., 2017), where human preferences are used to train MLP-based preference predictors for Markovian rewards. Later works extend this to non-Markovian settings with LSTMs (Early et al., 2022) and importance-weighted rewards using Preference Transformers (Kim et al., 2023). However, RLHF methods require extensive human annotation, with human annotators evaluating thousands of trajectory pairs. Recent research proposes Reinforcement Learning from AI Feedback (RLAIF) (Bai et al., 2022; Lee et al., 2024; Wang et al., 2024; Venkataraman et al., 2024), replacing human annotators with AI models. However, these approaches are limited to Markovian rewards and have only been tested in low-dimensional robotic tasks. Our work not only reduces annotation cost but allows for preference-driven RL in more complex task domains than those explored in existing RLHF or RLAIF frameworks.

**Preference Feedback for Robot Learning.** Learning from human or AI preferences has emerged as an alternative to explicit reward design (Christiano et al., 2017; Early et al., 2022; Kim et al., 2023; Yuan et al., 2024). These methods train reward models using preference labels over trajectory pairs, typically annotated by humans. While effective, annotation costs remain high.

More recent works adopt AI-generated feedback in place of human raters, such as RL-VLM-F (Wang et al., 2024; Venkataraman et al., 2024), which uses vision-language models to rank state images. However, such models operate under a fixed preference criterion throughout the entire policy learning process and assume Markovian rewards, limiting them to relatively simple low-DoF tasks like CartPole or tabletop manipulation. Specifically, the manipulation tasks in RL-VLM-F have 4 to 6 degrees of freedom, while the dexterous manipulation tasks in LAPP have 52 degrees of freedom, and our locomotion tasks have 12 degrees of freedom.

LAPP offers advancements in this direction in several key aspects. First, LAPP is the first work to operate on raw state-action trajectories to provide effective preference feedback from LLMs. This method avoids reliance on vision-based snapshots to query VLMs, which currently come with much higher costs than LLMs and still do not yet demonstrate strong reasoning capabilities, hence limiting the task complexity they can solve. Second, LAPP models both Markovian and non-Markovian preference rewards using transformer architectures to allow reasoning over long temporal sequences.

Moreover, LAPP enables dynamic preference shaping by prompting LLMs to evolve their evaluation criteria as training progresses, while prior works (Wang et al., 2024; Venkataraman et al., 2024) rank the states with a static standard. To our knowledge, LAPP is the first method to fully automate preference alignment via LLMs for training policies in complex and high-dimensional tasks, including quadruped backflips and dexterous hand manipulation.

## 3 Preliminaries

We consider an agent interacting with an environment over a sequence of discrete timesteps in a RL framework (Sutton & Barto, 2018). At each timestep $t$, the agent receives an observation $o_t$ of the current state $s_t$ and selects an action $a_t$ based on its policy $\pi$. The environment then provides a reward $r_t$ based on the

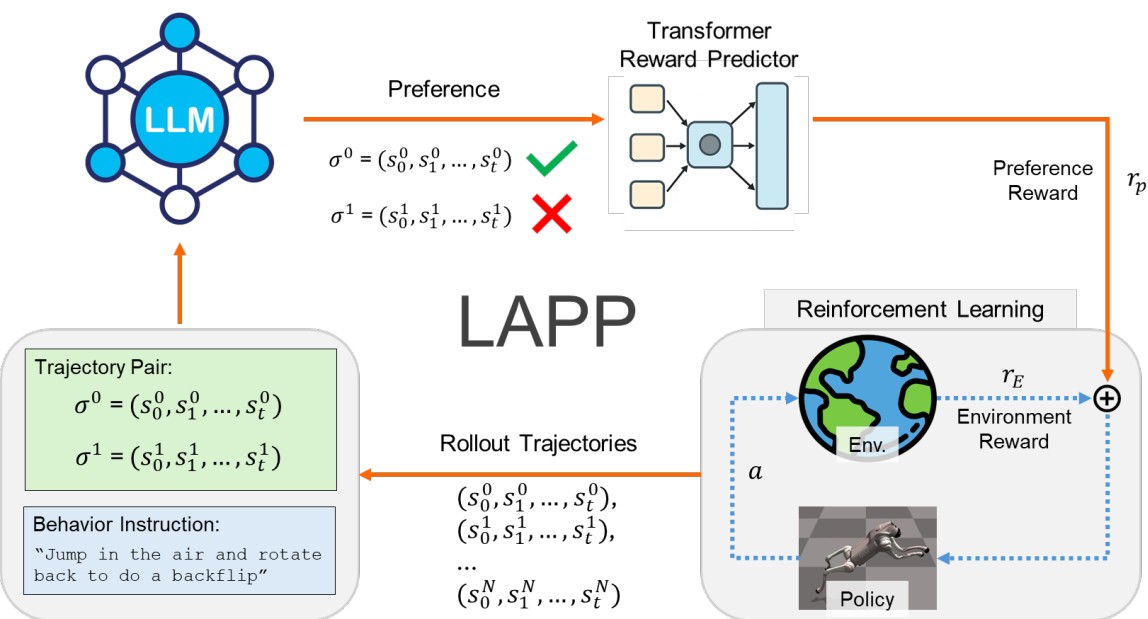

Figure 2: **LAPP** generates preference feedback from an LLM based on rollout trajectories pairs of raw state and actions as well as a high-level behavior instruction. A transformer-based reward predictor is trained using these preferences while simultaneously optimizing a robot policy to maximize a weighted sum of environment rewards and predicted preference rewards.

pre-designed reward functions and transitions the agent to the next state $s_{t+1}$. The goal of RL is to optimize $\pi$ to maximize the expected return $\mathcal{R}_t = \sum_{k=0}^{\infty} \gamma^k r_{t+k}$ where $\gamma$ is the discount factor.

However, designing a reward function that precisely captures high-level behavioral objectives or human preferences remains challenging. To address this, preference-based RL learns a reward model that predicts human preferences instead of relying on manually defined rewards. In this setting, we consider a pair of trajectory segments $(\sigma^0, \sigma^1)$ with length $H$: $\sigma = \{(s_1, a_1), ..., (s_H, a_H)\}$. A preference relation $\sigma^i \succ \sigma^j$ indicates that segment $\sigma^i$ is preferable over segment $\sigma^j$. Given a pair $(\sigma^0, \sigma^1)$, a human or AI provides a preference label $y \in \{0, 1, 0.5\}$:

$$y = \begin{cases} 0 & , \sigma^0 \succ \sigma^1 \\ 1 & , \sigma^1 \succ \sigma^0 \\ 0.5 & , \sigma^0 \text{ and } \sigma^1 \text{ are equally preferable} \end{cases}$$

The preference judgments are recorded in a dataset $\mathcal{D}$ of labeled preference triples $(\sigma^0, \sigma^1, y)$.

To obtain a preference-based reward model $\hat{r}$, prior works (Christiano et al., 2017; Ibarz et al., 2018; Lee et al., 2021b;a; Hejna III & Sadigh, 2023; Park et al., 2022) adopt the Bradley-Terry model (Bradley & Terry, 1952), assuming Markovian rewards (i.e. the reward depends only on the current state and action). The probability of preferring one segment over another is modeled as:

$$\hat{P}\left[\sigma^1 \succ \sigma^2\right] = \frac{\exp \sum \hat{r}\left(s_t^1, a_t^1\right)}{\exp \sum \hat{r}\left(s_t^1, a_t^1\right) + \exp \sum \hat{r}\left(s_t^2, a_t^2\right)} \tag{1}$$

However, Markovian rewards struggle with long-horizon tasks where preferences depend on past trajectories rather than only the current state-action pair. To address this, recent works (Kim et al., 2023; Early et al., 2022) propose non-Markovian rewards, where $\hat{r}$ considers the full preceding sub-trajectory segment $\{(\mathbf{s}_i, \mathbf{a}_i)\}_{i=1}^{t}$:

$$\hat{P}[\sigma^1 \succ \sigma^0] = \frac{\exp \hat{r}(\{(\mathbf{s}_i^1, \mathbf{a}_i^1)\}_{i=1}^{t})}{\sum_{j \in \{0,1\}} \exp \hat{r}(\{(\mathbf{s}_i^j, \mathbf{a}_i^j)\}_{i=1}^{t})} \tag{2}$$

The reward predictor $\hat{r}$ is then trained via supervised learning to fit the dataset $\mathcal{D}$ by minimizing the cross-entropy loss:

$$\mathcal{L}^{CE}(\hat{r}) = - \sum_{(\sigma^1, \sigma^2, y) \in \mathcal{D}} (1-y) \log \hat{P}\left[\sigma^0 \succ \sigma^1\right] + y \log \hat{P}\left[\sigma^1 \succ \sigma^0\right] \tag{3}$$

To mitigate the noisy LLM outputs, We assume that the LLM has $\epsilon = 15\%$ of chance to provide preference feedback uniformly at random. Therefore, the adjusted preference probability is:

$$\hat{P}'\left[\sigma^0 \succ \sigma^1\right] = (1-\epsilon)\hat{P}\left[\sigma^0 \succ \sigma^1\right] + \epsilon \cdot 0.5, \tag{4}$$

where $\epsilon = 0.15$ is the error rate. We also have:

$$\hat{P}'\left[\sigma^1 \succ \sigma^0\right] = 1 - \hat{P}'\left[\sigma^0 \succ \sigma^1\right]. \tag{5}$$

Therefore, the adjusted cross-entropy loss becomes:

$$\begin{aligned}
\mathcal{L}_{\epsilon}^{CE}(\hat{r}) &= - \sum_{(\sigma^0, \sigma^1, y) \in \mathcal{D}} \left[(1-y) \log \hat{P}'\left[\sigma^0 \succ \sigma^1\right] + y \log \hat{P}'\left[\sigma^1 \succ \sigma^0\right]\right] \\
&= - \sum_{(\sigma^0, \sigma^1, y) \in \mathcal{D}} \left[(1-y) \log \left((1-\epsilon)\hat{P}\left[\sigma^0 \succ \sigma^1\right] + \epsilon \cdot 0.5\right) \right. \\
&\quad + y \log \left(1 - \left((1-\epsilon)\hat{P}\left[\sigma^0 \succ \sigma^1\right] + \epsilon \cdot 0.5\right)\right)\Big].
\end{aligned} \tag{6}$$

Once trained, the reward predictor $\hat{r}$ can be used to guide policy optimization, where an RL algorithm maximizes the expected return from the learned preference rewards. We leave the exploration of tuning $\epsilon$ as future work. All our experiments have used the same value, hence we have observed that our method is not sensitive to the selection of this value.

## 4 LLMs-Assisted Preference Prediction

LAPP is a novel framework that enables preference-driven RL by integrating LLM-generated feedback into the policy training loop. It consists of three main components: 1) **Behavior Instruction:** a prompting strategy to elicit trajectory preferences from LLMs given language description of task objectives and preferred behaviors; 2) **Preference Predictor Training:** an ensemble of transformer-based models that learn to predict preference rewards; and 3) **Preference-Driven Reinforcement Learning:** a robot policy is optimized using both environment rewards and predicted preference rewards. An overview of LAPP is shown in Fig. 2.

### 4.1 Behavior Instruction: Generating Preference Labels from State-Action Trajectories

Conventional RLHF frameworks rely on human annotators to label trajectory preferences. However, this process is labor intensive. To reduce this burden, LAPP replaces human annotators with LLMs by prompting them to generate preference labels for pairs of trajectory segments $\sigma^0$ and $\sigma^1$.

Fig. 3 illustrates an example of a behavior instruction prompt. The first part defines the LLM's role the robot's goal and desired behavior (e.g., "walk forward with a bounding gait"). The second part provides numerical values and their descriptions of each trajectory (e.g., base velocity, orientation, foot contacts). The third part defines how preferences should be evaluated and formatted.

Unlike prior work that uses video clips (Christiano et al., 2017; Kim et al., 2023) for human annotation, we feed LLMs structured numerical state-action logs, since the current multimodal foundation models such as GPT-4o lacks fine-grained video understanding with high costs and slow responses.

Notably, to enhance learning efficiency, we encourage the LLM to refer to our provided evaluation criteria and generate *adaptive evaluation criteria, allowing the LLM to dynamically adjust its preferences as training progresses.* For instance, in quadruped locomotion, early-stage training should prioritize learning to stand, followed by developing stable movement, and ultimately refining gait patterns and command adherence. Instead of providing these stages explicitly by humans, our prompts ask the LLM to actively decide the important factors for different training stages by itself.

LAPP supports batched labeling of five trajectory pairs per prompt which can significantly reduce API latency and token costs. the output consists of preference labels in $0, 1, 2, 3$, indicating whether trajectory $\sigma^0$ is better, worse, equally preferable, or incomparable. To promote clear supervision, we encourage the LLM to avoid ambiguous labels. All labels are stored as triples $(\sigma^0, \sigma^1, y)$ in a growing preference dataset $\mathcal{D}$. Details of all state-action variables used for all tasks and full prompts can be found in Appendix A.2.

## 4.2 Preference Predictor Training: Modeling LLM Feedback

LAPP models LLM-generated preferences using either Markovian or non-Markovian reward functions, depending on task complexity. For tasks like flat-ground locomotion, a Markovian reward model following the Bradley-Terry formulation (Eq. 1) suffices. However, for more challenging tasks such as quadruped backflips or gait cadence control, a non-Markovian reward function (Eq. 2) is necessary to capture long-term dependencies in behavior. Training the predictor for non-Markovian rewards requires additional computational resources, as it must process historical states to infer the reward at a given timestep. LAPP adopts the appropriate reward model based on task requirements to balance the preference prediction accuracy and the predictor training efficiency.

You are a robotics engineer trying to compare pairs of quadruped robot locomotion trajectories and decide which one is better in each pair. Your feedback of the comparisons will be used as a reward signal to train a quadruped robot to ...

Each trajectory will contain 24 time steps of states of the robot trying to... The state includes:
1)  ...
2)  ...
...
To decide which trajectory is better in a pair, here are some criteria:
1)  ...
2)  ...
...

The user will provide 5 pairs of trajectories in a batch, and you should provide 1 preference value for each pair.
1) If the trajectory 0 is better, the preference value should be 0.
2) If the trajectory 1 is better, the preference value should be 1.
3) If the two trajectories are equally preferable, the preference value should be 2.
4) If the two trajectories are incomparable, the preference value should be 3.
Please give response with only one list of 5 preference values. ...
If each trajectory has its pros and cons, instead of saying they are equally preferable, you can decide which criteria are more important at this stage of training, and then decide which trajectory is preferable.

Figure 3: **Behavior Instruction Prompt Example.** The LLM prompt consists of three sections: (1) defining the LLM's role and the robotic task (blue box), (2) specifying the state variables and some evaluation criteria of preference (green box), and (3) establishing rules and semantics for generating preference labels (purple box).

The preference dataset $\mathcal{D}_p = \{(\sigma^0, \sigma^1, y)\}$ is split into training ($\mathcal{D}_p^{train}$) and validation ($\mathcal{D}_p^{val}$) sets at a $9:1$ ratio. We maintain an ensemble of $M$ preference predictor networks, each trained to minimize the cross-entropy loss (Eq. 3). To prevent overfitting, training stops early if the validation loss exceeds $\alpha$ times the training loss and the training has gone through a minimum number of iterations $K_{\min}$. If no early stopping is triggered, the training will finish after $K_{\max}$ iterations. After training all $P$ predictors, we select the top $C$ models with the lowest validation losses and compute the final preference reward as their ensemble average. In practice, we set $P = 9$, $C = 3$, $K_{\min} = 30$, $K_{\max} = 90$, and $\alpha = 1.3$. The full training procedure is detailed in Algorithm 1. This ensemble approach can help increase robustness to LLM label noise.

## 4.3 Preference-Driven Reinforcement Learning

LAPP continuously aligns robot behaviors with high-level task specifications throughout RL training by iteratively updating both the preference predictor and the policy network. Unlike previous RLHF approaches (Christiano et al., 2017; Kim et al., 2023) that train static preference models, LAPP dynamically refines preferences during training.

Initially, the policy generates rollout trajectory pairs $\{\sigma_i^0, \sigma_i^1\}$, which are evaluated by the LLM to generate preference labels $\{y_i\}$. To mitigate noisy outputs which could pose potential risks to training stability, we sample 15 preference labels for each trajectory pair and calculate the mode of them as the final selected preference labels $\{y_i\}$. The labeled data is stored in an initial preference dataset $\mathcal{D}_p = \{(\sigma_i^0, \sigma_i^1, y_i)\}_{i=1}^{|\mathcal{D}_p|}$, which is used to train the initial preference predictor via Algorithm 1.

During RL training, the reward at each timestep is computed as a weighted sum of the predicted preference reward $r_p$ and the environment reward $r_E$ defined by built-in explicit reward functions in our evaluation

---

**Algorithm 1:** LAPP - Preference Predictor Training

---

**1** **Require:** Ensemble of preference predictors $\{\hat{r}_i\}$, preference predictor training dataset $D_p$

**2** **Hyperparameters:** Minimum iteration $K_{min}$, Maximum iteration $K_{max}$, pool of predictors number $P$, selected predictors number $C$, overfitting scale $\alpha$, LLM feedback error rate $\epsilon$.

**3** //Split into training and validation sets

**4** $D_p^{train}, D_p^{val} \leftarrow split(D_p)$

**5** $val\_loss\_list \leftarrow [\ ]$

**6** **for** $P$ *predictors* **do**

**7** $\quad$ Randomly initialize $\hat{r}_i$

**8** $\quad$ **for** $K \leftarrow 0$ **to** $K_{max} - 1$ *epochs* **do**

**9** $\quad\quad$ // Sample from $D_p^{train}$

**10** $\quad\quad$ $(s_t^{train}, a_t^{train}) \sim D_p^{train}$

**11** $\quad\quad$ // Predict preference reward

**12** $\quad\quad$ $r_t^{train} = \hat{r}_i(s_t^{train}, a_t^{train})$

**13** $\quad\quad$ // Train the preference predictor

**14** $\quad\quad$ Calculate loss $\mathcal{L}_{train}^{CE}(r_t^{train})$ with Equation 6

**15** $\quad\quad$ $\hat{r}_i \leftarrow \text{Adam}\left(\hat{r}_i, \nabla_{\hat{r}_i}\mathcal{L}_{train}^{CE}(r_t^{train})\right)$

**16** $\quad\quad$ // Sample from $D_p^{val}$

**17** $\quad\quad$ $(s_t^{val}, a_t^{val}) \sim D_p^{val}$

**18** $\quad\quad$ // Predict preference reward

**19** $\quad\quad$ $r_t^{val} = \hat{r}_i(s_t^{val}, a_t^{val})$

**20** $\quad\quad$ Calculate loss $\mathcal{L}_{val}^{CE}(r_t^{val})$ with Equation 6

**21** $\quad\quad$ **if** $\mathcal{L}_{val}^{CE}(r_t^{val}) > \alpha \cdot \mathcal{L}_{train}^{CE}(r_t^{train})$ **and** $k > K_{min}$ **then**

**22** $\quad\quad\quad$ val\_loss\_list.append $(\mathcal{L}_{val}^{CE}(r_t^{val}))$

**23** $\quad\quad\quad$ break;

**24** $\quad\quad$ **if** $k == K_{max} - 1$ **then**

**25** $\quad\quad\quad$ val\_loss\_list.append $(\mathcal{L}_{val}^{CE}(r_t^{val}))$

**26** // Select the C predictors with smallest validation losses

**27** $\hat{r}_{i_1}, \hat{r}_{i_2}, ..., \hat{r}_{i_C} \leftarrow \arg\min_{\hat{r}_i \in \{\hat{r}_1, ..., \hat{r}_M\}}^{C} val\_loss\_list[i]$

**28** // Use the mean value of the selected predictors as the final predictor

**29** $\hat{r} = mean\left(\hat{r}_{i_1}, \hat{r}_{i_2}, ..., \hat{r}_{i_C}\right)$

**30** **return** $\hat{r}$

---

suites:

$$r = \beta r_p + r_E \tag{7}$$

where $\beta$ balances their contributions. We set the $\beta$ to be 1.0 in all the tasks except the Backflip. The Backflip task has some reward items with large scales, so the $\beta$ is set to 50.0 to ensure the effective influence of the preference rewards. We encode the key objectives from the environment rewards into LLM prompts as human languages to have LLM understand the key objectives of the task, such as following the speed commands for locomotion. Additionally, LAPP prompts LLMs to provide effective feedback on high-level behavior specifications that are difficult or impossible to ground into environment reward functions, such as "having a natural trotting gait". The policy is optimized using PPO (Schulman et al., 2017), while new trajectory pairs are continuously collected. Every $M$ epochs, newly collected trajectories are evaluated by the LLM, added to $\mathcal{D}_p$, and used to retrain the preference predictor. LAPP only uses the latest labeled trajectories to retrain the preference predictor. As shown in our ablation studies, this design provides higher performance than including all past trajectories. LAPP's online preference learning allows the policy to

---

**Algorithm 2:** LAPP

---

**1** **Require:** Robot behavior prompt *prompt*, preference generator LLM *LLM*, environment $E$, policy $\pi$, preference predictor $\hat{r}$, preference predictor training dataset $D_p$, preference data buffer $B_p$

**2** **Hyperparameters:** Policy optimization epoch number $N$, preference predictors update interval epoch number $M$, per epoch trajectories pairs collection number $K$, per epoch rollouts number $S$, steps in each epoch $T$, preference reward scale $\beta$

**3** **Initialization:** Randomly initialize $\pi$, $\hat{r}$.

$D_p \leftarrow \{\text{zeros triple}_i\}_{i=1}^{|D_p|}$, $B_p \leftarrow \{\text{zeros triple}_i\}_{i=1}^{M*K}$.

**4** // Collect initial preference dataset

**5** Rollout $\pi$ and sample $|D_p|$ trajectories pairs $\{(\sigma_i^0, \sigma_i^1)\}_{i=1}^{|D_p|}$.

**6** $\{y_i\}_{i=1}^{|D_p|} \sim \text{LLM}\left(\{\sigma_i^0, \sigma_i^1\}_{i=1}^{|D_p|}, \; prompt\right)$

**7** $D_p \leftarrow \{(\sigma_i^0, \sigma_i^1, y_i)\}_{i=1}^{|D_p|}$

**8** Update $\hat{r}$ with Algorithm 1.

**9** $obs \sim E.reset()$ // reset $E$, get initial observation

**10** **for** $i \leftarrow 0$ **to** $N-1$ *epochs* **do**

**11**      //rollout $\pi$ in $E$

**12**      **for** $T$ *steps* **do**

**13**          $a \sim \pi(obs)$   // sample action from policy

**14**          $r_E \sim E(obs)$   // get environment reward $r_E$

**15**          $r_p \sim \hat{r}(obs)$   // predict preference reward $r_p$

**16**          $r = \beta \cdot r_p + r_E$   // calculate weighted sum

**17**      Update $\pi$ with PPO algorithm (Schulman et al., 2017)

**18**      // Sample $K$ pairs from $S$ rollouts

**19**      $\{(\sigma_k^0, \sigma_k^1)\}_{k=1}^{K} \sim \{\sigma_s\}_{s=1}^{S}$

**20**      // Push into the preference data buffer

**21**      Push $\{(\sigma_k^0, \sigma_k^1, None)\}_{k=1}^{K}$ into $B_p$

**22**      // Update the preference dataset

**23**      **if** *(i+1) % M == 0* **then**

**24**          // Generate preference labels

**25**          $\{y_k\}_{k=1}^{M*K} \sim \text{LLM}\left(\{(\sigma_k^0, \sigma_k^1)\}_{k=1}^{M*K}, \; prompt\right)$

**26**          // Place preference labels into the buffer

**27**          $B_p \leftarrow \{(\sigma_k^0, \sigma_k^1, y_i)\}_{k=1}^{M*K}$

**28**          // Update preference dataset

**29**          $D_p \leftarrow \{\text{triple}_k | \text{triple}_k \in D_p\}_{k=M*K+1}^{|D_p|} \cup B_p$

**30**          // Update preference predictor

**31**          Update $\hat{r}$ with Algorithm 1.

**32**          $B_p \leftarrow \{\}$

**33** **return** $\pi$

---

progressively align with LLM preferences based on its dynamic evaluation criteria according to different learning stages. The full RL procedure is detailed in Algorithm 2, and the environment rewards $r_E$ of all the tasks can be found in Appendix A.3.

### 4.4 Network Architectures

The preference predictor is a transformer network (Waswani et al., 2017) based on the GPT architecture (Radford, 2018) with 6 masked self-attention layers. Inputs are embedded into a 128-dimensional space with sinusoidal positional encodings and processed by 8-headed attention layers. Each block includes a 2-layer MLP with GELU activations (Hendrycks & Gimpel, 2016) and layer normalization (Ba, 2016) to the output tensor from the last self-attention block. A final decoder outputs a scalar reward.

For Markovian rewards, the input sequence length is 1 and the casual mask in the self-attention layer is removed. For non-Markovian rewards, the input sequence length is 8, with zero-padding applied for shorter trajectories.

For quadruped tasks, the policy is an MLP with layers [512, 256, 128] and ELU activations (Clevert, 2015), outputting 12 target joint angles. A PD controller computes the torque commands. For dexterous manipulation, we use the same MLP architecture. The output is a 52-dimensional joint displacement vector for two 26-DoF Shadow Hands (ShadowRobot, 2005).

## 5 Experiments

We evaluate LAPP on a diverse set of quadruped locomotion and dexterous manipulation tasks to assess its ability to:

1. improve both training efficiency and task performance,

2. enable high-level behavior control via language instructions, and

3. solve highly challenging tasks that are very difficult or even infeasible with traditional reward engineering.

Additionally, we conduct ablation studies to analyze key design choices in LAPP, identifying the factors contributing to its performance gains. Finally, we deploy the trained policies on a physical quadruped robot across various terrains and tasks to demonstrate LAPP's real-world applicability.

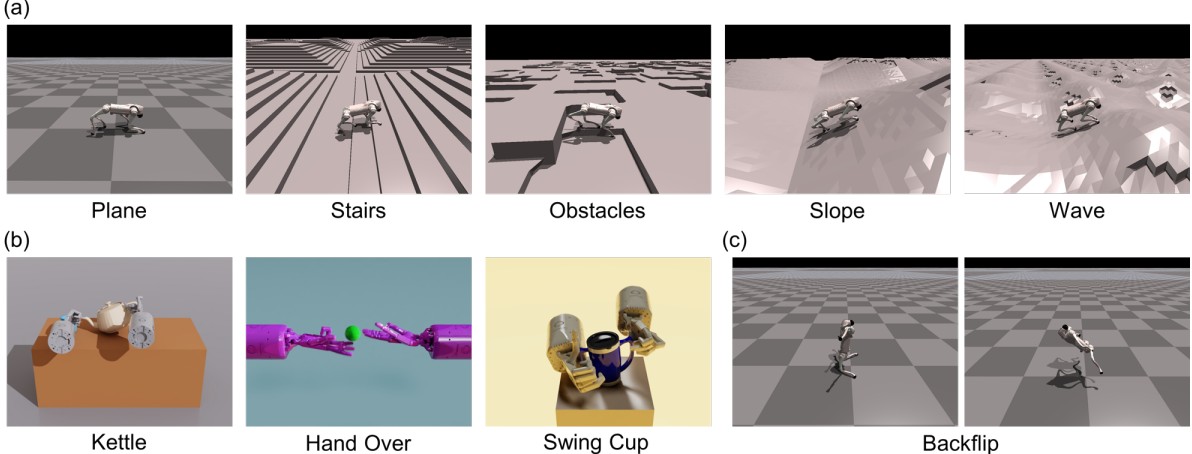

Figure 4: **Simulation Tasks.** (a) Quadruped locomotion. The robot learns to walk forward across various terrains following given velocity commands. The terrains include the flat plane, stairs pyramids, discrete obstacles, slope pyramids, and wave-pattern hills. (b) Dexterous manipulation. Each dexterous hand has 26 degrees of freedoms. Kettle requires the robot to pick up the kettle with one hand, and the cup with another hand, and then pour water into the kettle. Hand Over requires one hand to pass a ball to another hand. Swing Cup requires two hands to hold the cup and rotate it for 180°. (c) Quadruped backflip. The robot jumps in the air and rotate backwards for 360°, and then land on the ground.

We use GPT-4o mini (Achiam et al., 2023) (`gpt-4o-mini-2024-07-18` variant) as the LLM backbone for LAPP. A full training run for each policy (5000 epochs) costs approximately $2.5 to $3, which is significantly lower than the $40 to $50 required for the larger GPT-4o variant, while still achieving satisfactory results. Since LAPP involves frequent online LLM queries, its ability to succeed with a smaller and cheaper LLM is crucial for broader practical adoption. For the Eureka baseline, we use GPT-4o (`gpt-4o-2024-08-06` variant) to ensure a faithful reproduction of its full capabilities from the original work. Each evolutionary reward search with Eureka costs approximately $3. The videos of all the experiment results can be found at our project website.

## 5.1 Baselines

**PPO.** This baseline uses a well-tuned Proximal Policy Optimization (PPO) implementation (Rudin et al., 2022; Schulman et al., 2017). In each task, PPO is trained with the same environment reward functions as LAPP. These reward functions are directly adopted from state-of-the-art policies designed by expert robot learning researchers, representing the current best outcomes from human reward engineering. For a fair comparison, PPO shares all hyperparameters with LAPP. The only difference is that PPO does not incorporate preference rewards, allowing us to isolate and analyze the effect of LAPP's preference-guided learning design.

**Eureka.** Evolution-driven Universal Reward Kit for Agents (Eureka) (Ma et al., 2023) is a recent LLM-based approach for automated reward function design. It prompts an LLM with reward design guidelines and environment source code to generate executable Python reward functions. Eureka then performs an evolutionary search to refine the reward function over multiple iterations based on observed training performance. By following the original implementation, we conduct 5 evolutionary search iterations with 16 reward samples per iteration. Out of the $80 = 16 \times 5$ reward functions, the best-discovered reward function is then used to train the policy with PPO using the same hyperparameters as the PPO baseline.

## 5.2 Simulation Experiments

**Tasks:** We evaluate LAPP on five quadruped locomotion tasks, three dexterous manipulation tasks, and one quadruped backflip task, as shown in Fig. 4. The Unitree Go2 robot (Robotics, 2023) is used for quadruped experiments, while the Shadow Dexterous Hand (ShadowRobot, 2005; Andrychowicz et al., 2020) is used for dexterous manipulation. The locomotion and manipulation tasks are established RL benchmarks from prior works (Ma et al., 2023; 2024; Rudin et al., 2022). The quadruped backflip task is an extremely challenging control problem, previously studied in multi-objective RL (Kim et al., 2024a). While some RLHF studies have explored backflips, they have primarily used the Hopper model in Gym-Mujoco (Christiano et al., 2017; Kim et al., 2023), which is significantly easier due to its lower degrees of freedom (3 DoFs) with no real-world physical counterpart.

The quadruped locomotion tasks are derived from massively parallel RL experiments in Rudin et al. (2022)'s prior work. As shown in Fig. 4(a), we evaluate LAPP on five terrain types including a flat plane, stairs pyramids, discrete obstacles, slope pyramids, and a periodic wave terrain (with periodic wave-pattern hills).

The dexterous manipulation tasks are from the Bidexterous Manipulation (Dexterity) benchmark (Chen et al., 2022) and are also evaluated in Eureka (Ma et al., 2023). As shown in Fig. 4 (b), we evaluate LAPP on the Kettle, Hand Over, and Swing Cup tasks. Kettle requires one hand to hold a kettle and pour water into a cup held in the other hand. Hand Over requires to hand over a ball from one hand to another hand. Swing Cup requires the two hands to collaborate to turn a cup for 180°.

Finally, the quadruped backflip task (Fig. 4 (c)) requires the Unitree Go2 robot to perform a 360° backward rotation mid-air and land successfully. Unlike Hopper-based backflip tasks in prior RLHF studies (Christiano et al., 2017; Kim et al., 2023), which focus on low DoFs and lightweight dynamics, our setup utilizes the Go2's official simulator (Robotics, 2023), incorporating realistic physical parameters, which makes the task significantly more challenging for RL.

**LAPP improves training efficiency.** Fig. 5 shows the learning curves of LAPP, Eureka, and PPO across five locomotion tasks and three dexterous manipulation tasks. Locomotion tasks are evaluated with a fixed

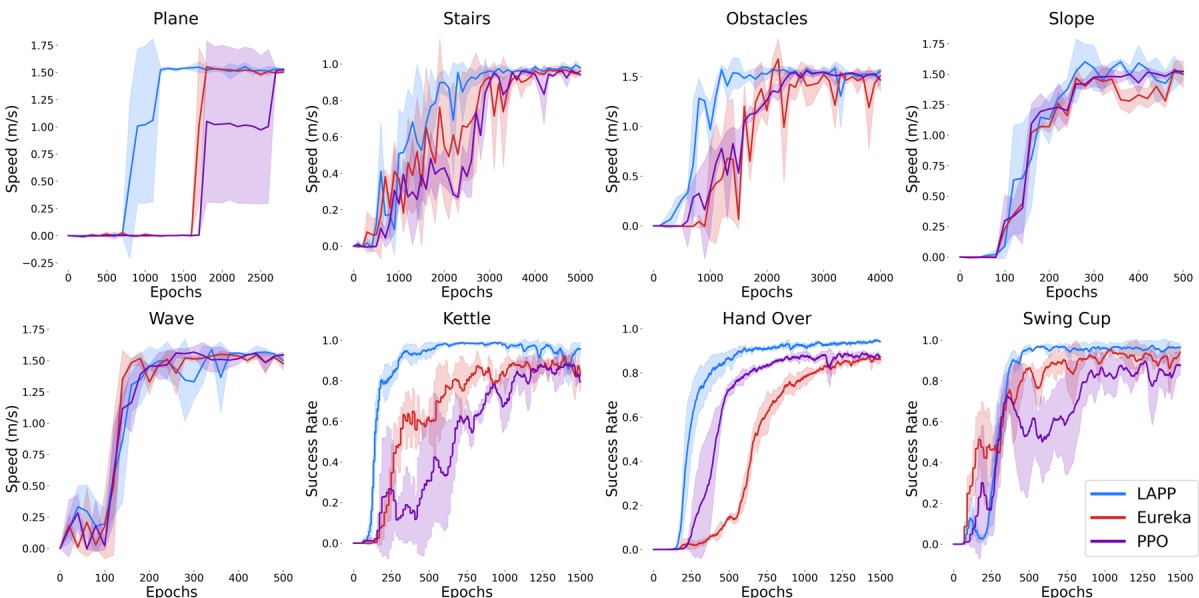

Figure 5: **Training Efficiency.** Training with LAPP converges faster in the Plane, Stairs, Obstacles, Hand Over, Swing Cup and Kettle tasks, while also exhibiting more stable performance post-convergence in Swing Cup. In the Slope and Wave tasks, LAPP performs similarly to baselines as these tasks are relatively easier for exploration, converging quickly for all algorithms.

velocity command of 1.0 m/s (Stairs) or 1.5 m/s (other terrains). For manipulation tasks, we report success rate progression throughout training.

LAPP demonstrates faster convergence in flat-plane, stairs, and discrete obstacle locomotion, as well as all manipulation tasks, achieving higher final success rates. These tasks pose non-trivial exploration challenges, where LAPP accelerates learning by dynamically adjusting preference rewards. This flexibility prioritizes different behaviors at different training stages, so that policy exploration can be guided more effectively. In contrast, Eureka struggles with reward balancing, as it relies on a static reward function throughout the training process. While this ensures a well-calibrated reward function, it often results in inferior performance compared to LAPP.

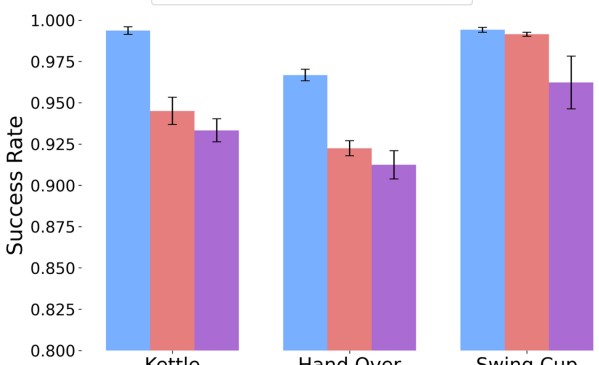

Figure 6: **Convergence Success Rate.** LAPP achieves higher success rates in Kettle, Hand Over, and Swing Cup after the training converges. It shows that the preference rewards can continuously refine the robot motions to improve the performance beyond the reach of explicit reward shaping.

Interestingly, in the Hand Over task, Eureka converges slower than PPO with human-designed rewards. This occurs because Eureka's evolutionary search optimizes for final performance as the fitness score rather than training efficiency. Therefore, this method can help improve the policy performance, but it may not help with training efficiency.

For relatively easier tasks like Slope and Wave, LAPP exhibits similar performance to baselines. This is because randomized robot initialization on smooth slopes can naturally lead to sliding motions to facilitate early exploration of velocity tracking rewards. As a result, all methods converge within 300 epochs in these tasks, suggesting no significant advantage for LAPP in environments where task exploration is inherently easier.

**LAPP achieves higher convergence performance.** Although well-designed reward functions from human experts or LLMs can effectively train quadruped robots to follow velocity commands across various

terrains, they often fail to reach optimal performance in more complex dexterous manipulation tasks such as Kettle, Hand Over, and Swing Cup. As shown in Fig. 6, LAPP significantly improves success rates over PPO desipte sharing the same environment reward functions, increasing from 92% to 99% in Kettle, 91% to 97% in Hand Over, and 96% to 99% in Swing Cup. These gains stem from the continuous motion refinement enabled by LAPP's dynamically updated preference predictor.

Compared to Eureka, LAPP achieves a 6% higher success rate in Kettle and a 5% improvement in Hand Over. In the Swing Cup task, both LAPP and Eureka reach near-optimal 99% success rates, but LAPP converges faster and maintains more stable performance over extended training epochs as in Fig. 5.

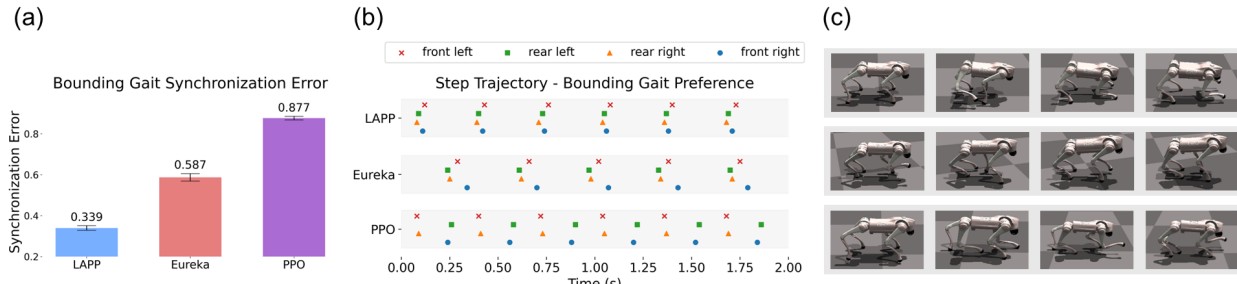

Figure 7: **Bounding Gait Pattern Control.** (a) Feet synchronization error calculated with Eq. 8. LAPP achieves the lowest synchronization error, indicating its closest adherence to a bounding gait. (b) Step trajectories of the robots. LAPP synchronizes both front and rear feet, while Eureka aligns only the rear feet, and PPO fails to produce a bounding gait. (c) Motion frames of the robots.

**LAPP enables behavior control via instruction.** Traditional RL can train robots to complete tasks but cannot typically control how they perform them in a way that aligns with high-level human preferences. Can LAPP guide robot behaviors using high-level specifications in the behavior instruction prompt? To investigate this question, we design two experiments: 1) enforcing a bounding gait in quadruped forward locomotion, and 2) controlling gait cadence to be either higher or lower in quadruped forward locomotion.

A bounding gait requires the quadruped's front and rear feet to make simultaneous ground contact in pairs. To quantify how closely a robot's gait adheres to this pattern, we adopt a synchronization error definition as in Eq. 8:

$$\text{sync\_error} = \frac{1}{N} \sum_{t=1}^{N} \left( |\text{FL}_t - \text{FR}_t| + |\text{RL}_t - \text{RR}_t| \right), \tag{8}$$

where $\text{FL}_t$, $\text{FR}_t$, $\text{RL}_t$, and $\text{RR}_t$ represent front left feet, front right feet, rear left feet, rear right feet contacts at time $t$. These are binary values: 1 if the foot is in contact with the ground and 0 if it is in the air. A lower synchronization error indicates a gait pattern closer to bounding.

As shown in Fig. 7 (a), LAPP trains the robot to achieve a bounding gait with the lowest synchronization error. While Eureka also encourages a bounding gait through reward shaping, its synchronization error remains higher than LAPP. PPO fails to enforce a bounding gait effectively.

To further illustrate gait patterns, Fig. 7 (b) presents the step trajectories of all methods, where each dot represents a foot contacting the ground. LAPP successfully trains the robot to synchronize its front and rear feet, ensuring that both front feet land simultaneously, followed by both rear feet. In contrast, Eureka achieves partial synchronization, aligning only the rear feet. PPO fails to learn a bounding gait with unsynchronized foot contacts.

Fig. 7 (c) provides motion frames of the robots trained with LAPP and the baselines. These visualizations complement the trajectory plots, clearly demonstrating that LAPP exhibits stronger behavior control through effective implicit reward shaping to follow the high-level gait patterns.

We also evaluate LAPP's ability to control step cadence through high-level instructions. In this experiment, the behavior instruction prompt specifies a preference for either faster or slower stepping frequency. We

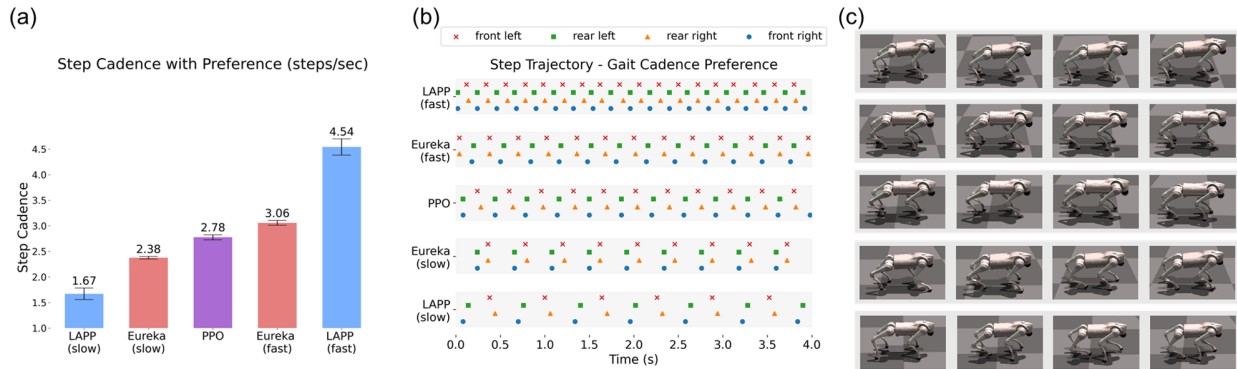

Figure 8: **Gait Cadence Control.** (a) Step trajectories under different cadence instructions. LAPP effectively modulates step frequency based on the high-level prompts by following faster and slower gaits. Eureka can adjust the cadence slightly, but it is less effective.(b) Step cadence comparison. LAPP provides precise control over cadence, while Eureka has limited effect. (c) Motion frames illustrating cadence variation. Robots with high cadence take quick and shallow steps, while those with low cadence take larger and higher steps.

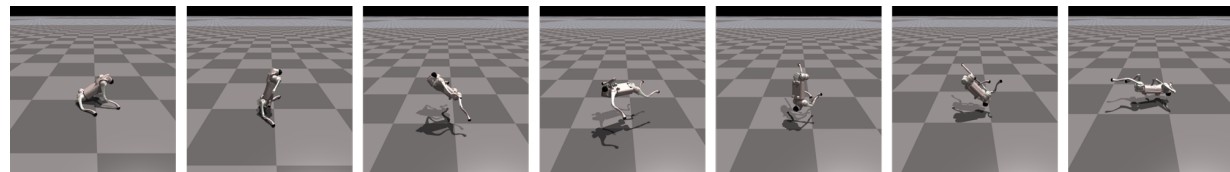

Figure 9: **Quadruped Robot Backflip.** LAPP successfully trains the Unitree Go2 robot to acomplish the backflip task. No baseline including PPO, curriculum learning, or Eureka is able to solve this task.

compare the effectiveness of LAPP and Eureka in enforcing these behaviors. Eureka is also prompted to generate reward functions that encourage the desired cadence. During testing, all robots follow a velocity command of 1.5 m/s. As shown in Fig. 8 (a), with a high-cadence instruction, LAPP achieves 4.54 steps/sec, significantly exceeding Eureka (3.06 steps/sec) and PPO (2.78 steps/sec). Similarly, with a low-cadence instruction, LAPP produces 1.67 steps/sec, notably lower than Eureka (2.38 steps/sec) and PPO (2.78 steps/sec). While Eureka can influence cadence through reward shaping, its effect is far weaker than LAPP's.

The step trajectories in Fig. 8 (b) further highlight LAPP's superior cadence control. Compared to all other methods, LAPP produces the densest step trajectory under a high-cadence instruction and the sparsest trajectory under a low-cadence instruction. Qualitatively, Fig. 8 (c) presents motion frames illustrating the impact of cadence control. With fast cadence, LAPP trains the robot to take small and rapid steps with minimal foot lift. Conversely, with slow cadence, the robot takes larger strides, keeping its feet in the air for extended periods.

Notably, this experiment uses the non-Markovian reward model from Eq. 2. We set the transformer preference predictor's input sequence length to 8. To control the step cadence, the preference predictor needs to consider the history states of the feet contacts to determine the latent reward value of the current state.

**LAPP solves challenging tasks.** Quadruped backflips have long been considered a challenging RL problem due to the need for precise whole-body coordination, complex dynamics, and controlled landing. Some previous RLHF works have trained a Hopper model to perform a backflip in the Gym-Mujoco simulator (Arumugam et al., 2019), but the Hopper has only three joints, is lightweight, and lacks a real-world counterpart.

A recent multi-objective RL (MORL) approach solves the quadruped backflip by dividing the motion into five stages, designing a separate handcrafted reward function for each stage (Kim et al., 2024a). However, this method requires significant expert knowledge, as practitioners must manually define stage transitions and fine-tune rewards for each specific robot.

In contrast, LAPP solves the backflip without intensive human labor for preference feedback or manual reward tuning. We train a Unitree Go2 robot using a weighted combination of a human-designed environment reward and a predicted preference reward. Our process still requires an initial warm-up to encourage the exploration, but we limit our process with simple reward designs for each step. Specifically, we first pre-train the robot to jump vertically. We then randomly initialize the robot in the air. We also reduce the robot's weight during training but restore its real-world weight for testing.

As shown in Fig. 9 and Fig. 10, LAPP successfully trains the robot to jump, rotate backward 360°, and land safely. We also evaluate PPO and Eureka on the same task, using identical exploration strategies (i.e., pre-training to jump, random air initialization, and reduced training weight). PPO follows the same human-designed backflip reward as LAPP but lacks a preference reward, while Eureka generates its own reward function via GPT-4o. Neither PPO nor Eureka succeeds in training the robot for a full backflip. Instead, PPO learns to jump and oscillate the robot's torso up and down but struggles to flip for over 180° with an average maximum rotation of 49.0°. Despite the iterative reward search mechanism of Eureka, it produces similar behavior to PPO with an average maximum rotation of 52.8°. These results demonstrate that explicit reward engineering struggles to capture the complex dynamics of a backflip, while LAPP's preference-driven learning and the adaptive online predictor updates enable successful learning of this highly dynamic capability. We believe that LAPP can shed light on automatically solving many difficult tasks in the near future that are previously unsolvable by conventional RL.

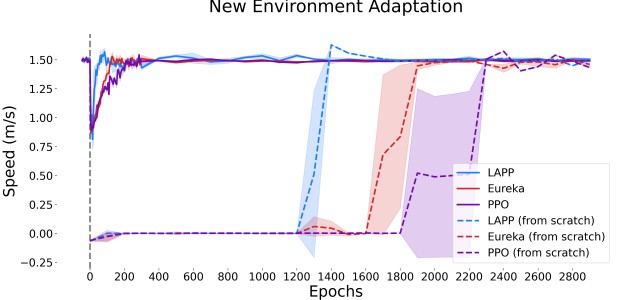

Figure 10: **Backflip rotate angle.** LAPP successfully enables the quadruped robot to complete backflips with full 360° rotation. In contrast, PPO and Eureka fail to generate sufficient rotation, highlighting the advantage of preference-driven learning in solving highly complex and dynamic tasks.

**LAPP enables faster transfer learning.** We evaluate the transfer learning performance of LAPP. For the Plane task with a forward speed command of 1.5 m/s, the Go2 robot is first trained on a flat ground with static friction of 1.0, dynamic friction of 1.0, and a restitution of 0.0. Then, it is transferred to a different flat ground with static friction of 0.01, dynamic friction of 0.01, and a restitution of 0.9. We fine-tune the policy from the source environment in the target environment with LAPP and other baselines. As shown in Fig. 11, the robot speed drops to about 0.9 m/s due to the more slippery ground surface, and LAPP trains the robot to adapt to this new environment faster than the other two baselines. The dashed curves show the learning processes of the robots trained in the target environment from scratch. The results show that transfer learning with a pre-trained policy in a different environment is much faster than training from scratch, and LAPP trains the robot to adapt faster than other baselines.

Figure 11: **Transfer Learning.** For the Plane locomotion task, the robot is transferred to a new environment with different friction and restitution. The solid curves show that LAPP enables the robot to adapt to the new environment faster. Compared with the dashed curves, transfer learning is generally faster than training from scratch.

## 5.3 Ablation Study

We carry out the ablation study to analyze the contributions of two key design decisions of LAPP: 1) updating the preference predictor with the latest rollout trajectories instead of trajectories sampled from the

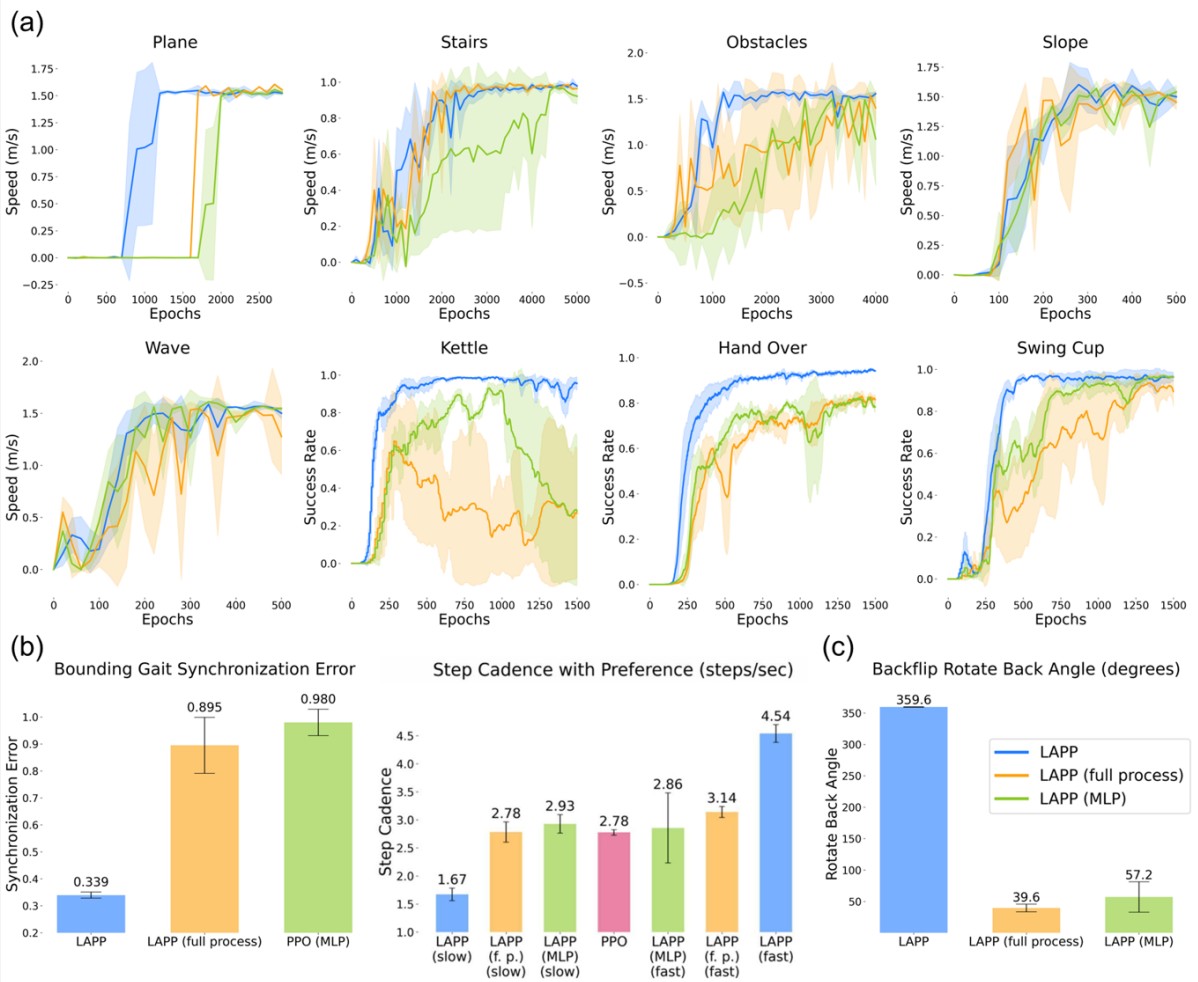

Figure 12: **Ablation Study.** We evaluate two key design choices of LAPP: (1) updating the preference predictor with the latest rollout trajectories (**LAPP** vs. LAPP (full process)), and (2) using a transformer-based reward predictor (**LAPP** vs. LAPP (MLP)). (a) Training Speed: LAPP generally converges faster, except in the Stairs task, where LAPP (full process) achieves similar speed, and in the Slope and Wave tasks, where both tasks are simple enough for LAPP and its ablations to have similar performance. In the Kettle task, both ablations fail at different stages, suggesting that a suboptimal training method for the preference predictor can disrupt policy learning. (b) Behavior Control: Only LAPP successfully controls gait pattern and cadence, while both ablations fail. (c) Challenging Task: LAPP enables successful backflips, achieving a full 360° rotation. In contrast, LAPP (MLP) and LAPP (full process) reach only 57.2° and 39.6°, respectively, and fail to complete the task.

full RL process, and 2) adopting a transformer architecture for the reward predictor network instead of a simple MLP.

To investigate the first design choice, we introduce LAPP (full process), which stores all rollout trajectories in a pool and samples from the entire RL process rather than only the latest epochs (set to 500 epochs in LAPP). The dataset size remains identical between LAPP and LAPP (full process), ensuring a controlled comparison. However, unlike LAPP, this variant may use trajectory data from early training, which potentially introduces biases to suboptimal policy rollouts. Note that we do not use the full trajectory pool for predictor training, as its continuous expansion would slow down online updates.

For the second ablation, LAPP (MLP) replaces the transformer-based predictor with an MLP, limiting it to Markovian preference rewards (Eq. 1). In contrast, LAPP supports both Markovian and non-Markovian rewards (Eq. 2), which are essential for modeling tasks like step cadence control and backflips.

We compare the performance of LAPP and its two ablations in the simulation tasks in Sec. 5.2. Fig. 12 (a) shows that replacing the transformer with an MLP reduces training speed across most locomotion tasks except for the Slope and Wave tasks, which are relatively simple and lead to similar performance of LAPP and its ablations. Moreover, sampling preference data from the full RL process does not affect speed in Stairs, Slope, and Wave, but slows down the training in other locomotion tasks. In Kettle, both ablations initially improve performance but then drop to 20%, suggesting that biased preference data or a predictor network that does not capture long-term dependencies in past states can introduce misleading preference rewards, which ultimately destabilize the policy training.

Fig. 12 (b) evaluates the ability to control gait pattern and step cadence in quadruped locomotion. Only LAPP succeeds, while both ablations fail. This supports the hypothesis that preference rewards must evolve dynamically with training. LAPP (full process) struggles because sampling from the full trajectory pool prevents the predictor from adapting to the current learning stage. LAPP (MLP) fails because MLPs lack the capacity to model non-Markovian rewards, which are essential for cadence control. Specifically, bounding gait requires analyzing foot synchronization from step history, and cadence control depends on tracking past states to predict step timing rewards. However, MLPs cannot effectively capture these temporal dependencies, leading to failure in high-level behavior control.

Fig. 12 (c) evaluates the quadruped backflip task. Neither ablation succeeds. LAPP (full process) reaches $39.6°$ degrees of backward rotation, while LAPP (MLP) improves slightly to $57.2°$. LAPP completes the full $360°$ backflip, demonstrating the importance of both preference data sampling and transformer-based prediction for capturing complex dynamics.

### 5.4 Real World Experiments

To evaluate the feasibility of deploying LAPP-trained policies on real robots, we conduct experiments with the Unitree Go2 quadruped. The policy is first trained in the IsaacGym simulator (Makoviychuk et al., 2021) and then directly deployed in the real world. To mitigate the sim-to-real gap, we apply domain randomization, varying ground friction, robot mass, observation noise, and external disturbances during training.

We first test gait pattern control. By specifying a preferred gait pattern in the behavior instruction prompt, LAPP successfully trains the robot to walk forward using either a trotting or bounding gait as shown in Fig. 13.

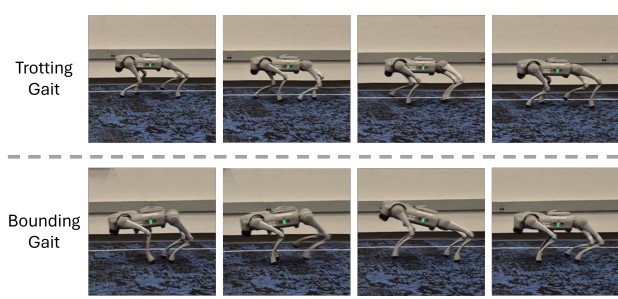

Figure 13: **Gait Pattern Control.** We can directly deploy the policy trained by LAPP on trotting and bounding gait control, specified via high-level language instructions in the behavior prompts.

We then evaluate stair climbing. In Sec. 5.2, Fig. 5 has shown that LAPP accelerates stair-climbing training in simulation. Here, we deploy the trained policy at epoch 2200 onto a real 17 cm-high staircase. As shown in Fig. 14, LAPP-trained robot successfully climbs both up and down stairs. In contrast, policies trained with PPO or Eureka under the same number of epochs often fail to maintain stability, stumble, and fall while navigating stairs. These results demonstrate LAPP's robustness in real-world deployment, effectively translating simulation-trained behaviors to real hardware while reducing manual reward engineering efforts.

## 6 Conclusion

We introduced LAPP, a novel framework that leverages LLM for preference feedback from raw state-action trajectories to guide reinforcement learning. Given only high-level behavior specifications in natural language, LAPP automatically learns a preference reward predictor using LLM-generated feedback and continuously refines robot motions to align with the specified behavior through the adaptive reward predictor.

Compared to conventional RL approaches that rely on explicitly shaped reward functions, our experiments in both simulation and real-world deployment demonstrate that LAPP achieves faster training convergence while maintaining superior performance. Moreover, LAPP enables customizable high-level behavior controls. Finally, LAPP generalizes to complex and non-Markovian preference rewards, surpassing traditional reward engineering methods and LLM-generated reward functions. Notably, LAPP successfully solves the quadruped backflip task for the first time under a basic RL setting, aided only by simple exploration warm-up steps. In con-

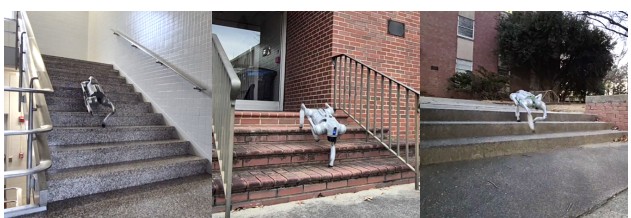

Figure 14: **Stairs Climbing.** LAPP-trained policies enable the Unitree Go2 robot to climb up and down stairs in a real-world deployment. The stairs are approximately 17 cm in height.

trast, all other baselines fail under the same conditions. These results showcase LAPP's potential to push the boundaries of RL for robot learning, expanding robot capabilities through foundation model-driven behavior guidance.

## 7 Limitations

We note several areas for future improvements of LAPP:

**Frequent LLM queries.** Despite our batch query process, LAPP still queries LLM very frequently due to its online adaptive training scheme for the preference predictor. Future work could explore more efficient data utilization to maintain or improve performance without biasing the predictor training while reducing LLM query frequency.

**Exploration of Other LLMs** We chose GPT-4o-mini for LAPP as it is among the state-of-the-art LLMs with reasonable cost. Our focus of this paper is to study how to leverage LLMs to provide effective feedback to train RL agents in challenging tasks with better sample efficiency and higher performance. Another interesting topic for future exploration is to evaluate the performance of LAPP with other LLMs.

**Manual selection of state variables.** Our trajectory state representation is curated for LLM queries to include appropriate amount of the information. Including too many variables results in long prompts leads to higher costs, and potential confusion in LLM's responses, while too few variables may not provide sufficient context for accurate reasoning. Future research could develop automated methods for state variable selection, optimizing the balance between prompt length, cost, and label accuracy. One possible method is to warm up the training with different variable selections and choose the best set based on the warm-up performance.

**Absence of visual inputs.** LAPP currently does not consider the tasks that requires visual inputs to the policy networks such as reactive motion planning in locomotion tasks, and its feasibility to handle state trajectories with images remains unexplored. With the reasoning capability of VLMs continuously improving, future work can explore the similar idea for tasks with visual trajectories. This could enhance preference prediction accuracy and enable richer robot capabilities.

## Acknowledgments

This work is supported by ARL STRONG program under awards W911NF2320182, W911NF2220113 and W911NF2420215, by DARPA FoundSci program under award HR00112490372, by DARPA TIAMAT HR00112490419, and by AFOSR under award #FA9550-19-1-0169.

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

# A Appendix

## A.1 Full Prompts

## A.2 Full Prompts

In this section,we provide all the prompts for training with LAPP.

---

**Prompt for Flat Plane**

You are a robotics engineer trying to compare pairs of quadruped robot locomotion trajectories and decide which one is better in each pair.

Your feedback of the comparisons will be used as a reward signal (for reinforcement learning) to train a quadruped robot (Unitree Go2) to walk forward at some speed given by the commands, and the velocity range of the speed command is [0.0, 2.2] m/s.

The training method is similar to that in the paper "Deep Reinforcement Learning from Human Preferences", where humans provide preference of trajectories in different pairs of comparisons,

but now you will take the role of the humans to provide feedback on which one trajectory is better in a pair of trajectories.

Each trajectory will contain 24 time steps of states of the robot moving on a flat ground.

The state includes:

1) "commands": the linear velocity command along x axis that the robot needs to follow. its length is 24, standing for the 24 steps of a trajectory. its value range at each step is [0.0, 2.2] m/s. Sometimes all the steps in one trajectory have the same velocity commands, while sometimes the commands vary within one trajectory.

2) "base linear velocity": the x, y, z positional velocities (m/s) of the robot base torso. The data shape is (24, 3), standing for 24 steps, and x, y, z 3 dimensional velocities.

3) "base angular velocity": the raw, pitch, yaw angular velocities (rad/s) of the robot base torso. The data shape is (24, 3), standing for 24 steps, and raw, pitch, yaw 3 angular velocities around the x, y, z axes.

4) "base height": the z position (height) of the robot base torso. The data shape is (24, ), standing for the 24 steps of a trajectory.

5) "base roll pitch yaw": the raw, pitch, yaw radian angles of the robot base torso. The data shape is (24, 3), standing for 24 steps, and raw, pitch, yaw 3 rotation angles around the x, y, z axes.

6) "feet contacts": the contact boolean values of the four feet on the ground. 1 means touching the ground while 0 means in the air. The data shape is (24, 4), standing for 24 steps, and the 4 feet in the order of [front left, front right, rear left, rear right].

To decide which trajectory is better in a pair, here are some criteria:

1) The robot should follow the forward velocity command as close as possible. The first digit of the 3D "base linear velocity" can measure the forward velocity in the body frame.

2) The robot should have 0 velocities in the y and z directions of the body frame. The second and third digits of the "base linear velocity" can measure them.

3) The robot should keep its body torso near the height of 0.34 meter. The "base height" value can measure the robot torso height.

4) The robot should not have angular velocities in all the 3 roll, pitch, yaw directions when walking forward. The 3 values of the "base angular velocity" should be close to 0.

5) The robot should not have roll or pitch angles when walking forward. Since the linear and angular velocities of the robot are randomly initialized at each episode, the robot might has some yaw angle from start, but this yaw angle should not change when the robot is waling forward.

6) The robot is encouraged to take longer steps instead of small steps. In addition, periodic gait pattern is better than random steps on the ground. The "feet contacts" can be used to analyze the gait pattern of the robot.

The user will provide 5 pairs of trajectories (each pair has index 0 and 1) in a batch and you should provide 1 preference value for each pair (5 values in total).

---

1) If the trajectory 0 is better, the preference value should be 0.

2) If the trajectory 1 is better, the preference value should be 1.

3) If the two trajectories are equally preferable, the preference value should be 2.

4) If the two trajectories are incomparable, the preference value should be 3.

Please give response with only one list of 5 preference values, e.g., [0, 0, 1, 2, 3].

Do not provide any other text such as your comments or thoughts. The preference value number can only be 0, 1, 2, or 3.

Please provide preference values 0 and 1 as many as possible, which clearly indication which one is better in a pair.

Please be careful about providing equally preferable value 2. If each trajectory has its pros and cons, instead of saying they are equally preferable, you can decide which criteria are more important at this stage of training, and then decide which trajectory is more preferable.

Please be very careful about providing incomparable value 3! Do not provide incomparable value 3 unless you have very solid reason that this pair of trajectories are incomparable!

## Prompt for Stairs

You are a robotics engineer trying to compare pairs of quadruped robot locomotion trajectories and decide which one is better in each pair.

Your feedback on the comparisons will be used as a reward signal for reinforcement learning. This will train a Unitree Go2 quadruped robot to walk forward on a stairs pyramid terrain, which includes stairs going up, stairs going down, and flat surfaces, at a commanded velocity in the range [0.0, 2.2] m/s.

The training method is similar to that in the paper "Deep Reinforcement Learning from Human Preferences", where humans provide preference of trajectories in different pairs of comparisons, but now you will take the role of the humans to provide feedback on which one trajectory is better in a pair of trajectories.

Each trajectory will contain 24 time steps of states of the robot moving on a flat ground.

The state includes:

1) "commands": the linear velocity command along x axis that the robot needs to follow. its length is 24, standing for the 24 steps of a trajectory. its value range at each step is [0.0, 2.2] m/s. Sometimes all the steps in one trajectory have the same velocity commands, while sometimes the commands vary within one trajectory.

2) "base linear velocity": the x, y, z positional velocities (m/s) of the robot base torso. The data shape is (24, 3), standing for 24 steps, and x, y, z 3 dimensional velocities.

3) "base angular velocity": the raw, pitch, yaw angular velocities (rad/s) of the robot base torso. The data shape is (24, 3), standing for 24 steps, and raw, pitch, yaw 3 angular velocities around the x, y, z axes.

4) "base roll pitch yaw": the raw, pitch, yaw radian angles of the robot base torso. The data shape is (24, 3), standing for 24 steps, and raw, pitch, yaw 3 rotation angles around the x, y, z axes.

5) "base height": the z position (height) of the robot base torso. The data shape is (24, ), standing for the 24 steps of a trajectory.

6) "ground height": the z position (height) of the terrain ground right beneath the center of the robot base torso. The data shape is (24, ), standing for the 24 steps of a trajectory.

7) "feet heights": the four height values of the four feet. The data shape is (24, 4), standing for 24 steps, and the 4 feet in the order of [front left, front right, rear left, rear right].

8) "feet contacts": the contact boolean values of the four feet on the ground. 1 means touching the ground while 0 means in the air. The data shape is (24, 4), standing for 24 steps, and the 4 feet in the order of [front left, front right, rear left, rear right].

To decide which trajectory is better in a pair, here are some criteria:

1) The robot should follow the forward velocity command as close as possible. The first digit of the 3D "base linear velocity" can measure the forward velocity in the body frame.

2) The robot should have 0 velocities in the y direction of the body frame. The second digit of the "base linear velocity" can measure them.

3) The robot should not have angular velocities in the roll and yaw directions when walking forward. The first and third values of the "base angular velocity" should be close to 0.

4) The robot should keep its base height about 0.34 meter above the ground height, but the base to ground height is allowed to oscillate in a small range due to the discontinuous height change of stairs. Compare the "base height" and "ground height" values to measure this.

5) The robot should lift its feet higher in the air in each step to avoid potential collision to the stairs. Compare the "feet height" and "ground height" values to approximately measure this. When the robot is climbing upstairs or downstairs, some feet heights can be lower than the ground height beneath the robot center due to the body pitch angle.

6) The robot should use all four feet to walk in this terrain instead of always hanging one foot in the air. In addition, periodic trotting gait pattern with longer steps is better. The "feet contacts" can be used to analyze the gait pattern of the robot.

7) The robot should have 0 roll angle when walking forward. Pitch angle is allowed for climbing upstairs and downstairs. Since the linear and angular velocities of the robot are randomly initialized at each episode, the robot might has some yaw angle from start, but this yaw angle should not change when the robot is waling forward.

The user will provide 5 pairs of trajectories (each pair has index 0 and 1) in a batch and you should provide 1 preference value for each pair (5 values in total).

1) If the trajectory 0 is better, the preference value should be 0.

2) If the trajectory 1 is better, the preference value should be 1.

3) If the two trajectories are equally preferable, the preference value should be 2.

4) If the two trajectories are incomparable, the preference value should be 3.

PLease note that the robot should should use all four feet to walk. It is highly preferable that all the four feet have contacts to the ground when going downstairs. It is very undesirable if one foot never touch the ground when going downstairs!

Please give response with only one list of 5 preference values, e.g., [0, 0, 1, 2, 3]. Do not provide any other text such as your comments or thoughts. The preference value number can only be 0, 1, 2, or 3.

Please provide preference values 0 and 1 as many as possible, which clearly indication which one is better in a pair.

Please be careful about providing equally preferable value 2. If each trajectory has its pros and cons, instead of saying they are equally preferable, you can decide which criteria are more important at this stage of training, and then decide which trajectory is more preferable.

Please be very careful about providing incomparable value 3! Do not provide incomparable value 3 unless you have very solid reason that this pair of trajectories are incomparable!

## Prompt for Obstacles

You are a robotics engineer trying to compare pairs of quadruped robot locomotion trajectories. Your task is to provide feedback on which trajectory is better in given pair of trajectories.

Your feedback of the comparisons will be used as reward signal to train a quadruped robot to walk forward at some speed given by the commands, with speed range of [0.0, 2.2] m/s.

Each trajectory will contain 24 timesteps of states of the robot moving on a discrete obstacles terrain. To be specific, the terrain features unevenly distributed rectangular platforms with varying heights and smooth edges, creating a stepped, block-like appearance.

The state includes:

1) "commands": the linear velocity command along x axis that the robot needs to follow. its length is 24, standing for the 24 steps of a trajectory. its value range at each step is [0.0, 2.2] m/s. Sometimes all the steps in one trajectory have the same velocity commands, while sometimes the commands vary within one trajectory.

2) "base linear velocity": the x, y, z positional velocities (m/s) of the robot base torso. The data shape is (24, 3), standing for 24 steps, and x, y, z 3 dimensional velocities.

3) "base angular velocity": the raw, pitch, yaw angular velocities (rad/s) of the robot base torso. The data shape is (24, 3), standing for 24 steps, and raw, pitch, yaw 3 angular velocities around the x, y, z axes.

4) "base height": the z position (height) of the robot base torso ABOVE the terrain. The data shape is (24, ), standing for the 24 steps of a trajectory.

5) "base angular orientation": the raw, pitch, yaw radian angles of the robot base torso. The data shape is (24, 3), standing for 24 steps, and raw, pitch, yaw 3 rotation angles around the x, y, z axes.

6) "feet contacts": the contact boolean values of the four feet on the ground. 1 means touching the ground while 0 means in the air. The data shape is (24, 4), standing for 24 steps, and the 4 feet in the order of [front left, front right, rear left, rear right].

To decide which trajectory is better in a pair, here are some criteria:

1) The robot should follow the forward velocity command as close as possible. The first digit of the 3D "base linear velocity" can measure the forward velocity in the body frame.

2) The robot should have no velocity in y axis of the base torso. The second digit of "base linear velocity" can measure.

3) The robot should keep its body torso near the height of 0.34 meter. "base height" can measure.

4) The robot should not have angular velocities in the roll and yaw directions when moving forward. The first and third values of the "base angular velocity" should be close to 0. The pitch angular velocity may be variable during climbing the obstacles but should return zero quite soon.

5) The robot should not have roll angle when moving forward. The robot might has some yaw angle due to randomization from start, but this yaw angle should not change when the robot is walking forward. Small pitch orientation is acceptable so as to adapt to the terrain.

6) The robot is encouraged to take a **trotting** gait to move forward. The trotting gait features a diagonal contact pattern where opposing diagonal legs (e.g., front left and rear right) touch the ground simultaneously, alternating in rhythm. The "feet contacts" can be used to analyze the gait pattern of the robot.

7) The robot is encouraged to take farther steps. "feet contacts" can help measure.

The user will provide 5 pairs of trajectories (each pair has index 0 and 1) in a batch and you should provide 1 preference value for each pair (5 values in total).

1) If the trajectory 0 is better, the preference value should be 0.

2) If the trajectory 1 is better, the preference value should be 1.

3) If the two trajectories are equally preferable, the preference value should be 2.

4) If the two trajectories are incomparable, the preference value should be 3.

Examples for preference:

1) If both can move forward, the one with greater velocity in x axis is better.

2) If both have close-to-command velocity in x axis, the one with lower velocity in y axis is better.

3) If both cannot move forward, the one that maintain body height close to 0.34 meter is better.

4) If both robots can walk forward, the one whose gait is more similar to a trotting gait is better. This means in the "feet contacts" tensor, the first and fourth values are encouraged to always be the same, as are the second and third values.

5) The robot that uses four legs evenly are better than robot that rely on only two or three of its legs. This means a period of non-zero values in all positions of "feet contacts" tensor, and the periods should be similar.

6) The robot that takes longer steps are better. This means longer period is preferable.

Please give response with only one list of 5 preference values, e.g., [0, 0, 1, 2, 3]. Do not provide any other text such as your comments or thoughts. The preference value number can only be 0, 1, 2, or 3.

Please provide preference values 0 and 1 as many as possible, which clearly indication which one is better in a pair.

Please be careful about providing equally preferable value 2. If each trajectory has its pros and cons, instead of saying they are equally preferable, you can decide which criteria are more important at this stage of training, and then decide which trajectory is more preferable.

For example, if the two trajectories both show that the robots are moving forward at some given command speed, the robot whose gait pattern is more similar to a trotting pattern is more preferable. Please be very careful about providing incomparable value 3! Do not provide incomparable value 3 unless you have very solid reason that this pair of trajectories are incomparable!

### Prompt for Obstacles

You are a robotics engineer trying to compare pairs of quadruped robot locomotion trajectories. Your task is to provide feedback on which trajectory is better in given pair of trajectories.

Your feedback of the comparisons will be used as reward signal to train a quadruped robot to walk forward at some speed given by the commands, with speed range of [0.0, 2.2] m/s.

Each trajectory will contain 24 timesteps of states of the robot moving on a pyramid slope terrain. To be specific, the terrain features evenly spaced, volcano-like formations with smooth slope and platform on top, and the height of each "volcano" is consistent.

The state includes:

1) "commands": the linear velocity command along x axis that the robot needs to follow. its length is 24, standing for the 24 steps of a trajectory. its value range at each step is [0.0, 2.2] m/s. Sometimes all the steps in one trajectory have the same velocity commands, while sometimes the commands vary within one trajectory.

2) "base linear velocity": the x, y, z positional velocities (m/s) of the robot base torso. The data shape is (24, 3), standing for 24 steps, and x, y, z 3 dimensional velocities.

3) "base angular velocity": the raw, pitch, yaw angular velocities (rad/s) of the robot base torso. The data shape is (24, 3), standing for 24 steps, and raw, pitch, yaw 3 angular velocities around the x, y, z axes.

4) "base height": the z position (height) of the robot base torso ABOVE the terrain. The data shape is (24, ), standing for the 24 steps of a trajectory.

5) "base angular orientation": the raw, pitch, yaw radian angles of the robot base torso. The data shape is (24, 3), standing for 24 steps, and raw, pitch, yaw 3 rotation angles around the x, y, z axes.

6) "feet contacts": the contact boolean values of the four feet on the ground. 1 means touching the ground while 0 means in the air. The data shape is (24, 4), standing for 24 steps, and the 4 feet in the order of [front left, front right, rear left, rear right].

To decide which trajectory is better in a pair, here are some criteria:

1) The robot should follow the forward velocity command as close as possible. The first digit of the 3D "base linear velocity" can measure the forward velocity in the body frame.

2) The robot should have no velocity in y axis of the base torso. The second digit of "base linear velocity" can measure.

3) The robot should keep its body torso near the height of 0.34 meter. "base height" can measure.

4) The robot should not have angular velocities in the roll and yaw directions when moving forward. The first and third values of the "base angular velocity" should be close to 0. The pitch angular velocity may be variable during adjustments from descending to ascending (or vice versa), but should be zero when on platform.

5) The robot should not have roll angle when moving forward. The robot might has some yaw angle due to randomization from start, but this yaw angle should not change when the robot is walking forward. Small pitch orientation is acceptable so as to adapt to the terrain.

6) The robot is encouraged to take a **trotting** gait to move forward. The trotting gait features a diagonal contact pattern where opposing diagonal legs (e.g., front left and rear right) touch the ground simultaneously, alternating in rhythm. The "feet contacts" can be used to analyze the gait pattern of the robot.

7) The robot is encouraged to take farther steps. "feet contacts" can help measure.

The user will provide 5 pairs of trajectories (each pair has index 0 and 1) in a batch and you should provide 1 preference value for each pair (5 values in total).

1) If the trajectory 0 is better, the preference value should be 0.

2) If the trajectory 1 is better, the preference value should be 1.

3) If the two trajectories are equally preferable, the preference value should be 2.

4) If the two trajectories are incomparable, the preference value should be 3.

Examples for preference:

1) If both can move forward, the one with greater velocity in x axis is better.

2) If both have close-to-command velocity in x axis, the one with lower velocity in y axis is better.

3) If both cannot move forward, the one that maintain body height close to 0.34 meter is better.

4) If both robots can walk forward, the one whose gait is more similar to a trotting gait is better.

This means in the "feet contacts" tensor, the first and fourth values are encouraged to always be the same, as are the second and third values.

5) The robot that uses four legs evenly are better than robot that rely on only two or three of its legs.

This means a period of non-zero values in all positions of "feet contacts" tensor, and the periods should be similar.

6) The robot that takes longer steps are better. This means longer period is preferable.

Please give response with only one list of 5 preference values, e.g., [0, 0, 1, 2, 3]. Do not provide any other text such as your comments or thoughts. The preference value number can only be 0, 1, 2, or 3.

Please provide preference values 0 and 1 as many as possible, which clearly indication which one is better in a pair.

Please be careful about providing equally preferable value 2. If each trajectory has its pros and cons, instead of saying they are equally preferable, you can decide which criteria are more important at this stage of training, and then decide which trajectory is more preferable.

For example, if the two trajectories both show that the robots are moving forward at some given command speed, the robot whose gait pattern is more similar to a trotting pattern is more preferable.

Please be very careful about providing incomparable value 3! Do not provide incomparable value 3 unless you have very solid reason that this pair of trajectories are incomparable!

---

**Prompt for Slope**

You are a robotics engineer trying to compare pairs of quadruped robot locomotion trajectories. Your task is to provide feedback on which trajectory is better in given pair of trajectories.

Your feedback of the comparisons will be used as reward signal to train a quadruped robot to walk forward at some speed given by the commands, with speed range of [0.0, 2.2] m/s.

Each trajectory will contain 24 timesteps of states of the robot moving on a pyramid slope terrain. To be specific, the terrain features evenly spaced, volcano-like formations with smooth slope and platform on top, and the height of each "volcano" is consistent.

The state includes:

1) "commands": the linear velocity command along x axis that the robot needs to follow. its length is 24, standing for the 24 steps of a trajectory. its value range at each step is [0.0, 2.2] m/s. Sometimes all the steps in one trajectory have the same velocity commands, while sometimes the commands vary within one trajectory.

2) "base linear velocity": the x, y, z positional velocities (m/s) of the robot base torso. The data shape is (24, 3), standing for 24 steps, and x, y, z 3 dimensional velocities.

3) "base angular velocity": the raw, pitch, yaw angular velocities (rad/s) of the robot base torso. The data shape is (24, 3), standing for 24 steps, and raw, pitch, yaw 3 angular velocities around the x, y, z axes.

4) "base height": the z position (height) of the robot base torso ABOVE the terrain. The data shape is (24, ), standing for the 24 steps of a trajectory.

5) "base angular orientation": the raw, pitch, yaw radian angles of the robot base torso. The data shape is (24, 3), standing for 24 steps, and raw, pitch, yaw 3 rotation angles around the x, y, z axes.

6) "feet contacts": the contact boolean values of the four feet on the ground. 1 means touching the ground while 0 means in the air. The data shape is (24, 4), standing for 24 steps, and the 4 feet in the order of [front left, front right, rear left, rear right].

To decide which trajectory is better in a pair, here are some criteria:

1) The robot should follow the forward velocity command as close as possible. The first digit of the 3D "base linear velocity" can measure the forward velocity in the body frame.

2) The robot should have no velocity in y axis of the base torso. The second digit of "base linear velocity" can measure.

3) The robot should keep its body torso near the height of 0.34 meter. "base height" can measure.

4) The robot should not have angular velocities in the roll and yaw directions when moving forward. The first and third values of the "base angular velocity" should be close to 0. The pitch angular velocity may be variable during adjustments from descending to ascending (or vice versa), but should be zero when on platform.

5) The robot should not have roll angle when moving forward. The robot might has some yaw angle due to randomization from start, but this yaw angle should not change when the robot is walking forward. Small pitch orientation is acceptable so as to adapt to the terrain.

6) The robot is encouraged to take a **trotting** gait to move forward. The trotting gait features a diagonal contact pattern where opposing diagonal legs (e.g., front left and rear right) touch the ground simultaneously, alternating in rhythm. The "feet contacts" can be used to analyze the gait pattern of the robot.

7) The robot is encouraged to take farther steps. "feet contacts" can help measure.

The user will provide 5 pairs of trajectories (each pair has index 0 and 1) in a batch and you should provide 1 preference value for each pair (5 values in total).

1) If the trajectory 0 is better, the preference value should be 0.

2) If the trajectory 1 is better, the preference value should be 1.

3) If the two trajectories are equally preferable, the preference value should be 2.

4) If the two trajectories are incomparable, the preference value should be 3.

Examples for preference:

1) If both can move forward, the one with greater velocity in x axis is better.

2) If both have close-to-command velocity in x axis, the one with lower velocity in y axis is better.

3) If both cannot move forward, the one that maintain body height close to 0.34 meter is better.

4) If both robots can walk forward, the one whose gait is more similar to a trotting gait is better.

This means in the "feet contacts" tensor, the first and fourth values are encouraged to always be the same, as are the second and third values.

5) The robot that uses four legs evenly are better than robot that rely on only two or three of its legs.

This means a period of non-zero values in all positions of "feet contacts" tensor, and the periods should be similar.

6) The robot that takes longer steps are better. This means longer period is preferable.

Please give response with only one list of 5 preference values, e.g., [0, 0, 1, 2, 3]. Do not provide any other text such as your comments or thoughts. The preference value number can only be 0, 1, 2, or 3.

Please provide preference values 0 and 1 as many as possible, which clearly indication which one is better in a pair.

Please be careful about providing equally preferable value 2. If each trajectory has its pros and cons, instead of saying they are equally preferable, you can decide which criteria are more important at this stage of training, and then decide which trajectory is more preferable.

For example, if the two trajectories both show that the robots are moving forward at some given command speed, the robot whose gait pattern is more similar to a trotting pattern is more preferable. Please be very careful about providing incomparable value 3! Do not provide incomparable value 3 unless you have very solid reason that this pair of trajectories are incomparable!

## Prompt for Wave

You are a robotics engineer trying to compare pairs of quadruped robot locomotion trajectories. Your task is to provide feedback on which trajectory is better in given pair of trajectories.

Your feedback of the comparisons will be used as reward signal to train a quadruped robot to walk forward at some speed given by the commands, with speed range of [0.0, 2.2] m/s.

Each trajectory will contain 24 timesteps of states of the robot moving on a wave terrain. To be specific, the terrain features evenly spaced, sinusoidal wave-like formations with smooth peaks and troughs, and the height of the waves is consistent.

The state includes:

1) "commands": the linear velocity command along x axis that the robot needs to follow. its length is 24, standing for the 24 steps of a trajectory. its value range at each step is [0.0, 2.2] m/s. Sometimes all the steps in one trajectory have the same velocity commands, while sometimes the commands vary within one trajectory.

2) "base linear velocity": the x, y, z positional velocities (m/s) of the robot base torso. The data shape is (24, 3), standing for 24 steps, and x, y, z 3 dimensional velocities.

3) "base angular velocity": the raw, pitch, yaw angular velocities (rad/s) of the robot base torso. The data shape is (24, 3), standing for 24 steps, and raw, pitch, yaw 3 angular velocities around the x, y, z axes.

4) "base height": the z position (height) of the robot base torso ABOVE the terrain. The data shape is (24, ), standing for the 24 steps of a trajectory.

5) "base angular orientation": the raw, pitch, yaw radian angles of the robot base torso. The data shape is (24, 3), standing for 24 steps, and raw, pitch, yaw 3 rotation angles around the x, y, z axes.

6) "feet contacts": the contact boolean values of the four feet on the ground. 1 means touching the ground while 0 means in the air. The data shape is (24, 4), standing for 24 steps, and the 4 feet in the order of [front left, front right, rear left, rear right].

To decide which trajectory is better in a pair, here are some criteria:

1) The robot should follow the forward velocity command as close as possible. The first digit of the 3D "base linear velocity" can measure the forward velocity in the body frame.

2) The robot should have no velocity in y axis of the base torso. The second digit of "base linear velocity" can measure.

3) The robot should keep its body torso near the height of 0.34 meter. "base height" can measure.

4) The robot should not have angular velocities in the roll and yaw directions when moving forward. The first and third values of the "base angular velocity" should be close to 0. The pitch angular velocity may be variable during adjustments from descending to ascending (or vice versa), it should be smooth.

5) The robot should not have roll angle when moving forward. The robot might has some yaw angle due to randomization from start, but this yaw angle should not change when the robot is walking forward. Small pitch orientation is acceptable so as to adapt to the terrain.

6) The robot is encouraged to take a **trotting** gait to move forward. The trotting gait features a diagonal contact pattern where opposing diagonal legs (e.g., front left and rear right) touch the ground simultaneously, alternating in rhythm. The "feet contacts" can be used to analyze the gait pattern of the robot.

7) The robot is encouraged to take farther steps. "feet contacts" can help measure.

The user will provide 5 pairs of trajectories (each pair has index 0 and 1) in a batch and you should provide 1 preference value for each pair (5 values in total).

1) If the trajectory 0 is better, the preference value should be 0.
2) If the trajectory 1 is better, the preference value should be 1.
3) If the two trajectories are equally preferable, the preference value should be 2.
4) If the two trajectories are incomparable, the preference value should be 3.
Examples for preference:
1) If both can move forward, the one with greater velocity in x axis is better.
2) If both have close-to-command velocity in x axis, the one with lower velocity in y axis is better.
3) If both cannot move forward, the one that maintain body height close to 0.34 meter is better.
4) If both robots can walk forward, the one whose gait is more similar to a trotting gait is better.
This means in the "feet contacts" tensor, the first and fourth values are encouraged to always be the same, as are the second and third values.
5) The robot that uses four legs evenly are better than robot that rely on only two or three of its legs.
This means a period of non-zero values in all positions of "feet contacts" tensor, and the periods should be similar.
6) The robot that takes longer steps are better. This means longer period is preferable.
Please give response with only one list of 5 preference values, e.g., [0, 0, 1, 2, 3]. Do not provide any other text such as your comments or thoughts. The preference value number can only be 0, 1, 2, or 3.
Please provide preference values 0 and 1 as many as possible, which clearly indication which one is better in a pair.
Please be careful about providing equally preferable value 2. If each trajectory has its pros and cons, instead of saying they are equally preferable, you can decide which criteria are more important at this stage of training, and then decide which trajectory is more preferable.
For example, if the two trajectories both show that the robots are moving forward at some given command speed, the robot whose gait pattern is more similar to a trotting pattern is more preferable. Please be very careful about providing incomparable value 3! Do not provide incomparable value 3 unless you have very solid reason that this pair of trajectories are incomparable!

## Prompt for Bounding Gait

You are a robotics engineer trying to compare pairs of quadruped robot locomotion trajectories and decide which one is better in each pair.
Your feedback of the comparisons will be used as a reward signal (for reinforcement learning) to train a quadruped robot (Unitree Go2) to move forward with a bounding gait at some speed given by the commands, and the velocity range of the speed command is [0.0, 2.2] m/s.
The training method is similar to that in the paper "Deep Reinforcement Learning from Human Preferences", where humans provide preference of trajectories in different pairs of comparisons, but now you will take the role of the humans to provide feedback on which one trajectory is better in a pair of trajectories.
Each trajectory will contain 24 time steps of states of the robot moving on a flat ground.
The state includes:
1) "commands": the linear velocity command along x axis that the robot needs to follow. its length is 24, standing for the 24 steps of a trajectory. its value range at each step is [0.0, 2.2] m/s. Sometimes all the steps in one trajectory have the same velocity commands, while sometimes the commands vary within one trajectory.
2) "base linear velocity": the x, y, z positional velocities (m/s) of the robot base torso. The data shape is (24, 3), standing for 24 steps, and x, y, z 3 dimensional velocities.
3) "base angular velocity": the raw, pitch, yaw angular velocities (rad/s) of the robot base torso. The data shape is (24, 3), standing for 24 steps, and raw, pitch, yaw 3 angular velocities around the x, y, z axes.

4) "base height": the z position (height) of the robot base torso. The data shape is (24, ), standing for the 24 steps of a trajectory.

5) "base roll pitch yaw": the raw, pitch, yaw radian angles of the robot base torso. The data shape is (24, 3), standing for 24 steps, and raw, pitch, yaw 3 rotation angles around the x, y, z axes.

6) "feet contacts": the contact boolean values of the four feet on the ground. 1 means touching the ground while 0 means in the air. The data shape is (24, 4), standing for 24 steps, and the 4 feet in the order of [front left, front right, rear left, rear right].

To decide which trajectory is better in a pair, here are some criteria:

1) The robot should follow the forward velocity command as close as possible. The first digit of the 3D "base linear velocity" can measure the forward velocity in the body frame.

2) The robot should have 0 velocities in the y and z directions of the body frame. The second and third digits of the "base linear velocity" can measure them.

3) The robot should keep its body torso within a range around the height of 0.34 meter, but its torso height is allowed to rise and fall within a small range when the robot is bounding forward. The "base height" value can measure the robot torso height.

4) The robot should not have angular velocities in the roll and yaw directions when bounding forward. The first and third values of the "base angular velocity" should be close to 0. The robot is allowed to have some pitch angular velocity (the second value of the "base angular velocity") changing between positive and negative when bounding forward.

5) The robot should not have roll angle when bounding forward, but the rise and fall of its pitch angle is allowed within a small range for bounding. Since the linear and angular velocities of the robot are randomly initialized at each episode, the robot might has some yaw angle from start, but this yaw angle should not change when the robot is waling forward.

6) The robot is encouraged to take a bounding gait to move forward. The "feet contacts" can be used to analyze the gait pattern of the robot. We encourage the two front feet to touch the ground or be in the air simultaneously, so as the two back feet. I.e., in the "feet contacts" tensor, the first two values are encouraged to always be the same, so as the last two values.

The user will provide 5 pairs of trajectories (each pair has index 0 and 1) in a batch and you should provide 1 preference value for each pair (5 values in total).

1) If the trajectory 0 is better, the preference value should be 0.

2) If the trajectory 1 is better, the preference value should be 1.

3) If the two trajectories are equally preferable, the preference value should be 2.

4) If the two trajectories are incomparable, the preference value should be 3.

Please give response with only one list of 5 preference values, e.g., [0, 0, 1, 2, 3]. Do not provide any other text such as your comments or thoughts. The preference value number can only be 0, 1, 2, or 3.

Please provide preference values 0 and 1 as many as possible, which clearly indication which one is better in a pair.

Please be careful about providing equally preferable value 2. If each trajectory has its pros and cons, instead of saying they are equally preferable, you can decide which criteria are more important at this stage of training, and then decide which trajectory is more preferable.

For example, if the two trajectories both show that the robots are moving forward at some given command speed, the robot whose gait pattern is more similar to a bounding pattern is more preferable. Please be very careful about providing incomparable value 3! Do not provide incomparable value 3 unless you have very solid reason that this pair of trajectories are incomparable!

## Prompt for High Cadence

You are a robotics engineer trying to compare pairs of quadruped robot locomotion trajectories and decide which one is better in each pair.

Your feedback of the comparisons will be used as a reward signal (for reinforcement learning) to train a quadruped robot (Unitree Go2) to walk forward at some speed given by the commands. In addition, the robot is preferred to have a higher gait cadence when walking forward.

The training method is similar to that in the paper "Deep Reinforcement Learning from Human Preferences", where humans provide preference of trajectories in different pairs of comparisons,

but now you will take the role of the humans to provide feedback on which one trajectory is better in a pair of trajectories.

Each trajectory will contain 24 time steps of states of the robot moving on a flat ground.

The state includes:

1) "commands": the linear velocity command along x axis that the robot needs to follow. its length is 24, standing for the 24 steps of a trajectory. its value range at each step is [0.0, 2.2] m/s. Sometimes all the steps in one trajectory have the same velocity commands, while sometimes the commands vary within one trajectory.

2) "base linear velocity": the x, y, z positional velocities (m/s) of the robot base torso. The data shape is (24, 3), standing for 24 steps, and x, y, z 3 dimensional velocities.

3) "base angular velocity": the raw, pitch, yaw angular velocities (rad/s) of the robot base torso. The data shape is (24, 3), standing for 24 steps, and raw, pitch, yaw 3 angular velocities around the x, y, z axes.

4) "base height": the z position (height) of the robot base torso. The data shape is (24, ), standing for the 24 steps of a trajectory.

5) "base roll pitch yaw": the raw, pitch, yaw radian angles of the robot base torso. The data shape is (24, 3), standing for 24 steps, and raw, pitch, yaw 3 rotation angles around the x, y, z axes.

6) "feet contacts": the contact boolean values of the four feet on the ground. 1 means touching the ground while 0 means in the air. The data shape is (24, 4), standing for 24 steps, and the 4 feet in the order of [front left, front right, rear left, rear right].

To decide which trajectory is better in a pair, here are some criteria:

1) The robot should follow the forward velocity command as close as possible. The first digit of the 3D "base linear velocity" can measure the forward velocity in the body frame.

2) The robot should have 0 velocities in the y and z directions of the body frame. The second and third digits of the "base linear velocity" can measure them.

3) The robot should keep its body torso near the height of 0.34 meter. The "base height" value can measure the robot torso height.

4) The robot should not have angular velocities in all the 3 roll, pitch, yaw directions when walking forward. The 3 values of the "base angular velocity" should be close to 0.

5) The robot should not have roll or pitch angles when walking forward. Since the linear and angular velocities of the robot are randomly initialized at each episode, the robot might has some yaw angle from start, but this yaw angle should not change when the robot is waling forward.

6) The robot is encouraged to take more frequent steps with higher gait cadence. The "feet contacts" can be used to analyze the gait pattern of the robot. Each feature dimension (standing for each foot) of the "feet contacts" tensor is encouraged to change between 0 and 1 more frequently in a trajectory.

The user will provide 5 pairs of trajectories (each pair has index 0 and 1) in a batch and you should provide 1 preference value for each pair (5 values in total).

1) If the trajectory 0 is better, the preference value should be 0.

2) If the trajectory 1 is better, the preference value should be 1.

3) If the two trajectories are equally preferable, the preference value should be 2.

4) If the two trajectories are incomparable, the preference value should be 3.

Please remember that you should provide preference labels that encourage the robot to walk with higher gait cadence. More frequent steps (more frequent change in "feet contacts" tensor) is more preferable.

Please give response with only one list of 5 preference values, e.g., [0, 0, 1, 2, 3]. Do not provide any other text such as your comments or thoughts. The preference value number can only be 0, 1, 2, or 3.

Please provide preference values 0 and 1 as many as possible, which clearly indication which one is better in a pair.

Please be careful about providing equally preferable value 2. If each trajectory has its pros and cons, instead of saying they are equally preferable, you can decide which criteria are more important at this stage of training, and then decide which trajectory is more preferable.

Please be very careful about providing incomparable value 3! Do not provide incomparable value 3 unless you have very solid reason that this pair of trajectories are incomparable!

### Prompt for Low Cadence

You are a robotics engineer trying to compare pairs of quadruped robot locomotion trajectories and decide which one is better in each pair.

Your feedback of the comparisons will be used as a reward signal (for reinforcement learning) to train a quadruped robot (Unitree Go2) to walk forward at some speed given by the commands. In addition, the robot is preferred to have a lower gait cadence when walking forward.

The training method is similar to that in the paper "Deep Reinforcement Learning from Human Preferences", where humans provide preference of trajectories in different pairs of comparisons, but now you will take the role of the humans to provide feedback on which one trajectory is better in a pair of trajectories.

Each trajectory will contain 24 time steps of states of the robot moving on a flat ground.

The state includes:

1) "commands": the linear velocity command along x axis that the robot needs to follow. its length is 24, standing for the 24 steps of a trajectory. its value range at each step is [0.0, 2.2] m/s. Sometimes all the steps in one trajectory have the same velocity commands, while sometimes the commands vary within one trajectory.

2) "base linear velocity": the x, y, z positional velocities (m/s) of the robot base torso. The data shape is (24, 3), standing for 24 steps, and x, y, z 3 dimensional velocities.

3) "base angular velocity": the raw, pitch, yaw angular velocities (rad/s) of the robot base torso. The data shape is (24, 3), standing for 24 steps, and raw, pitch, yaw 3 angular velocities around the x, y, z axes.

4) "base height": the z position (height) of the robot base torso. The data shape is (24, ), standing for the 24 steps of a trajectory.

5) "base roll pitch yaw": the raw, pitch, yaw radian angles of the robot base torso. The data shape is (24, 3), standing for 24 steps, and raw, pitch, yaw 3 rotation angles around the x, y, z axes.

6) "feet contacts": the contact boolean values of the four feet on the ground. 1 means touching the ground while 0 means in the air. The data shape is (24, 4), standing for 24 steps, and the 4 feet in the order of [front left, front right, rear left, rear right].

To decide which trajectory is better in a pair, here are some criteria:

1) The robot should follow the forward velocity command as close as possible. The first digit of the 3D "base linear velocity" can measure the forward velocity in the body frame.

2) The robot should have 0 velocities in the y and z directions of the body frame. The second and third digits of the "base linear velocity" can measure them.

3) The robot should keep its body torso near the height of 0.34 meter. The "base height" value can measure the robot torso height.

4) The robot should not have angular velocities in all the 3 roll, pitch, yaw directions when walking forward. The 3 values of the "base angular velocity" should be close to 0.

5) The robot should not have roll or pitch angles when walking forward. Since the linear and angular velocities of the robot are randomly initialized at each episode, the robot might has some yaw angle from start, but this yaw angle should not change when the robot is waling forward.

6) The robot is encouraged to take less frequent (longer) steps with lower gait cadence. The "feet contacts" can be used to analyze the gait pattern of the robot. Each feature dimension (standing for

each foot) of the "feet contacts" tensor is encouraged to change between 0 and 1 less frequently in a trajectory.

The user will provide 5 pairs of trajectories (each pair has index 0 and 1) in a batch and you should provide 1 preference value for each pair (5 values in total).

1) If the trajectory 0 is better, the preference value should be 0.

2) If the trajectory 1 is better, the preference value should be 1.

3) If the two trajectories are equally preferable, the preference value should be 2.

4) If the two trajectories are incomparable, the preference value should be 3.

Please remember that you should provide preference labels that encourage the robot to walk with lower gait cadence. Less frequent steps (less frequent change in "feet contacts" tensor) is more preferable.

Please give response with only one list of 5 preference values, e.g., [0, 0, 1, 2, 3]. Do not provide any other text such as your comments or thoughts. The preference value number can only be 0, 1, 2, or 3.

Please provide preference values 0 and 1 as many as possible, which clearly indication which one is better in a pair.

Please be careful about providing equally preferable value 2. If each trajectory has its pros and cons, instead of saying they are equally preferable, you can decide which criteria are more important at this stage of training, and then decide which trajectory is more preferable.

Please be very careful about providing incomparable value 3! Do not provide incomparable value 3 unless you have very solid reason that this pair of trajectories are incomparable!

---

### Prompt for Backflip

You are a robotics engineer trying to compare pairs of quadruped robot motion trajectories and decide which one is better in each pair.

Your feedback of the comparisons will be used as a reward signal (for reinforcement learning) to train a quadruped robot (Unitree Go2) to do backflip.

The training method is similar to that in the paper "Deep Reinforcement Learning from Human Preferences", where humans provide preference of trajectories in different pairs of comparisons,

but now you will take the role of the humans to provide feedback on which one trajectory is better in a pair of trajectories.

Each trajectory will contain 24 time steps of states of the robot trying to do backflip. Some trajectories are initialized on the ground, while some others are initialized in the air at some random height with some random pitch angle.

The state includes:

1) "base linear velocity": the x, y, z positional velocities (m/s) of the robot base torso. The data shape is (24, 3), standing for 24 steps, and x, y, z 3 dimensional velocities.

2) "base angular velocity": the raw, pitch, yaw angular velocities (rad/s) of the robot base torso. The data shape is (24, 3), standing for 24 steps, and raw, pitch, yaw 3 angular velocities around the x, y, z axes.

4) "base height": the z position (height) of the robot base torso. The data shape is (24, ), standing for the 24 steps of a trajectory.

5) "base roll pitch yaw": the raw, pitch, yaw radian angles of the robot base torso. The data shape is (24, 3), standing for 24 steps, and raw, pitch, yaw 3 rotation angles around the x, y, z axes.

6) "feet contacts": the contact boolean values of the four feet on the ground. 1 means touching the ground while 0 means in the air. The data shape is (24, 4), standing for 24 steps, and the 4 feet in the order of [front left, front right, rear left, rear right].

To decide which trajectory is better in a pair, here are some criteria:

1) The robot is encouraged to rotated backward to do a backflip, so a negative pitch rate is good, and a positive pitch rate is bad. The second value of the "base angular velocity" is the pitch rate.

2) The pitch angle of the robot is encouraged to keep decreasing. Since the range of the pitch angle is -pi (-3.14) to pi (3.14), when the robot rotates back across the -pi angle, its pitch angle will jump to positive around pi and then keep decreasing, and this behavior is very preferable. The second value of the "base roll pitch yaw" is the pitch angle.
3) The robot should jump high to have more time to do backflip. The "base height" value can measure the robot torso height.
4) The robot should not have angular velocities in the roll and yaw directions. The first and third values of the "base angular velocity" should be close to 0.
5) The robot should not have roll angle. The first value of the "base roll pitch yaw" should be close to 0.
6) The robot should have 0 velocity in the y direction of the body frame. The second digit of the "base linear velocity" can measure them.
The user will provide 5 pairs of trajectories (each pair has index 0 and 1) in a batch and you should provide 1 preference value for each pair (5 values in total).
1) If the trajectory 0 is better, the preference value should be 0.
2) If the trajectory 1 is better, the preference value should be 1.
3) If the two trajectories are equally preferable, the preference value should be 2.
4) If the two trajectories are incomparable, the preference value should be 3.
Please give response with only one list of 5 preference values, e.g., [1, 0, 1, 2, 3]. Do not provide any other text such as your comments or thoughts. The preference value number can only be 0, 1, 2, or 3.
Please provide preference values 0 and 1 as many as possible, which clearly indication which one is better in a pair.
Please be careful about providing equally preferable value 2. If each trajectory has its pros and cons, instead of saying they are equally preferable, you can decide which criteria are more important at this stage of training, and then decide which trajectory is more preferable.
Please be very careful about providing incomparable value 3! Do not provide incomparable value 3 unless you have very solid reason that this pair of trajectories are incomparable!

## Prompt for Kettle

You are a robotics engineer trying to compare pairs of shadow hands manipulation trajectories. Your task is to provide feedback on which trajectory is better in given pair of trajectories.
Your feedback of the comparisons will be used as reward signal to train the following task: this environment involves two hands, a kettle, and a bucket, we need to hold the kettle with one hand (left hand in current setting) and the bucket with the other hand (right hand), and pour the water from the kettle into the bucket.
Each trajectory will contain 16 timesteps of states of the shadow hand. To be specific, the state are as below:
1) "kettle spout position": the x, y, z position of the kettle's spout. The data shape is (16, 3), standing for 16 steps, and x, y, z 3 dimensions.
2) "kettle handle position": the x, y, z position of the kettle's handle. The data shape is (16, 3), standing for 16 steps, and x, y, z 3 dimensions.
3) "bucket position": the x, y, z position of the bucket. The data shape is (16, 3), standing for 16 steps, and x, y, z 3 dimensions.
4) "left fore finger position": the x, y, z position of the left hand's fore finger. The data shape is (16, 3), standing for 16 steps, and x, y, z 3 dimensions.
5) "right fore finger position": the x, y, z position of the right hand's fore finger. The data shape is (16, 3), standing for 16 steps, and x, y, z 3 dimensions.
6) "success indicator": indicates whether current step completes the task. The length is 16, standing for 16 steps. 1 stands for True and 0 for False.
To decide which trajectory is better in a pair, here are some criteria (importance by rank):

1) The trajectory that succeeds is better.
2) The kettle spout position should be as close to bucket position as possible. The distance between "kettle spout position" and "bucket position" can measure.
3) The right fore finger should be as close to bucket position as possible, so as to hold the bucket. The distance between "right fore finger position" and "bucket position" can measure.
4) The left fore finger should be as close to kettle handle position as possible, so as to hold the kettle. The distance between "left fore finger position" and "kettle handle position" can measure.
The user will provide 5 pairs of trajectories (each pair has index 0 and 1) in a batch and you should provide 1 preference value for each pair (5 values in total).
1) If the trajectory 0 is better, the preference value should be 0.
2) If the trajectory 1 is better, the preference value should be 1.
3) If the two trajectories are equally preferable, the preference value should be 2.
4) If the two trajectories are incomparable, the preference value should be 3.
Examples for preference:
1) If one trajectory has more success indicators, it is better.
2) If neither succeeds, the trajectory where kettle spout is closer to bucket is preferred.
3) If similar distance, the trajectory where left fore finger is closer to bucket is preferred.
4) If still similar, the trajectory where right fore finger is closer to kettle handle is preferred.
Please give response with only one list of 5 preference values, e.g., [0, 0, 1, 2, 3]. Do not provide any other text such as your comments or thoughts. The preference value number can only be 0, 1, 2, or 3.
Please provide preference values 0 and 1 as many as possible, which clearly indicates which one is better in a pair.
Please be very careful about providing equally preferable value 2. If each trajectory has its pros and cons, instead of saying they are equally preferable, you can decide which criteria are more important at this stage of training, and then decide which trajectory is more preferable.
Please avoid providing incomparable value 3! Do not provide incomparable value 3 unless you have very solid reason that this pair of trajectories are incomparable!

---

### Prompt for Hand Over

You are a robotics engineer trying to compare pairs of shadow hands manipulation trajectories. Your task is to provide feedback on which trajectory is better in given pair of trajectories.
Your feedback of the comparisons will be used as reward signal to train tossing an object (a ball in this case) from hand 0 to hand 1.
For your reference, the palm position of hand 0, the releasing hand, is [0.000, -0.290, 0.490]. And the palm position of hand 1, the catching hand, is [0.000, -0.640, 0.540].
Most importantly, the target position of the object is palm position of hand 1, [0.000, -0.640, 0.540].
Each trajectory will contain 16 timesteps of states of the shadow hand. To be specific, the state are as below:
1) "object position": the x, y, z position of the object. The data shape is (16, 3), standing for 16 steps, and x, y, z 3 dimensional position.
2) "object linear velocity": the x, y, z positional velocities (m/s) of the object. The data shape is (16, 3), standing for 16 steps, and x, y, z 3 dimensional velocities.
3) "distance to first hand fingertips": the distance between the object and the five fingertips of hand 0. The data shape is (16, 5), standing for 16 steps, and 5 fingertips.
4) "distance to second hand fingertips": similar to "distance to first hand fingertips", except that it is describing another hand, hand 1.
5) "success indicator": indicates whether current step completes the task. The length is 16, standing for 16 steps. 1 stands for True and 0 for False.
To decide which trajectory is better in a pair, here are some criteria:
1) The trajectory that succeeds is better.

2) The object (ball) should be as close to the target position as possible. The distance between "object position" and target position can measure. And the second and third digits of "object position" should matter the most.

3) The object should keep a distance from any fingertips for both hands. Being smaller than *threshold of 0.03* is highly penalized.

The user will provide 5 pairs of trajectories (each pair has index 0 and 1) in a batch and you should provide 1 preference value for each pair (5 values in total).

1) If the trajectory 0 is better, the preference value should be 0.

2) If the trajectory 1 is better, the preference value should be 1.

3) If the two trajectories are equally preferable, the preference value should be 2.

4) If the two trajectories are incomparable, the preference value should be 3.

Examples for preference:

1) If one trajectory has more success indicators, it is better.

2) If neither succeeds, the trajectory where object position is closer to target position is preferred.

3) If both succeed, the trajectory with closer distances in y and z axes between object and target is preferred.

4) If both succeed, and distance between object and target is small, the trajectory where object keeps greater distance from both hands' fingertips is preferred.

Please give response with only one list of 5 preference values, e.g., [0, 0, 1, 2, 3]. Do not provide any other text such as your comments or thoughts. The preference value number can only be 0, 1, 2, or 3.

Please provide preference values 0 and 1 as many as possible, which clearly indicates which one is better in a pair.

Please be very careful about providing equally preferable value 2. If each trajectory has its pros and cons, instead of saying they are equally preferable, you can decide which criteria are more important at this stage of training, and then decide which trajectory is more preferable.

Please avoid providing incomparable value 3! Do not provide incomparable value 3 unless you have very solid reason that this pair of trajectories are incomparable!

## Prompt for Swing Cup

You are a robotics engineer trying to compare pairs of shadow hands manipulation trajectories. Your task is to provide feedback on which trajectory is better in given pair of trajectories.

Your feedback of the comparisons will be used as reward signal to train a pair of shadow hands to swing a cup with two handles positioned on opposite sides. They are pushing the handles in a coordinated manner to achieve a 180-degree counter-clockwise rotation along the z-axis

Most importantly, the goal rotation of the cup is [ 0.0000, -0.0000, -1.5708].

Each trajectory will contain 16 timesteps of states of the shadow hand. To be specific, the state are as below:

1) "object linear velocity": the x, y, z positional velocities (m/s) of the object. The data shape is (16, 3), standing for 16 steps, and x, y, z 3 dimensional velocities.

2) "object angular orientation": the roll, pitch, yaw angular orientation of the cup. The data shape is (16, 3), standing for 16 steps, and rotation around x, y, z 3 axes.

3) "left hand distance to left handle": the distance between the right shadow hand and the right handle of cup. The length is 16, standing for 16 steps.

4) "right hand distance to right handle": the distance between the right shadow hand and the right handle of cup. The length is 16, standing for 16 steps.

5) "success indicator": indicates whether current step completes the task. The length is 16, standing for 16 steps. 1 stands for True and 0 for False.

To decide which trajectory is better in a pair, here are some criteria (importance by rank):

1) The trajectory that succeeds is better.

2) The object rotation should be as close to target rotation as possible. The "object angular orientation" can help measure.
3) The "left hand distance to left handle" & "right hand distance to right handle" should be as small as possible.
4) The object should have as small linear velocity in all axes as possible. The "object linear velocity" can measure
The user will provide 5 pairs of trajectories (each pair has index 0 and 1) in a batch and you should provide 1 preference value for each pair (5 values in total).
1) If the trajectory 0 is better, the preference value should be 0.
2) If the trajectory 1 is better, the preference value should be 1.
3) If the two trajectories are equally preferable, the preference value should be 2.
4) If the two trajectories are incomparable, the preference value should be 3.
Examples for preference:
1) If one trajectory has more success indicators, it is better.
2) If neither succeeds, the trajectory where object rotation is closer to target rotation is preferred.
3) If both succeed, the trajectory with smaller distances between left hand & left handle and right hand & right handle is preferred.
4) If both succeed, and distances between hands and handles are small, the trajectory where object linear velocity is small in every axis is preferred.
Please give response with only one list of 5 preference values, e.g., [0, 0, 1, 2, 3]. Do not provide any other text such as your comments or thoughts. The preference value number can only be 0, 1, 2, or 3.
Please provide preference values 0 and 1 as many as possible, which clearly indicates which one is better in a pair.
Please be very careful about providing equally preferable value 2. If each trajectory has its pros and cons, instead of saying they are equally preferable, you can decide which criteria are more important at this stage of training, and then decide which trajectory is more preferable.
Please avoid providing incomparable value 3! Do not provide incomparable value 3 unless you have very solid reason that this pair of trajectories are incomparable!

### A.3 Full Rewards

In this section,we provide all the explicit environment rewards for training with LAPP. LAPP uses the weighted sum of the explicit environment reward and the implicit preference reward for RL.

```
Reward for Flat Plane locomotion and bounding control and cadence control

def compute_reward(self):
    """ Compute rewards
        Compute each reward component first
        Then compute the total reward
        Return the total reward, and the recording of all reward components
    """
    env = self.env  # Do not skip this line. Afterwards, use env.{parameter_name}
    ↪  to access parameters of the environment.

    # Tracking of linear velocity commands (xy axes)
    lin_vel_error = torch.sum(torch.square(env.commands[:, :2] -
    ↪  env.base_lin_vel[:, :2]), dim=1)
    tracking_lin_vel_reward = 1.0 * torch.exp(-lin_vel_error / 0.25)

    # Tracking of angular velocity commands (yaw)
```

```
    ang_vel_error = torch.square(env.commands[:, 2] - env.base_ang_vel[:, 2])
    tracking_ang_vel_reward = 0.5 * torch.exp(-ang_vel_error / 0.25)

    # Penalize z axis base linear velocity
    lin_vel_z_reward = -2.0 * torch.square(env.base_lin_vel[:, 2])

    # Penalize xy axes base angular velocity
    ang_vel_xy_reward = -0.05 * torch.sum(torch.square(env.base_ang_vel[:, :2]),
    ↪   dim=1)

    # Penalize torques
    torques_reward = -0.0002 * torch.sum(torch.square(env.torques), dim=1)

    # Penalize dof accelerations
    dof_acc_reward = -2.5e-7 * torch.sum(torch.square((env.last_dof_vel -
    ↪   env.dof_vel) / env.dt), dim=1)

    # Reward long steps
    # Need to filter the contacts because the contact reporting of PhysX is
    ↪   unreliable on meshes
    contact = env.contact_forces[:, env.feet_indices, 2] > 1.
    contact_filt = torch.logical_or(contact, env.last_contacts)
    env.last_contacts = contact
    first_contact = (env.feet_air_time > 0.) * contact_filt
    env.feet_air_time += env.dt
    rew_airTime = torch.sum((env.feet_air_time - 0.5) * first_contact, dim=1)  #
    ↪   reward only on first contact with the ground
    rew_airTime *= torch.norm(env.commands[:, :2], dim=1) > 0.1  # no reward for
    ↪   zero command
    env.feet_air_time *= ~contact_filt
    feet_air_time_reward = 1.0 * rew_airTime

    # Penalize collisions on selected bodies
    collision_reward = -1.0 * torch.sum(1. * (torch.norm(env.contact_forces[:,
    ↪   env.penalised_contact_indices, :], dim=-1) > 0.1), dim=1)

    # Penalize changes in actions
    action_rate_reward = -0.01 * torch.sum(torch.square(env.last_actions -
    ↪   env.actions), dim=1)

    # Penalize dof positions too close to the limit
    out_of_limits = -(env.dof_pos - env.dof_pos_limits[:, 0]).clip(max=0.)  # lower
    ↪   limit
    out_of_limits += (env.dof_pos - env.dof_pos_limits[:, 1]).clip(min=0.)
    dof_pos_limits_reward = -10.0 * torch.sum(out_of_limits, dim=1)

    # # Penalize base height away from target
    # target_height_z = 0.34  # Ideal height of the robot's torso
    # base_height = env.root_states[:, 2]
    # height_reward = -0.05 * torch.square(base_height - target_height_z)  # reward
    ↪   to maintain height
```

```
    # Height reward component
    target_height_z = 0.34  # Ideal height of the robot's torso
    base_height = env.root_states[:, 2]
    height_error = torch.abs(base_height - target_height_z)
    temperature_height = 5.0  # Temperature parameter for the height reward
    height_reward = 1.0 * torch.exp(-temperature_height * height_error)  # More
    ↪   weight to maintain height

    # Combine reward components to compute the total reward in this step
    total_reward = (tracking_lin_vel_reward + tracking_ang_vel_reward +
    ↪   lin_vel_z_reward +
                    ang_vel_xy_reward + torques_reward + dof_acc_reward +
                    ↪   feet_air_time_reward +
                    collision_reward + action_rate_reward + dof_pos_limits_reward +
                    ↪   height_reward)

    # # Normalizing the total reward to avoid exploding values
    # total_reward = total_reward / (1 + torch.abs(total_reward))  # Additional
    ↪   normalization for stability

    # Debug information
    reward_components = {"tracking_lin_vel_reward": tracking_lin_vel_reward,
                         "tracking_ang_vel_reward": tracking_ang_vel_reward,
                         "lin_vel_z_reward": lin_vel_z_reward,
                         "ang_vel_xy_reward": ang_vel_xy_reward,
                         "torques_reward": torques_reward,
                         "dof_acc_reward": dof_acc_reward,
                         "feet_air_time_reward": feet_air_time_reward,
                         "collision_reward": collision_reward,
                         "action_rate_reward": action_rate_reward,
                         "dof_pos_limits_reward": dof_pos_limits_reward,
                         "height_reward": height_reward}
    return total_reward, reward_components
```

**Reward for Stairs**

```
def compute_reward(self):
    """ Compute improved rewards
        Compute each reward component first
        Then compute the total reward
        Return the total reward, and the recording of all reward components
    """
    env = self.env  # Do not skip this line. Afterwards, use env.{parameter_name}
    ↪   to access parameters of the environment.

    # Tracking of linear velocity commands (xy axes)
    lin_vel_error = torch.sum(torch.square(env.commands[:, :2] -
    ↪   env.base_lin_vel[:, :2]), dim=1)
    tracking_lin_vel_reward = 1.5 * torch.exp(-lin_vel_error / 0.20)

    # Tracking of angular velocity commands (yaw)
```

```
ang_vel_error = torch.square(env.commands[:, 2] - env.base_ang_vel[:, 2])
tracking_ang_vel_reward = 0.5 * torch.exp(-ang_vel_error / 0.1)

# # Penalize z axis base linear velocity
lin_vel_z_reward = -0.00001 * torch.square(env.base_lin_vel[:, 2])

# Penalize xy axes base angular velocity
ang_vel_xy_reward = -0.1 * torch.sum(torch.square(env.base_ang_vel[:, :2]),
↪  dim=1)

# Penalize torques
torques_reward = -0.0005 * torch.sum(torch.square(env.torques), dim=1)

# Penalize dof accelerations
dof_acc_reward = -1.0e-7 * torch.sum(torch.square((env.last_dof_vel -
↪  env.dof_vel) / env.dt), dim=1)

# Reward air time
contact = env.contact_forces[:, env.feet_indices, 2] > 1.
contact_filt = torch.logical_or(contact, env.last_contacts)
env.last_contacts = contact
first_contact = (env.feet_air_time > 0.) * contact_filt
env.feet_air_time += env.dt
rew_airTime = torch.sum((env.feet_air_time - 0.5) * first_contact, dim=1)
rew_airTime *= torch.norm(env.commands[:, :2], dim=1) > 0.1
env.feet_air_time *= ~contact_filt
feet_air_time_reward = 0.8 * rew_airTime

# Penalize collisions
collision_reward = -5.0 * torch.sum(1. * (torch.norm(env.contact_forces[:,
↪  env.penalised_contact_indices, :], dim=-1) > 0.1), dim=1)

# Penalize changes in actions
action_rate_reward = -0.008 * torch.sum(torch.square(env.last_actions -
↪  env.actions), dim=1)

# Penalize dofs close to limits
out_of_limits = -(env.dof_pos - env.dof_pos_limits[:, 0]).clip(max=0.)
out_of_limits += (env.dof_pos - env.dof_pos_limits[:, 1]).clip(min=0.)
dof_pos_limits_reward = -7.0 * torch.sum(out_of_limits, dim=1)

# Penalize base height away from target
target_height_z = 0.34
base_height = env.root_states[:, 2]
# get the ground height of the terrain
ground_x = env.root_states[:, 0]
ground_y = env.root_states[:, 1]
ground_z = env._get_stairs_terrain_heights(ground_x, ground_y)
# calculate the base-to-ground height
base2ground_height = base_height - ground_z
height_reward = -0.000002 * torch.square(base2ground_height - target_height_z)
```

```
    # stumbling penalty
    stumble = (torch.norm(env.contact_forces[:, env.feet_indices, :2], dim=2) > 5.)
    ↪  * (torch.abs(env.contact_forces[:, env.feet_indices, 2]) < 1.)
    stumble_reward = -2.0 * torch.sum(stumble, dim=1)

    # Combine reward components to compute the total reward in this step
    total_reward = (tracking_lin_vel_reward + tracking_ang_vel_reward +
    ↪  lin_vel_z_reward +
                    ang_vel_xy_reward + torques_reward + dof_acc_reward +
                    ↪  feet_air_time_reward +
                    collision_reward + action_rate_reward + dof_pos_limits_reward +
                    ↪  height_reward + stumble_reward)

    # Debug information
    reward_components = {"tracking_lin_vel_reward": tracking_lin_vel_reward,
                         "tracking_ang_vel_reward": tracking_ang_vel_reward,
                         "lin_vel_z_reward": lin_vel_z_reward,
                         "ang_vel_xy_reward": ang_vel_xy_reward,
                         "torques_reward": torques_reward,
                         "dof_acc_reward": dof_acc_reward,
                         "feet_air_time_reward": feet_air_time_reward,
                         "collision_reward": collision_reward,
                         "action_rate_reward": action_rate_reward,
                         "dof_pos_limits_reward": dof_pos_limits_reward,
                         "height_reward": height_reward,
                         "stumble_reward": stumble_reward}

    return total_reward, reward_components
```

**Reward for Obstacles**

```
def compute_reward(self):
    """ Compute rewards for wave terrain """
    env = self.env

    # 1. Linear velocity tracking along x-axis
    lin_vel_error = torch.sum(torch.square(env.commands[:, :2] -
    ↪  env.base_lin_vel[:, :2]), dim=1)
    tracking_lin_vel_reward = 2.3 * torch.exp(-lin_vel_error / 0.20)

    # 2. Tracking of angular velocity commands (yaw)
    ang_vel_error = torch.square(env.commands[:, 2] - env.base_ang_vel[:, 2])
    tracking_ang_vel_reward = 0.8 * torch.exp(-ang_vel_error / 0.1)

    # 3. Penalize z-axis velocity
    # lin_vel_z_reward = -0.001 * torch.square(env.base_lin_vel[:, 2])
    ang_vel_x_reward = -0.002 * torch.square(env.base_ang_vel[:, 0])

    # 4. Base height tracking (adjusted for wave terrain)
    target_height_z = 0.34
```

```
        base_height = env.root_states[:, 2]
        ground_x = env.root_states[:, 0]
        ground_y = env.root_states[:, 1]
        ground_z = env._get_terrain_heights(ground_x, ground_y)
        base2ground_height = base_height - ground_z
        height_reward = -1.5 * torch.square(base2ground_height - target_height_z)

        # 5. Penalize torques
        torques_reward = -0.00001 * torch.sum(torch.square(env.torques), dim=1)

        # 6. Penalize changes in actions
        action_rate_reward = -0.0055 * torch.sum(torch.square(env.last_actions -
        ↪  env.actions), dim=1)

        # 7. Encourage smoother joint motions (penalize excessive joint accelerations)
        dof_acc_penalty = -1e-8 * torch.sum(torch.square((env.dof_vel -
        ↪  env.last_dof_vel) / env.dt), dim=1)

        # 8. Air time reward for dynamic gaits
        contact = env.contact_forces[:, env.feet_indices, 2] > 1.0
        contact_filt = torch.logical_or(contact, env.last_contacts)
        env.last_contacts = contact
        first_contact = (env.feet_air_time > 0.0) * contact_filt
        env.feet_air_time += env.dt
        rew_airTime = torch.sum((env.feet_air_time - 0.4) * first_contact, dim=1)
        rew_airTime *= torch.norm(env.commands[:, :2], dim=1) > 0.1
        env.feet_air_time *= ~contact_filt
        air_time_reward = 0.5 * rew_airTime

        # 9. Collision penalty (avoid collisions with terrain or robot parts)
        collision_penalty = -1. * torch.sum(
            1.0 * (torch.norm(env.contact_forces[:, env.penalised_contact_indices, :],
            ↪  dim=-1) > 0.13),
            dim=1
        )

        # 10. Gait pattern reward (encourage trot gait using phase alignment)
        diag_sync = (contact[:, 0] == contact[:, 3]) & (contact[:, 1] == contact[:, 2])
        gait_pattern_reward = 0.0001 * torch.sum(diag_sync.float())

        # 11. Penalize use only two feet
        stumble = (torch.norm(env.contact_forces[:, env.feet_indices, :2], dim=2) > 5.)
        ↪  * (torch.abs(env.contact_forces[:, env.feet_indices, 2]) < 1.)
        stumble_penalty = -20.0 * torch.sum(stumble, dim=1)

        # Combine all components into the total reward
        total_reward = (
            tracking_lin_vel_reward +
            tracking_ang_vel_reward +
            ang_vel_x_reward +
            height_reward +
            torques_reward +
```

```
        action_rate_reward +
        dof_acc_penalty +
        air_time_reward +
        collision_penalty +
        gait_pattern_reward +
        stumble_penalty
    )

    # Debug information for reward components
    reward_components = {
        "tracking_lin_vel_reward": tracking_lin_vel_reward,
        "tracking_ang_vel_reward": tracking_ang_vel_reward,
        "ang_vel_x_reward": ang_vel_x_reward,
        "height_reward": height_reward,
        "torques_reward": torques_reward,
        "action_rate_reward": action_rate_reward,
        "dof_acc_penalty": dof_acc_penalty,
        "air_time_reward": air_time_reward,
        "collision_penalty": collision_penalty,
        "gait_pattern_reward": gait_pattern_reward,
        "stumble_penalty": stumble_penalty
    }

    return total_reward, reward_components
```

**Reward for Wave**

```
def compute_reward(self):
    """ Compute rewards for wave terrain """
    env = self.env

    # 1. Tracking of linear velocity commands (xy axes)
    lin_vel_error = torch.sum(torch.square(env.commands[:, :2] -
    ↪  env.base_lin_vel[:, :2]), dim=1)
    tracking_lin_vel_reward = 2.3 * torch.exp(-lin_vel_error / 0.20)

    # 2. Tracking of angular velocity commands (yaw)
    ang_vel_error = torch.square(env.commands[:, 2] - env.base_ang_vel[:, 2])
    tracking_ang_vel_reward = 0.8 * torch.exp(-ang_vel_error / 0.1)

    # 3. Base height tracking (adjusted for wave terrain)
    target_height_z = 0.34
    base_height = env.root_states[:, 2]
    ground_x = env.root_states[:, 0]
    ground_y = env.root_states[:, 1]
    ground_z = env._get_terrain_heights(ground_x, ground_y)
    base2ground_height = base_height - ground_z
    height_reward = -1.5 * torch.square(base2ground_height - target_height_z)

    # 4. Penalize torques
    torques_reward = -0.00001 * torch.sum(torch.square(env.torques), dim=1)
```

```python
        # 5. Penalize changes in actions
        action_rate_reward = -0.0045 * torch.sum(torch.square(env.last_actions -
        ↪  env.actions), dim=1)

        # 6. Encourage smoother joint motions (penalize excessive joint accelerations)
        dof_acc_penalty = -1e-8 * torch.sum(torch.square((env.dof_vel -
        ↪  env.last_dof_vel) / env.dt), dim=1)

        # 7. Air time reward for dynamic gaits
        contact = env.contact_forces[:, env.feet_indices, 2] > 1.0
        contact_filt = torch.logical_or(contact, env.last_contacts)
        env.last_contacts = contact
        first_contact = (env.feet_air_time > 0.0) * contact_filt
        env.feet_air_time += env.dt
        rew_airTime = torch.sum((env.feet_air_time - 0.4) * first_contact, dim=1)
        rew_airTime *= torch.norm(env.commands[:, :2], dim=1) > 0.1
        env.feet_air_time *= ~contact_filt
        air_time_reward = 0.5 * rew_airTime

        # 8. Collision penalty (avoid collisions with terrain or robot parts)
        collision_penalty = -1. * torch.sum(
            1.0 * (torch.norm(env.contact_forces[:, env.penalised_contact_indices, :],
            ↪  dim=-1) > 0.13),
            dim=1
        )

        # 9. Gait pattern reward (encourage trot gait using phase alignment)
        diag_sync = (contact[:, 0] == contact[:, 3]) & (contact[:, 1] == contact[:, 2])
        gait_pattern_reward = 0.0001 * torch.sum(diag_sync.float())

        # 10. Penalize use only two feet
        lack_of_foot_usage = (~contact).float().sum(dim=1)
        lack_of_foot_usage_penalty = -0.01 * lack_of_foot_usage

        # Combine all components into the total reward
        total_reward = (
            tracking_lin_vel_reward +
            tracking_ang_vel_reward +
            height_reward +
            torques_reward +
            action_rate_reward +
            dof_acc_penalty +
            air_time_reward +
            collision_penalty +
            gait_pattern_reward +
            lack_of_foot_usage_penalty
        )

        # Debug information for reward components
        reward_components = {
            "tracking_lin_vel_reward": tracking_lin_vel_reward,
```

```
        "tracking_ang_vel_reward": tracking_ang_vel_reward,
        "height_reward": height_reward,
        "torques_reward": torques_reward,
        "action_rate_reward": action_rate_reward,
        "dof_acc_penalty": dof_acc_penalty,
        "air_time_reward": air_time_reward,
        "collision_penalty": collision_penalty,
        "gait_pattern_reward": gait_pattern_reward,
        "lack_of_foot_usage_penalty": lack_of_foot_usage_penalty
    }

    return total_reward, reward_components
```

Reward for Slope

```
    def compute_reward(self):
        """ Compute improved rewards
            Compute each reward component first
            Then compute the total reward
            Return the total reward, and the recording of all reward components
        """
        env = self.env  # Do not skip this line. Afterwards, use
        ↪   env.{parameter_name} to access parameters of the environment.

        # Tracking of linear velocity commands (xy axes)
        lin_vel_error = torch.sum(torch.square(env.commands[:, :2] -
        ↪   env.base_lin_vel[:, :2]), dim=1)
        tracking_lin_vel_reward = 3.0 * torch.exp(-lin_vel_error / 0.10)

        # Tracking of angular velocity commands (yaw)
        ang_vel_error = torch.square(env.commands[:, 2] - env.base_ang_vel[:, 2])
        tracking_ang_vel_reward = 1.0 * torch.exp(-ang_vel_error / 0.05)

        # # Penalize z axis base linear velocity
        lin_vel_z_reward = -0.00001 * torch.square(env.base_lin_vel[:, 2])

        # Penalize xy axes base angular velocity
        ang_vel_xy_reward = -0.1 * torch.sum(torch.square(env.base_ang_vel[:, :2]),
        ↪   dim=1)

        # Penalize torques
        torques_reward = -0.0001 * torch.sum(torch.square(env.torques), dim=1)

        # Penalize dof accelerations
        dof_acc_reward = -5.0e-8 * torch.sum(torch.square((env.last_dof_vel -
        ↪   env.dof_vel) / env.dt), dim=1)

        # Reward air time
        contact = env.contact_forces[:, env.feet_indices, 2] > 1.
        contact_filt = torch.logical_or(contact, env.last_contacts)
        env.last_contacts = contact
        first_contact = (env.feet_air_time > 0.) * contact_filt
```

```
        env.feet_air_time += env.dt
        rew_airTime = torch.sum((env.feet_air_time - 0.5) * first_contact, dim=1)
        rew_airTime *= torch.norm(env.commands[:, :2], dim=1) > 0.1
        env.feet_air_time *= ~contact_filt
        feet_air_time_reward = 0.6 * rew_airTime

        # Penalize collisions
        collision_reward = -15.0 * torch.sum(1. * (torch.norm(env.contact_forces[:,
        ↪  env.penalised_contact_indices, :], dim=-1) > 0.1), dim=1)

        # Penalize changes in actions
        action_rate_reward = -0.015 * torch.sum(torch.square(env.last_actions -
        ↪  env.actions), dim=1)

        # Penalize dofs close to limits
        out_of_limits = -(env.dof_pos - env.dof_pos_limits[:, 0]).clip(max=0.)
        out_of_limits += (env.dof_pos - env.dof_pos_limits[:, 1]).clip(min=0.)
        dof_pos_limits_reward = -6.0 * torch.sum(out_of_limits, dim=1)

        # Penalize base height away from target
        target_height_z = 0.34
        base_height = env.root_states[:, 2]
        # get the ground height of the terrain
        ground_x = env.root_states[:, 0]
        ground_y = env.root_states[:, 1]
        ground_z = env._get_terrain_heights(ground_x, ground_y)
        # calculate the base-to-ground height
        base2ground_height = base_height - ground_z
        height_reward = -0.000001 * torch.square(base2ground_height -
        ↪  target_height_z)

        # stumbling penalty
        # stumble = (torch.norm(env.contact_forces[:, env.feet_indices, :2], dim=2)
        ↪  > 5.) * (torch.abs(env.contact_forces[:, env.feet_indices, 2]) < 1.)
        # stumble_reward = -4.0 * torch.sum(stumble, dim=1)

        # Combine reward components to compute the total reward in this step
        total_reward = (tracking_lin_vel_reward + tracking_ang_vel_reward +
        ↪  lin_vel_z_reward +
                    ang_vel_xy_reward + torques_reward + dof_acc_reward +
                    ↪  feet_air_time_reward +
                    collision_reward + action_rate_reward +
                    ↪  dof_pos_limits_reward + height_reward)
                    # + stumble_reward)

        # Debug information
        reward_components = {"tracking_lin_vel_reward": tracking_lin_vel_reward,
                        "tracking_ang_vel_reward": tracking_ang_vel_reward,
                        "lin_vel_z_reward": lin_vel_z_reward,
                        "ang_vel_xy_reward": ang_vel_xy_reward,
                        "torques_reward": torques_reward,
                        "dof_acc_reward": dof_acc_reward,
```

```
                                    "feet_air_time_reward": feet_air_time_reward,
                                    "collision_reward": collision_reward,
                                    "action_rate_reward": action_rate_reward,
                                    "dof_pos_limits_reward": dof_pos_limits_reward,
                                    "height_reward": height_reward,}

        return total_reward, reward_components
```

Reward for Jump high

```
def compute_reward(self):
    """ Compute rewards for the backflip task """
    env = self.env  # Do not skip this line. Afterwards, use env.{parameter_name}
    →  to access parameters of the environment.

    # Penalize angular velocity around x-axis (roll) and z-axis (yaw)
    roll_rate = env.root_states[:, 10]  # Angular velocity around x-axis (roll)
    yaw_rate = env.root_states[:, 12]  # Angular velocity around z-axis (yaw)
    roll_rate_reward = -0.05 * torch.square(roll_rate)  # Penalize deviation in
    →  roll, org -5.0, 0.5
    yaw_rate_reward = -0.2 * torch.square(yaw_rate)  # Penalize deviation in yaw,
    →  org -5.0, -2.0

    # Penalize roll and yaw around x-axis (roll) and z-axis (yaw)
    roll = env.rpy[:, 0]
    pitch = env.rpy[:, 1]
    yaw = env.rpy[:, 2]
    roll_reward = -0.2 * torch.square(roll)  # Penalize deviation in roll
    pitch_reward = -0.2 * torch.square(pitch)  # Penalize deviation in pitch
    yaw_reward = -1.0 * torch.square(yaw)  # Penalize deviation in yaw

    # Encourage z axis base linear velocity
    lin_vel_z_reward = 0.4 * torch.square(env.root_states[:, 9])

    # Penalize x axis base linear velocity forward
    lin_vel_x_reward  = -0.4 * torch.square(torch.relu(env.root_states[:, 7]))

    # Penalize torques
    torques_reward = -0.0005 * torch.sum(torch.square(env.torques), dim=1)

    # Penalize dof accelerations
    dof_acc_reward = -1.0e-7 * torch.sum(torch.square((env.last_dof_vel -
    →  env.dof_vel) / env.dt), dim=1)

    # # Penalize dofs close to limits
    # out_of_limits = -(env.dof_pos - env.dof_pos_limits[:, 0]).clip(max=0.)
    # out_of_limits += (env.dof_pos - env.dof_pos_limits[:, 1]).clip(min=0.)
    # dof_pos_limits_reward = -7.0 * torch.sum(out_of_limits, dim=1)

    # Penalize base torso hitting the ground
```

```
    base_hit_ground = torch.any(torch.norm(env.contact_forces[:,
    ↪  env.termination_contact_indices, :], dim=-1) > 0.1, dim=1)  # >1.0
    ↪  originally
    base_hit_ground_reward = -20.0 * base_hit_ground  # 10

    # Penalize non flat base orientation
    gravity_reward = -0.01 * torch.sum(torch.square(env.projected_gravity[:, :2]),
    ↪  dim=1)

    # Reward for highest z position reached in an episode
    height_history = env.height_history_buf
    height_history_reward = 1.0 * height_history  # 0.5

    # Combine reward components to compute the total reward in this step
    total_reward = (roll_rate_reward + yaw_rate_reward + roll_reward + pitch_reward
    ↪  + yaw_reward + lin_vel_z_reward + lin_vel_x_reward +
                    torques_reward + dof_acc_reward+ base_hit_ground_reward +
                    ↪  gravity_reward + height_history_reward)

    # Debug information
    reward_components = {
        "roll_rate_reward": roll_rate_reward,
        "yaw_rate_reward": yaw_rate_reward,
        "roll_reward": roll_reward,
        "pitch_reward": pitch_reward,
        "yaw_reward": yaw_reward,
        "lin_vel_z_reward": lin_vel_z_reward,
        "lin_vel_x_reward": lin_vel_x_reward,
        "torques_reward": torques_reward,
        "dof_acc_reward": dof_acc_reward,
        "base_hit_ground_reward": base_hit_ground_reward,
        "gravity_reward": gravity_reward,
        "height_history_reward": height_history_reward
    }
    return total_reward, reward_components
```

**Reward for Backflip**

```
def compute_reward(self):
    """ Compute rewards for the backflip task """
    env = self.env  # Do not skip this line. Afterwards, use env.{parameter_name}
    ↪  to access parameters of the environment.

    # Penalize angular velocity around x-axis (roll) and z-axis (yaw)
    roll_rate = env.root_states[:, 10]  # Angular velocity around x-axis (roll)
    yaw_rate = env.root_states[:, 12]  # Angular velocity around z-axis (yaw)
    roll_rate_reward = -0.05 * torch.square(roll_rate)  # Penalize deviation in
    ↪  roll, org -5.0, 0.5
    yaw_rate_reward = -0.2 * torch.square(yaw_rate)  # Penalize deviation in yaw,
    ↪  org -5.0, -2.0
```

```
# Penalize roll and yaw around x-axis (roll) and z-axis (yaw)
roll = env.rpy[:, 0]
yaw = env.rpy[:, 2]
roll_reward = -0.2 * torch.square(roll)  # Penalize deviation in roll
yaw_reward = -1.0 * torch.square(yaw)  # Penalize deviation in yaw

# Reward for angular velocity around y-axis (pitch), encourage backflip
↪ rotation
pitch_rate = env.base_ang_vel[:, 1]  # Angular velocity around y-axis (pitch)
# Clamp the pitch rate to a minimum of -7 rad/s (maximum negative rotation
↪ speed)
clamped_pitch_rate = torch.clamp(pitch_rate, max=5.0, min=-7.0)
# Compute the backflip reward
pitch_rate_reward = torch.where(
    clamped_pitch_rate < 0,
    1.0 * (-clamped_pitch_rate),  # Positive reward for negative pitch rate
    -0.1 * clamped_pitch_rate  # Negative penalty for positive pitch rate
)

# pitch_rate_reward = 1.0 * (-clamped_pitch_rate)  # Reward negative pitch
↪ rates

# Reward for pitch frontwards for the back flip
pitch = env.rpy[:, 1]
last_pitch = env.last_rpy[:, 1]
delta_pitch = (pitch - last_pitch + torch.pi) % (2 * torch.pi) - torch.pi
delta_pitch_reward = torch.where(delta_pitch > 0, delta_pitch,
↪ torch.zeros_like(delta_pitch))
delta_pitch_reward = 20.0 * delta_pitch_reward

# Encourage z axis base linear velocity
lin_vel_z_reward = 0.2 * torch.square(env.root_states[:, 9])

# Penalize torques
torques_reward = -0.0005 * torch.sum(torch.square(env.torques), dim=1)

# Penalize dof accelerations
dof_acc_reward = -1.0e-7 * torch.sum(torch.square((env.last_dof_vel -
↪ env.dof_vel) / env.dt), dim=1)

# # Penalize dofs close to limits
# out_of_limits = -(env.dof_pos - env.dof_pos_limits[:, 0]).clip(max=0.)
# out_of_limits += (env.dof_pos - env.dof_pos_limits[:, 1]).clip(min=0.)
# dof_pos_limits_reward = -7.0 * torch.sum(out_of_limits, dim=1)

# Penalize base torso hitting the ground
base_hit_ground = torch.any(torch.norm(env.contact_forces[:,
↪ env.termination_contact_indices, :], dim=-1) > 0.1, dim=1)  # >1.0
↪ originally
base_hit_ground_reward = -20.0 * base_hit_ground  # 10

# Penalize projected gravity along y axis
```

```
    gravity_reward = -0.01 * torch.square(env.projected_gravity[:, 1])

    # Encourage backwards rotation of the projected gravity
    delta_gravity_x = env.projected_gravity[:, 0] - env.last_projected_gravity[:,
    ↪  0]
    condition = env.projected_gravity[:, 2] < 0  # Corrected to use the z-axis as
    ↪  per your description
    desired_direction = torch.where(
        condition,
        -1.0,  # Encourage decrease in delta_gravity_x
        1.0  # Encourage increase in delta_gravity_x
    )

    # Compute the alignment between desired and actual change
    alignment = desired_direction * delta_gravity_x

    # Compute the reward with different scaling factors
    rotate_gravity_reward = torch.where(
        alignment > 0,
        20.0 * torch.abs(delta_gravity_x),  # Positive reward
        -2.0 * torch.abs(delta_gravity_x)  # Negative penalty
    )

    # Reward for highest z position reached in an episode
    height_history = env.height_history_buf
    height_history_reward = 0.8 * height_history  # 0.5

    # Combine reward components to compute the total reward in this step
    total_reward = (roll_rate_reward + yaw_rate_reward + roll_reward + yaw_reward +
    ↪  pitch_rate_reward +
                    delta_pitch_reward + lin_vel_z_reward + torques_reward +
                    ↪  dof_acc_reward+
                    base_hit_ground_reward + gravity_reward + rotate_gravity_reward
                    ↪  + height_history_reward)

    # Debug information
    reward_components = {
        "roll_rate_reward": roll_rate_reward,
        "yaw_rate_reward": yaw_rate_reward,
        "roll_reward": roll_reward,
        "yaw_reward": yaw_reward,
        "pitch_rate_reward": pitch_rate_reward,
        "delta_pitch_reward": delta_pitch_reward,
        "lin_vel_z_reward": lin_vel_z_reward,
        "torques_reward": torques_reward,
        "dof_acc_reward": dof_acc_reward,
        "base_hit_ground_reward": base_hit_ground_reward,
        "gravity_reward": gravity_reward,
        "rotate_gravity_reward": rotate_gravity_reward,
        "height_history_reward": height_history_reward
    }
```

```
    return total_reward, reward_components
```

### Reward for Kettle

```
def kettle_compute_reward(kettle_spout_pos: torch.Tensor,
                          bucket_handle_pos: torch.Tensor,
                          left_hand_pos: torch.Tensor,
                          right_hand_pos: torch.Tensor,
                          left_hand_ff_pos: torch.Tensor,
                          right_hand_ff_pos: torch.Tensor
) -> Tuple[torch.Tensor, Dict[str, torch.Tensor]]:

    # Redefine proximity reward for less dominance
    kettle_to_bucket_distance = torch.norm(kettle_spout_pos - bucket_handle_pos,
    ↪   dim=1)
    temp_proximity = 1.0
    proximity_reward = torch.exp(-kettle_to_bucket_distance / temp_proximity)

    # Rescale grip rewards and refine detection logic
    left_grip_distance = torch.norm(left_hand_ff_pos - kettle_spout_pos, dim=1)
    right_grip_distance = torch.norm(right_hand_ff_pos - bucket_handle_pos, dim=1)

    temp_grip = 0.5
    left_grip_reward = torch.exp(-left_grip_distance / temp_grip)
    right_grip_reward = torch.exp(-right_grip_distance / temp_grip)

    # Improved task-specific reward detection logic
    task_success_indicator = torch.tensor([0], device=kettle_spout_pos.device)  #
    ↪   Replace with task success condition
    task_completion_threshold = 0.1  # Define a threshold to reflect task
    ↪   completion
    task_complete = kettle_to_bucket_distance < task_completion_threshold
    transformed_task_score = task_complete.float()

    # Total reward balancing individual components
    total_reward = (
        0.2 * proximity_reward +
        0.3 * left_grip_reward +
        0.3 * right_grip_reward +
        1.0 * transformed_task_score
    )

    # Reward components provided for diagnosis
    reward_components = {
        "kettle_spout_proximity": proximity_reward,
        "kettle_handle_grip": left_grip_reward,
        "bucket_handle_grip": right_grip_reward,
        "task_success": transformed_task_score
    }
```

```
      return total_reward, reward_components
```

Reward for Hand Over

```
def hand_over_compute_reward(object_pos: torch.Tensor,
                             object_rot: torch.Tensor,
                             object_linvel: torch.Tensor,
                             goal_pos: torch.Tensor,
                             goal_rot: torch.Tensor,
                             fingertip_pos: torch.Tensor,
                             fingertip_another_pos: torch.Tensor
) -> Tuple[torch.Tensor, Dict[str, torch.Tensor]]:
    # Constants
    distance_threshold: float = 0.03
    rotation_threshold: float = 0.1
    catch_reward_weight: float = 30.0
    toss_reward_weight: float = 10.0
    toss_temperature: float = 1.5
    catch_temperature: float = 0.5
    penalty_weight: float = 10.0

    # Compute distance and rotation differences between object and goal
    object_goal_distance = torch.norm(object_pos - goal_pos, dim=-1)
    object_goal_rotation_diff = torch.norm(object_rot - goal_rot, dim=-1)

    # Reward for object being close to the goal position and rotation
    toss_reward = torch.exp(-toss_reward_weight * (object_goal_distance /
    ↪  toss_temperature))

    # Calculate the catch reward based on the difference between the object's
    ↪  linear velocity and the goal position
    catch_reward = torch.norm(goal_pos - object_linvel, dim=-1)
    catch_reward = torch.exp(-catch_reward_weight * (catch_reward /
    ↪  catch_temperature))

    # Penalty for direct rolling or touching the target instead of tossing and
    ↪  catching
    penalty = torch.zeros_like(object_goal_distance)
    for i in range(fingertip_pos.shape[1]):
        dist_to_fingertip = torch.norm(object_pos - fingertip_pos[:, i, :], dim=-1)
        dist_to_fingertip_another = torch.norm(object_pos -
        ↪  fingertip_another_pos[:, i, :], dim=-1)
        penalty = torch.where(
            (dist_to_fingertip < distance_threshold) | (dist_to_fingertip_another <
            ↪  distance_threshold),
            penalty + 1.0, penalty
        )

    penalty_ratio = torch.sigmoid(penalty * penalty_weight) * 0.5
    penalty = catch_reward * penalty_ratio  # Adapt the penalty to the
    ↪  catch_reward's dynamic range
```

```
    # Total reward
    total_reward = toss_reward + catch_reward - penalty
    reward_terms = {"toss_reward": toss_reward, "catch_reward": catch_reward,
    ↪  "penalty": penalty}

    return total_reward, reward_terms
```

**Reward for Swing Cup**

```
def swing_cup_compute_reward(object_pos: torch.Tensor,
                             object_rot: torch.Tensor,
                             object_linvel: torch.Tensor,
                             cup_right_handle_pos: torch.Tensor,
                             cup_left_handle_pos: torch.Tensor,
                             left_hand_pos: torch.Tensor,
                             right_hand_pos: torch.Tensor,
                             goal_pos: torch.Tensor,
                             goal_rot: torch.Tensor
    ) -> Tuple[torch.Tensor, Dict[str, torch.Tensor]]:
    object_goal_distance = torch.norm(object_pos - goal_pos, dim=-1)
    distance_reward_temperature = 0.1
    object_goal_distance_reward_weight = 0.1
    object_goal_distance_reward = -torch.exp(-distance_reward_temperature *
    ↪  object_goal_distance) * object_goal_distance_reward_weight

    right_cup_handle_dist = torch.norm(cup_right_handle_pos - right_hand_pos,
    ↪  dim=-1)
    left_cup_handle_dist = torch.norm(cup_left_handle_pos - left_hand_pos, dim=-1)

    cup_orientation_diff = 1 - torch.sum(torch.mul(object_rot, goal_rot), dim=-1)
    ↪  ** 2
    cup_orientation_reward_weight = 1.
    cup_orientation_reward = -(cup_orientation_diff *
    ↪  cup_orientation_reward_weight)

    grasp_temperature_1 = 0.25
    grasp_temperature_2 = 0.25
    right_grasp_reward = torch.exp(-grasp_temperature_1 * right_cup_handle_dist)
    left_grasp_reward = torch.exp(-grasp_temperature_2 * left_cup_handle_dist)
    grasp_reward = (right_grasp_reward + left_grasp_reward - 1.0)

    cup_linvel_norm = torch.norm(object_linvel, dim=-1)
    cup_linvel_penalty_weight = 0.2
    cup_linvel_penalty = -(cup_linvel_norm * cup_linvel_penalty_weight)

    touch_reward_temperature = 0.25
    touch_reward_weight = 0.125
    touch_reward = (torch.exp(-touch_reward_temperature * right_cup_handle_dist) +
    ↪  torch.exp(-touch_reward_temperature * left_cup_handle_dist) - 1.0) *
    ↪  touch_reward_weight
```

```
    total_reward = grasp_reward + object_goal_distance_reward +
    ↪  cup_orientation_reward + cup_linvel_penalty + touch_reward

    reward_dict = {
        "grasp_reward": grasp_reward,
        "object_goal_distance_reward": object_goal_distance_reward,
        "cup_orientation_reward": cup_orientation_reward,
        "cup_linvel_penalty": cup_linvel_penalty,
        "touch_reward": touch_reward
    }

    return total_reward, reward_dict
```

## A.4 Preference Reward vs. Environment Reward

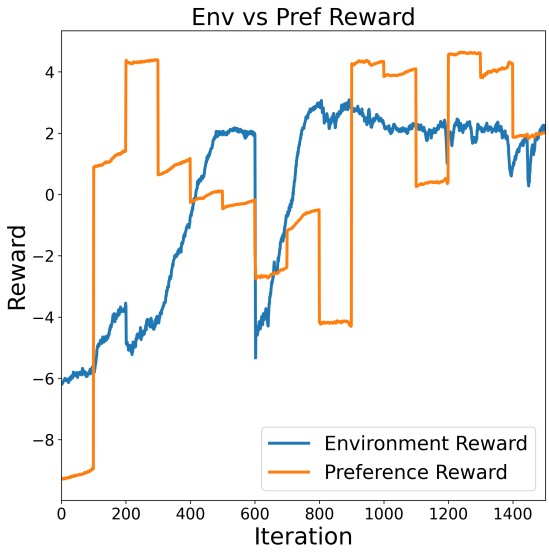

Figure 15: **Preference Reward vs. Environment Reward.** Comparison of the preference reward and the environment reward of the Backflip task throughout the training process.

To understand the effect of the preference reward and the environment reward in the training process, we take the Backflip task as an example and visualize them in Figure 15. The overall trends of both rewards are similar, growing from lower values and converging to higher values. The environment reward is relatively more continuous, showing its effectiveness of stabilizing the training and ensuring the basic performance. The preference reward has periodic jumps, indicating the variation of the priority of the preference criteria throughout the training process. This flexible and dynamic reward signal enables the success in very challenging tasks. In summary, both types of rewards are essential for stable and effective learning.

## A.5 Reward Predictor Training Cost

In order to understand the extra cost of training the reward predictor in LAPP,we take the Backflip task as an example and analyze the time consumption of training the robot policy, reward predictor, and generating the preference labels with the LLM. The training process runs on an NVIDIA RTX A6000 GPU. As shown

| Cost | Policy Training | Predictor Training | LLM Labeling |
|---|---|---|---|
| Time (second) | 11237 | 1995 | 2333 |
| Percentage | 72.2% | 12.8% | 15.0% |

Table 1: Cost of training the reward predictor in the Backflip task.

in Table 1, the policy training takes 72.2% of the total training time, which is the majority. The reward predictor training takes 12.8% of the time and the LLM Labeling process takes 15.0%. This result shows that extra cost of training the reward predictor in LAPP is the minority of the entire training cost. With the future advancement of the LLM responding efficiency, we believe the LLM Labeling time will be further shorten, making the extra cost more neglectable.

