# OpenReview forum: "LAPP: Large Language Model Feedback for Preference-Driven Reinforcement Learning"
_TMLR — Accepted by TMLR_

### Review · Reviewer_68np · 2025-05-24

**Summary Of Contributions:**

This work uses LLM to generate rewards and then uses Transformer for reward generator. The experiment prove the efffectiness of this work.

**Audience:**

Yes

**Claims And Evidence:**

Yes

**Requested Changes:**

N/A

**Strengths And Weaknesses:**

Strength:
* The presentation is clear, and the paper is well-written.
* The experiment supports the effectiveness of the method.

Weakness:
* As the LLM could judge the reward, why additionally train a reward Transformer to generate reward?
* For Figure 5,the LAPP convergence speed may be lower than PPO at the early stage. Any explanation?
* The detailed prompt example should be provided.

---

> ### Author Response · Authors · 2025-05-30
> **Response to Reviewer 68np**
>
> We thank the reviewer for taking the time and effort to review our manuscript. We appreciate your positive feedback on our presentation and experiments. We would like to address all your concerns and questions below with point responses:
>
> - **“As the LLM could judge the reward, why additionally train a reward Transformer to generate reward?”**
>
> There are two reasons for this design choice.
>
> Firstly, it is relatively easier for LLM to compare a pair of trajectories and decide which one is better, than directly produce the preference _reward value_ of a state. Therefore, the reward Transformer predicts the reward of each state, and these rewards of states in a trajectory are summed up for a comparison and calculate the cross-entropy loss. The introduction of a reward predictor follows the previous methods of RLHF, where humans are better at comparing a pair of trajectory videos than providing an absolute preference score of a state. Training the additional reward model enables us to obtain the final reward value from the preference feedback from the LLM model. This final reward value is then used as typical supervision for reinforcement learning.
>
> Secondly, using LLM to directly judge the reward is inefficient for online policy training. It requires way too many prompts to LLM and will lead to very expensive utility fee. Therefore, it is not feasible and scalable to query LLMs for all samples in RL training. Instead, the reward model can learn to approximate the LLM responses to provide signals for large number of samples required for RL training. Querying this reward model for RL training is both time and cost efficient. Specifically, in RL training, we have 4096 parallel environments and 200 steps for each rollout. If we directly use LLM to judge the reward for each step of each environment in the training process, the utility cost of LLM will be extremely high. Therefore, we use supervised learning to train a reward transformer to predict the preference reward from LLM and then use the predicted preference reward as a free alternative to the real preference reward.
>
> - **“For Figure 5, the LAPP convergence speed may be lower than PPO at the early stage. Any explanation?”**
>
> We would like to clarify that, out of the 8 experiments, LAPP outperforms all other baselines in convergence speed in 5 experiments (i.e., Plane, Stairs, Obstacles, Kettle, Hand Over). In the other 3 experiments, LAPP performs on par, even at the early stage, in Slope and Wave.
>
> We acknowledge that LAPP convergence speed is lower than PPO only in Swing Cup with better final performance in both convergence speed and reward. We believe that this is because robots are not well-trained at the early stage and may have very random behaviors. Therefore, the preference feedback generated by the LLM can be very noisy. As a result, the noisy preference reward at the early stage of the training can sometimes be harmful to the training.
>
> However, we do have specific algorithmic design to prevent the noisy preference rewards from hurting the performance as the training goes on. As shown in Algorithm 2, the reward transformer is re-trained every 100 epochs based on the latest policy rollouts. As the robot performs better at later stage, the newly trained reward transformer can produce more useful reward signal with less noise.
>
> - **“The detailed prompt example should be provided.”**
>
> We have already provided detailed prompts for all the tasks in the Appendix A.1. We also referred to the readers to these prompts by the end of page 6 of the main texts.

---

> > ### Comment · Reviewer_68np · 2025-07-15
> >
> > Thank you very much for the response. I have increase the score.

---

> > > ### Author Response · Authors · 2025-07-16
> > > **Thank you!**
> > >
> > > We appreciate your support and efforts!

---

### Review · Reviewer_VNHv · 2025-06-23

**Summary Of Contributions:**

The paper presents Large Language Model-Assisted Preference Prediction (LAPP), which 1) uses LLMs to generate preference labels for state-action trajectories, 2) use the generated data to train a reward predictor, 3) the reward predictor is then used to train a policy that maximizes both the original rewards from the environment and the predicted preference rewards.

This works aims to address the following limitations of prior work that uses LLMs for generating preference labels are that 1) those works are limited to simple tasks with low-dimensional action spaces, 2) they assume a Markovian decision process, and 3) do not integrate LLM-generated feedback directly into the RL loop.  To tackle these, the work is evaluated on complex control tasks (1), use transformers for non-Markovian reward preference prediction (2), and direct integration of the LLM-generated feedback into the RL loop (3).

The proposed framework is evaluated on control tasks with quadruped locomotion / backflip and dexterous manipulation, with comparisons to baselines (PPO, Eureka) and ablations (on sampled trajectories, use of MLP vs transformer-based network for the reward predictor network).

**Audience:**

Yes

**Claims And Evidence:**

Yes

**Requested Changes:**

1. Improve presentation to clarify the key difference of this work wrt prior work such as RL-VLM-F.
   - Figure 1 can be replaced, or Figure 2 updated to show the key elements of this work that is different from prior work on using LLMs for generating preference data and modeling preferences
2. Add a clear discussion of what parts would a person need to specify for this framework
3. Add more discussion and comparison of the overhead of using LLMs for generating preference data and cost of training the ensemble of reward models
4. Ablation of the number of ensembled models (including M=1)
5. Add discussion of hyperparameter selection (e.g. error rate for Eq 4 set to $\epsilon = 0.15$, how is that value determined? Selection of M for number of epochs for retraining the preference model in section 4.3, etc)
6. Add videos to allow for better comparisons of the learned behaviors
7. Other writing improvements
   - Either tone down and adjust claims
   - Add comparison of tasks handled in prior work using LLMs (e.g. why are those tasks less complex, what is the action space) vs this work.
   - Avoid using of the same letter for different things (e.g. M is used both for the number of models that are ensembled in section 4.2, and the number of epochs in section 4.3)
   - Shorten main paper (the text can be made more concise with some details, e.g. Algorithms and some experiments can potentially be moved to the appendix)
   - "Consequently, We have:" => "We also have:"

**Strengths And Weaknesses:**

**Strengths**
1. The proposed method to use LLMs to augment RL training with generated preferences seems to be reasonable
2. Experiments show the proposed method is effective compared with the PPO baseline

**Weaknesses**
1. The key differences of the proposed *framework* compared to RL-VLM-F was not that clear
  - The reviewer believes that the key difference at the framework level is that this work also incorporate manually designed environment rewards, while RL-VLM-F does not.  Is that the case?
2. Training efficiency is measured mainly by the number of epochs
  - There is limited discussion of the overhead of the using the LLM to generate the preferences and training of the reward model.  The cost (e.g. time, compute, memory) should be included in comparisons and discussions.
3. Despite the initial discussion of the challenges and limitations of reward function design, it seems that the proposed framework actually introduces an additional component that needs to be designed as it requires the following:
  1) designing the prompt for input to the LLM
  2) writing out the code for the explicit environment reward
  It is not clear whether 2) is different from manual reward function design, but now there is an step of designing the prompt.
4. Limited comparisons and ablations
  - The reviewer would have liked to see a comparison with RL-VLM-F / other recent methods that uses LLMs for preference modeling
  - The work uses an ensemble of M preference predictor networks, there should be an ablation on the this
5. It is difficult to determine the quality of the results - videos should be included
6. Some claims are not fully substantiated / accurate
  - "Unlike prior approaches, LAPP integrates the LLM-generated feedback directly into the RL loop"
     - The reviewer believes that RL-VLM-F also integrates the LLM-generated feedback directly
  - the framework "enables robots to learn efficient, customizable, and expressive behaviors from human language specifications"
     - There is limited evidence of this - the efficiency, customizability, and expressiveness of behaviors are not evaluated

---

> ### Author Response · Authors · 2025-06-29
> **Response to Reviewer VNHv (part 1)**
>
> We thank the reviewer for your detailed review. We would like to address all your comments with point-by-point responses.
>
> **The key differences of the proposed framework compared to RL-VLM-F was not that clear. The reviewer believes that the key difference at the framework level is that this work also incorporates manually designed environment rewards, while RL-VLM-F does not. Is that the case?**
>
> We would like to clarify that the key difference is not incorporating manually designed environment rewards. We list several key differences, advantages, and contributions of LAPP compared with RL-VLM-F as follows:
>
> 1) **Preference labels generated from raw state-action trajectories without visuals:** LAPP leverages the strong reasoning capabilities of LLM for solving very challenging tasks such as backflip and dexterous manipulation. In contrast, the VLM adopted in the RL-VLM-F framework does not have such strong reasoning capability yet. In addition, the usage cost of LLM is much smaller than VLM when we input a trajectory of states sequence into the LLM instead of inputting a trajectory of images sequence into the VLM.
>
> 2) **The first paper that considers non-Markovian reward for RL-AI-F in robotics:** LAPP considers non-Markovian rewards for very challenging tasks such as backflip and gait cadence control. The LLM takes in two trajectories of state sequence for comparison. In contrast, RL-VLM-F only considers Markovian reward and input one pair of states (images) into the VLM. They did not rank two trajectories of a task but only rank two states (as images) of a task. LAPP is the first work that models the non- Markovian decision process with a transformer reward model, ranks two trajectories of a task with LLMs, and achieves satisfying performance on challenging tasks. Our ablation study shows the importance of this algorithmic design.
>
> 3) **Transformer-based Preference Predictor:** Our predictor is a transformer network while RL-VLM-F adopts ResNet or CNN. This design decision is crucial for handling the non-Markovian rewards mentioned above.
>
> 4) **Latest trajectories collection for preference adjustment throughout the training process:** Previous works train the preference predictor with the data collected online from the entire training process (RL-AL-F) or offline (offline RL-AI-F). Ours is the first work that proposes maintaining a data buffer with the latest rollouts (latest 500 epochs) for iterative online training, which is shown to outperform collecting data from the full training process in our ablation study. The LLM can decide the priority of preference according to the current training stage.
>
> 5) **First application of RL-AI-F to robot locomotion and dexterous manipulation:** Ours is the first work in RL-AI-F that solves robot locomotion across various terrains, dexterous manipulation, and quadruped backflip. While RL-AI-F is limited to manipulation tasks where robots have only 7 DoFs, our tasks involve robots with as much as 52 DoFs. The robot platforms are different, and the tasks are harder.
>
> In summary, this LAPP paper solves completely different and much harder tasks with many several novel technical designs compared to RL-AI-F.
>
> Regarding the incorporation of the environment rewards, LAPP incorporates them because they are well-designed reward functions from previous work and can ensure the basic performance of the RL training. We compare LAPP with the PPO baseline that only has the environment rewards but not the LLM preference feedback, and we find that LAPP has pronounced advantage to PPO in many aspects, demonstrating the strong effect of the preference reward in LAPP.
>
> We believe that the value of RL-VLM-F is to verify that the pure preference feedback from a foundation model is capable of solving some basic robotic tasks nowadays, while the value of LAPP is to push the limit of the RL-AI-F idea to very challenging tasks such as backflip and dexterous manipulation tasks by incorporating the static environment rewards and the strong preference reasoning capability of LLM. Both RL-VLM-F and LAPP should be the representative studies in RL-AI-F for robotics, and they can complement each other perfectly.
>
> We included some of these discussions in our paper, but we will clarify more with the above information with our final version.

---

> ### Author Response · Authors · 2025-06-29
> **Response to Reviewer VNHv (part 2)**
>
> **Improve presentation to clarify the key differences of this work with prior work such as RL-VLM-F. Figure 1 can be replaced, or Figure 2 updated to show the key elements of this work that are different from prior work on using LLMs for generating preference data and modeling preferences.**
>
> In Figure 1, we have presented that LAPP can train versatile behaviors for quadruped locomotion and dexterous manipulation in both simulation and real worlds. It can also improve the training efficiency, and the converged performance compared to baselines with only environment rewards. Moreover, LAPP can solve some very challenging tasks such as the quadruped backflip. These are all very challenging experiments that cannot be solved by RL-VLM-F, which clearly present the key differences of the outcomes of LAPP and RL-VLM-F, and, as a teaser figure, can effectively demonstrate the performance gains of LAPP.
>
> In Figure 2, we have presented the key elements of LAPP that are different from RL-VLM-F. We show that LAPP has a transformer reward predictor, it adopts LLM in the loop instead of a VLM, and the LLM takes in history sequences of states instead of images of the current step. These key elements have been clearly presented in Figure 2, and for other details of this method, the reader can refer to the text description and the algorithm pseudocodes. We believe the method figure should present the key elements of the algorithm, while omitting some other design details for a clearer presentation of the main pipeline of the algorithm.
>
> Following our response to the previous comment, we will add more clarifications in the Intro and Related works to clarify these differences.
>
> **There is limited discussion of the overhead of the using the LLM to generate the preferences and training of the reward model. The cost (e.g. time, compute, memory) should be included in comparisons and discussions.**
>
> We acknowledge that training and updating the reward predictor transformer will lead to some extra cost. We will include the cost of time, computation, and memory in the final version of this paper under Appendix and refer to it in our main paper.
>
> We would also like to highlight that the large performance gains and the enabling behaviors brought by LAPP, such as achieving tasks that could not be achieved by prior work such as backflip or gait pattern and gait cadence control, will make the additional cost less of a concern. With the rapid progress in cost-efficient LLMs, we believe that such costs will be greatly reduced over the years. Following this trend, the benefits by methods like LAPP will be more pronounced.
>
> **Despite the initial discussion of the challenges and limitations of reward function design, it seems that the proposed framework actually introduces an additional component that needs to be designed as it requires the following: 1) designing the prompt for input to the LLM, and 2) writing out the code for the explicit environment reward, but now there is an step of designing the prompt.**
>
> We acknowledge that LAPP requires prompt engineering for input to the LLM, which is commonly required for all the methods that have foundation models in the loop, such as RL-VLM-F and Eureka.
>
> We would like to clarify that one of the key contributions of this paper is to propose a template for the prompt engineering for the RL-AI-F framework with an LLM. As shown in our Figure 3, the prompt template with three sections of content has been proved to be effective in various robotic tasks in our paper. LAPP does not require re-designing the prompt for each specific task but provides a prompt template for filling in the states of different tasks and has proven the success of this prompt structure.
>
> The environment rewards are designed by other previous studies for different tasks, and LAPP can directly incorporate them into its framework and augment the performance in many aspects with a universal prompt template for preference feedback.

---

> ### Author Response · Authors · 2025-06-29
> **Response to Reviewer VNHv (part 3)**
>
> **Limited comparisons and ablations. Ablation of the number of ensembled models (including M=1).**
>
> We would like to clarify that this paper has abundant comparisons (with Eureka and PPO) across 9 challenging tasks and ablations to study the two key components, latest rollout trajectories update and transformer predictor network structures.
>
> We did not compare it with RL-VLM-F because this method only works for tasks with Markovian rewards and robots with low DoFs, while many tasks in LAPP have non-Markovian rewards and robots with much higher DoFs. In addition, the current reasoning capability of VLMs is not strong enough to understand a sequence of images accurately, and the usage fee of prompting with thousands of image sequences will be way too expensive to run. In addition, LAPP can produce flexible and dynamic reward signals based on the training progression with ongoing reward model training and therefore leads to various advantages to a static reward function as in RL-VLM-F, no matter if this reward function is tuned by humans as shown in the PPO baseline or tuned by LLM as shown in the Eureka baseline. We believe that our current comparisons can clearly support this claim.
>
> This work uses an ensemble of M preference predictor networks following the frequently cited prior work [1] of RLHF. Previous study [1] in RLHF has thoroughly studied the effects of using an ensemble of reward predictors and has reported the ablation results with only one predictor. Therefore, we directly used the previous verified conclusions in our framework and did not conduct additional ablations for this aspect.
>
> [1] Deep Reinforcement Learning from Human Preferences
>
> **It is difficult to determine the quality of the results - videos should be included. Add videos to allow for better comparisons of the learned behaviors.**
>
> We will include the video of results in the supplementary material.
>
> **Some claims are not fully substantiated / accurate**
>
> **"Unlike prior approaches, LAPP integrates the LLM-generated feedback directly into the RL loop". The reviewer believes that RL-VLM-F also integrates the LLM-generated feedback directly.**
>
> We would like to clarify that RL-VLM-F integrates VLM-generated feedback from image inputs. In contrast, LAPP adopts LLM, and pure language inputs and solves more challenging non-Markovian tasks. Therefore, our claim holds.
>
> **The framework "enables robots to learn efficient, customizable, and expressive behaviors from human language specifications" There is limited evidence of this - the efficiency, customizability, and expressiveness of behaviors are not evaluated.**
>
> The efficiency is shown in Figure 5, where LAPP leads to fewer epochs of optimization to converge to a high performance in many difficult tasks. The customizability and expressiveness are shown in Figure 7 and 8, where LAPP enables customizing the gait pattern and gait cadence of the quadruped robot with only natural language behavior instructions, but none of the baselines can achieve these.
>
> **Add a clear discussion of what parts a person would need to specify for this framework**
>
> A person needs to specify three things:
>
> 1) In the first section of the prompt, we need to describe the task and the behavior we want the robot to have.
> 2) In the second section of the prompt, we need to specify what states of the robot trajectories we want to input into the LLM and describe the meaning of the values of each state.
> 3) In the second section of the prompt, we also need to specify the criteria to judge which trajectory is better in a pair.
>
> These three parts differ from task to task, and a person would need to specify them accordingly. For all other parts of the prompt, they are shared for all the tasks and people can directly use the provided template in the Appendix.
>
> **Add more discussion and comparison of the overhead of using LLMs for generating preference data and cost of training the ensemble of reward models.**
>
> In the second paragraph of the Experiment section, we have discussed the overhead of using LLMS for generating preference data in detail. We not only report the usage fee of using different versions of LLMs but also compare it with the LLM usage fee of Eureka.
>
> Training the ensemble of reward models takes much less time and computational resources compared with training the robot policy networks, since this is a supervised learning process with a very small dataset. With the advancement of computational hardware, we believe the cost of training the reward models will be further diminished.

---

> ### Author Response · Authors · 2025-06-29
> **Response to Reviewer VNHv (part 4)**
>
> **Add discussion of hyperparameter selection (e.g. error rate for Eq 4 set to 0.15, how is that value determined? Selection of M for number of epochs for retraining the preference model in section 4.3, etc)**
>
> The prior work of RLHF [1] selects the error rate for Eq 4 as 0.10, which assumes that there is a 10% chance that humans respond uniformly at random. We select this hyperparameter to be 0.15 since the current intelligence of LLM is not as good as humans’ yet and we assume a slightly larger chance for LLMs to respond uniformly at random. Our experiment results show that with the same error rate, LAPP can solve many challenging tasks such as quadruped backflip and dexterous manipulation, indicating that the good performance of LAPP does not rely on an optimal error rate.
>
> In all our experiments, we select M=9 preference predictors. This number balances the cost of the training and the accuracy of the preference prediction.
>
> [1] Deep Reinforcement Learning from Human Preferences
>
> **Other writing improvements**
>
> **Either tone down or adjust claims.**
>
> As we responded earlier, the claims are accurate with experimental results to justify them.
>
> **Add a comparison of tasks handled in prior work using LLMs (e.g. why are those tasks less complex, what is the action space) vs this work.**
>
> We have stated in the introduction that the tasks handled in prior work using VLMs have low-dimensional action spaces. Specifically, the manipulation tasks in RL-VLM-F have 4 degrees of freedom. In comparison, the dexterous manipulation tasks in LAPP have 52 degrees of freedom, and the locomotion tasks have 12 degrees of freedom. We will add this comparison with these clear numbers in the final version of the paper.
>
> **Avoid using the same letter for different things (e.g. M is used both for the number of models that are ensembled in section 4.2, and the number of epochs in section 4.3).**
>
> We will correct this typo in our final version of the paper.
>
> **Shorten main paper (the text can be made more concise with some details, e.g. Algorithms and some experiments can potentially be moved to the appendix).**
>
> We will shorten the main paper based on your suggestion.
>
> **"Consequently, we have:" => "We also have:"**
>
> We will revise our expression based on your suggestion.

---

### Review · Reviewer_kmca · 2025-07-14

**Summary Of Contributions:**

1. LAPP proposes a novel framework of using LLMs to generate feedback for preference signals in robot learning. This process mitigates issues introduced by human annotation.
2. LAPP demonstrates through experiments that it outperforms state-of-the-art baselines in training speed, final performance, adaptation efficiency, and controllability of high-level behaviors across a range of complex robotic tasks.

**Audience:**

Yes

**Claims And Evidence:**

Yes

**Requested Changes:**

Please address the Weaknesses & Questions mentioned above.

**Strengths And Weaknesses:**

Strengths:
1. Sufficient Experimentation: The paper evaluates LAPP on a diverse and challenging set of tasks, including quadruped locomotion over various terrains, high-DoF dexterous manipulation, and the challenging backflip task.
2. Significant Results: The experimental results strongly support LAPP's effectiveness. It not only improves training efficiency and final performance on multiple tasks but also successfully achieves fine-grained control over robot behaviors via high-level language instructions (e.g., gait patterns and cadence). Furthermore, it solves the backflip task, which traditional methods struggle with.

Weaknesses & Questions:
1. Inconsistent Writing: The Authors use LAPP in most places but use Lapp occasionally and are advised to further polish the writing.
2. Incomplete Related Work: In the first section of related work (i.e., Foundation Models for Robotics), the authors fail to give a brief introduction. I advise authors to reorganize the related work section as it is not clear enough and it shares overlap with the introduction section when it comes to the details of LAPP method.
3. Why do authors use ensemble learning strategy when training the preference predictor? Authors mentioned that it could increase robustness but I wonder if training multiple independent predictors help. If there is any evidence back up?
4. Can authors analyze more about the two rewards used? For example, their distinct effect on the results, their correlation and intuitive interpretation?
5. Is the title very appropriate? It does not convey any robot learning related information.

---

> ### Author Response · Authors · 2025-07-16
> **Response to Reviewer kmca**
>
> We thank the reviewer for taking the time and effort to review our manuscript. We appreciate your positive feedback on our sufficient experimentation and significant results. We would like to address all your concerns and questions below with point responses:
>
> **Inconsistent Writing: The Authors use LAPP in most places but use Lapp occasionally and are advised to further polish the writing.**
>
> Thank you for pointing out this typo. We found two usages of “Lapp” on page 12. We will correct them to use “LAPP”.
>
> **Incomplete Related Work: In the first section of related work (i.e., Foundation Models for Robotics), the authors fail to give a brief introduction. I advise authors to reorganize the related work section as it is not clear enough and it shares overlap with the introduction section when it comes to the details of LAPP method.**
>
> We will polish the first section of the related work to give a clearer introduction of the foundation models for robotics. Our discussions about LAPP, especially in the last section of the Related Work, aims to provide essential comparisons with the closest literature. We will highlight these differences in a more concise way.
>
> **Why do authors use ensemble learning strategy when training the preference predictor? Authors mentioned that it could increase robustness, but I wonder if training multiple independent predictors helps. If there is any evidence back up?**
>
> Our design choice of using ensemble learning follows the frequently cited prior work [1] on RLHF. Previous study [1] has thoroughly studied the effects of using an ensemble of reward predictors and has reported its advantage in robustness and the converged performance to the ablation results with only one predictor. Therefore, we directly used the previous conclusions in our framework, and we refer the readers to the prior work [1] for detailed evidence.
>
> [1] Deep Reinforcement Learning from Human Preferences
>
> **Can authors analyze more about the two rewards used? For example, their distinct effect on the results, their correlation and intuitive interpretation?**
>
> The preference reward is the predicted preference of the LLM, which provides a more flexible reward signal that evolves dynamically throughout the training process. The environment reward is a well-designed static reward function from previous work and can ensure the basic performance of the RL training. The combination of both rewards leads to the significant performance advantage of LAPP. As a comparison, the PPO baseline only has the environment reward but not the preference reward. Our experiments have shown LAPP has stronger performance than PPO in various tasks and can solve the challenging backflip task which PPO struggles with. To help build more intuitions, we plan to include some trajectories paired with LAPP rewards in Appendix.
>
> **Is the title very appropriate? It does not convey any robot learning related information.**
>
> Thank you for pointing this out. While considering the length and the information of the title, we chose “reinforcement learning” over “robot learning” since our work emphasizes more on the algorithm aspects of preference-based reinforcement learning (PbRL). Similar to many past papers in PbRL, though experiments in robot control are common choices due to its challenging nature in dimensionality and complexity, the titles do not necessarily contain robot learning. We choose RL to reach a broader audience and invite future research from other RL application domains.

---

### Review · Reviewer_Rgnk · 2025-07-16

**Summary Of Contributions:**

This paper introduces Large Language Model-Assisted Preference Prediction (LAPP), a novel robot learning framework that leverages feedback from large language models (LLMs) to guide the training of both the preference predictor and the policy network. At regular intervals, new trajectories are collected and evaluated by the LLM, which are then used to retrain the preference predictor. This forms an online learning strategy that progressively improves the policy over time. Experimental results demonstrate the effectiveness of the proposed method across a variety of tasks, achieving rapid convergence and superior performance compared to baseline approaches.

**Audience:**

Yes

**Claims And Evidence:**

Yes

**Requested Changes:**

Please see the weaknesses.

**Strengths And Weaknesses:**

Strengths
1. The proposed idea is both interesting and novel. The online learning procedure effectively leverages the capabilities of LLMs, transferring their rich knowledge into robot learning.
2. The writing is clear and comprehensive. The paper first introduces the fundamentals of reinforcement learning, and then clearly explains the motivation and details of each module.

Weaknesses
1. Figure 2 could be improved for better clarity and visual appeal. Additionally, the diagram of the Transformer Reward Predictor currently resembles that of an MLP network, which may cause confusion.
2. Given the numerous hyperparameters involved in the proposed method, the authors should explain how the optimal values are selected and provide a sensitivity analysis.
3. The statements regarding the advantages are not sufficiently precise. For example, the abstract mentions "efficient, customizable behavior acquisition," but the experimental results primarily demonstrate faster convergence in terms of epochs. Compared to vanilla PPO, the proposed method leverages additional components such as the LLM and preference predictor, potentially leading to differences in computational time and resources. Therefore, the claim of efficiency may not be fully justified. Furthermore, the experiments do not clearly demonstrate the "customizable" aspect.
4. The experimental evaluation should include more LLMs, such as Llama and DeepSeek, to better illustrate the effects of different language models.

---

> ### Author Response · Authors · 2025-07-18
> **Response to Reviewer Rgnk (Part 1)**
>
> We thank the reviewer for taking the time and effort to review our manuscript. We appreciate your positive feedback on our novel idea and writing style. We would like to address all your concerns and questions below with point responses:
>
> **Figure 2 could be improved for better clarity and visual appeal. Additionally, the diagram of the Transformer Reward Predictor currently resembles that of an MLP network, which may cause confusion.**
>
> In Figure 2, present the key elements of the LAPP algorithm, while omitting some other design details for a clearer presentation of the main pipeline of the algorithm. We acknowledge that the diagram of the Transformer Reward Predictor currently resembles that of an MLP network, and we will modify this diagram to represent a transformer more precisely.
>
> **Given the numerous hyperparameters involved in the proposed method, the authors should explain how the optimal values are selected and provide a sensitivity analysis.**
>
> We provide the explanations of the hyperparameters as follows and will clarify them in the paper.
>
> For the error rate in Eq 4, the frequently cited prior work of RLHF [1] selects this error rate as 0.10, which assumes that there is a 10% chance that humans respond uniformly at random. We select this hyperparameter to be 0.15 since the current intelligence of LLM is not as good as humans’ yet and we assume a slightly larger chance for LLMs to respond uniformly at random. In other words, this number reflects how much we trust the LLM preference feedback (or human preference feedback in our referenced work [1]). Our experiment results show that with this same error rate, LAPP can solve all our tasks, indicating that the good performance of LAPP is not sensitive to this parameter. Running a comprehensive experiments would be too expensive due to the LLM API cost.
>
> For the number of the ensembles of preference predictors, we choose 9 to balance the cost of the training and the accuracy of the preference prediction. As in past literature, this parameter is often chosen to be large enough to maintain a robust prediction but remain reasonable computational costs.
>
> For the number of epochs for retraining the preference model in section 4.3, we choose 100 to ensure the preference model is updated frequently enough to reflect the latest preference of the LLM, while it is not retrained too frequently to slow down the entire training process too much.
>
> For other hyperparameters involved in the RL algorithm, we directly adopt the well-tuned hyperparameters from previous open-sourced projects of quadruped locomotion [2] and dexterous manipulation [3].
>
> Importantly, we adopt uniformly the same set of hyperparameters across all the tasks and LAPP constantly achieves higher performance than other baselines. These results show that LAPP is not sensitive to hyperparameters and does not require a unique set of optimal hyperparameters for achieving good performance on each task.
>
> [1] Deep Reinforcement Learning from Human Preferences
>
> [2] Learning to Walk in Minutes Using Massively Parallel Deep Reinforcement Learning
>
> [3] Eureka: Human-Level Reward Design via Coding Large Language Models
>
> **The statements regarding the advantages are not sufficiently precise. For example, the abstract mentions "efficient, customizable behavior acquisition," but the experimental results primarily demonstrate faster convergence in terms of epochs. Compared to vanilla PPO, the proposed method leverages additional components such as the LLM and preference predictor, potentially leading to differences in computational time and resources. Therefore, the claim of efficiency may not be fully justified. Furthermore, the experiments do not clearly demonstrate the "customizable" aspect.**
>
> We would like to highlight that the customizability and expressiveness are shown in Figure 7 and 8, where LAPP enables customizing the gait pattern and gait cadence of the quadruped robot with only natural language behavior instructions, but none of the baselines can achieve these. In terms of efficiency, LAPP leads to fewer epochs of optimization to converge to a higher performance in many difficult tasks in Figure 5. We will clarify that this efficiency refers to sample efficiency since the LLM and preference predictor do not require any additional environment interactions. We agree that LLM brought extra cost. However, this is a common overhead for all works involving LLM in the loop. In our case, LAPP not only reduces the environment interactions, but also obtains higher performance and more customizable and expressive behaviors. Moreover, challenging tasks such as backflip was not possible with other baseline methods without LLMs or with LLMs but a different approach. We leave the exploration to further bring the cost of LLM calls down in future work.

---

> ### Author Response · Authors · 2025-07-18
> **Response to Reviewer Rgnk (Part 2)**
>
> **The experimental evaluation should include more LLMs, such as Llama and DeepSeek, to better illustrate the effects of different language models.**
>
> We chose GPT-4o-mini as it is among the state-of-the-art LLMs with reasonable cost. Our focus of this paper is to study how to leverage LLMs to provide effective feedback to train RL agents in challenging tasks with better sample efficiency and higher performance. The comparison between different LLMs is out of scope of this paper. We agree that this can be an interesting study for future work. We will add this to the future work.

---

### Review · Reviewer_7jqc · 2025-07-18

**Summary Of Contributions:**

This paper introduces LAPP (Large Language Model-Assisted Preference Prediction), a framework that uses the feedback of LLMs to train the preference models to generate the preference feedback over the trajectories of reinforcement learning. It aims to reduce the dependence of human annotations during the reinforcement learning process, so that the whole learning process can be more efficient and scalable. Both the policy and the preference model are updated iteratively during the training process. Empirical studies are carried out to demonstrate LAPP is superior compared to the baselines in terms of training efficiency, convergence speed, behavior control.

**Audience:**

Yes

**Broader Impact Concerns:**

None.

**Claims And Evidence:**

Yes

**Requested Changes:**

None.

**Strengths And Weaknesses:**

**Strengths**
- The proposed LAPP framework effectively integrates LLM-generated feedback into the RL loop, bypassing the need for costly human labeling or complex reward engineering. It makes the whole training process more scalable and effective.
- The empirical experiments demonstrates faster convergence and higher final performance across diverse robotic tasks, including manipulation and locomotion.

**Weaknesses**
- The choice of the LLM seems to be essential. The paper only adopts GPT-4o and GPT-4o-mini as the backbone model. It is highly recommended to incorporate more popular open-sourced models, like the Llama series, Qwen series or Deepseek series, into the experiments to show the effectiveness of LAPP across models.
- As indicated by the authors in the limitation section, the current framework still requires manual selection of state variables. Therefore, the human efforts are not completely avoided in the training loop.
- The framework requires updating the preference model every $M$ epoch, it is appreciated if this additional computational overhead is included in the final result.
- The ablation study can be further extended. For instance,
	- How the error probability $15\%$ of LLM is selected? It is a model-specific parameter and requires further validations.
	- I am curious how the $\beta=50$ is determined in the reward design. It seems the preference feedback is dominating the reward, then why not remove the environmental reward $r_E$? Is the choice of $\beta$ task-dependent?

---

> ### Author Response · Authors · 2025-07-19
> **Response to Reviewer 7jqc**
>
> We thank the reviewer for taking the time and effort to review our manuscript. We appreciate your positive feedback on our successful integration of LLM-generated feedback into the RL loop and the performance gain of LAPP across diverse robotics tasks. We would like to address all your concerns and questions below with point responses:
>
> **The choice of the LLM seems to be essential. The paper only adopts GPT-4o and GPT-4o-mini as the backbone model. It is highly recommended to incorporate more popular open-sourced models, like the Llama series, Qwen series or Deepseek series, into the experiments to show the effectiveness of LAPP across models.**
>
> We chose GPT-4o-mini for LAPP as it is among the state-of-the-art LLMs with reasonable cost. Our focus of this paper is to study how to leverage LLMs to provide effective feedback to train RL agents in challenging tasks with better sample efficiency and higher performance. The comparison between different LLMs is out of scope of this paper as our focus is on the methodology over benchmark among LLMs. We agree that this is an interesting direction for future work. We will add this to the section of future work.
>
> **As indicated by the authors in the limitation section, the current framework still requires manual selection of state variables. Therefore, the human efforts are not completely avoided in the training loop.**
>
> In the section of limitation, we have acknowledged that the current framework still requires manual selection of state variables. We also proposed a potential solution: initializing training with different variable subsets and selecting the best based on warm-up performance. As LLM reasoning improves, irrelevant variables could also be filtered automatically. Our current main reason for dropping some of them is limited by the contextual window, which is a popular topic among LLM advancements. We leave the exploration to the solution to the future work.
>
> **The framework requires updating the preference model every M epoch, it is appreciated if this additional computational overhead is included in the final result.**
>
> We acknowledge that training and updating the reward predictor transformer will lead to some extra cost. We will include this detail in our Appendix.
>
> **The ablation study can be further extended. For instance, how is the error probability 15 of LLM selected? It is a model-specific parameter and requires further validations.**
>
> The frequently cited prior work of RLHF [1] selects the error rate for Eq 4 as 0.10, which assumes that there is a 10% chance that humans respond uniformly at random. We select this hyperparameter to be 0.15 since the current intelligence of LLM is not as good as humans’ yet and we assume a slightly larger chance for LLMs to respond uniformly at random. Our experiment results show that with the same error rate, LAPP can solve many challenging tasks such as quadruped backflip and dexterous manipulation, indicating that LAPP is not sensitive to this parameter.
>
> [1] Deep Reinforcement Learning from Human Preferences
>
> **I am curious how the beta=50 is determined in the reward design. It seems the preference feedback is dominating the reward, then why not remove the environmental reward rE? Is the choice of beta task-dependent?**
>
> We used beta=1.0 in all our other 8 tasks. The only exception is the backflip task which is particularly challenging and has not been shown to be possible with end-to-end RL or LLM-guided learning. We therefore manually tuned its environment reward function for the PPO to obtain a competitive baseline. In this case, we increased β to 50 to balance the scales of the environment reward and preference feedback. Our strategy is to set β such that both rewards contribute comparably. This does not involve further tuning and remains consistent. We emphasize that both types of rewards are essential for stable and effective learning.

---

### Comment · Editors_In_Chief · 2025-10-08

On October 8, after the paper was published, a new camera-ready version was uploaded by the Editors-in-Chief, since they noticed the previous camera-ready version did not follow TMLR's style file, and the authors promptly furnished a revised version. Otherwise, content remains the same.

---

### Decision · Action_Editor_mH4b · 2025-08-07

**Recommendation:** Accept with minor revision

**Additional Comments:**

While the paper presents a novel and interesting framework that leverages LLM feedback to guide reinforcement learning, there remain several points mentioned by Reviewers VNHv and Rgnk that require clarification and revision before the manuscript can be considered ready. Specifically, the claims regarding the distinction between LLM- vs. VLM-generated feedback should be stated more precisely, as the current framing leaves room for ambiguity. The authors are also expected to add supporting evidence or discussion for some of the design choices, to ensure that the claims are sufficiently substantiated. In addition, the presentation would benefit from improved readability of the overview figure, and an ablation analysis across different LLM backbones would strengthen the empirical contribution.

**Audience:**

Yes

**Audience Explanation:**

The paper is highly relevant to the TMLR audience, particularly those interested in reinforcement learning, robot learning, and the use of large language models for control tasks.

**Claims And Evidence:**

Yes

**Claims Explanation:**

The paper introduces LAPP, a framework that integrates large language models into preference-driven reinforcement learning. The central claims that LAPP improves sample efficiency, enables customizable and expressive behaviors, and outperforms existing baselines on complex tasks are generally well supported by experiments and analysis. The authors present evaluations across multiple robotic control tasks including quadruped locomotion and dexterous manipulation, and achieve improvements over PPO and Eureka.

---

> ### Author Response · Authors · 2025-09-07
> **Revise and submit the camera-ready version**
>
> We thank the action editor for the efforts and support. We have revised the paper according to the reviewers’ requests for the camera-ready version. Here is a summarization of the modifications:
>
> 1) Add comparison tasks handled in prior work using VLMs vs. this work with clear numbers of degrees of freedom.
>
> 2) Replace Figure 2 with a proper transformer diagram.
>
> 3) Include the cost of time in the final version of this paper in Appendix A.5 and refer to it in our main paper.
>
> 4) Include the videos via the link of the project website.
>
> 5) Include some trajectories paired with two LAPP rewards in Appendix A.4 to help build more intuitions of their distinct effect on the results, their correlation and intuitive interpretation.
>
> 6) Add the discussion for the comparison between different LLMs to the section of future work.
>
> 7) Polish the first section of the related work to give a clearer introduction of the foundation models for robotics. Highlight the differences of LAPP with the closest literature in the last section of the related work.
>
> 8) Modify the letters of the variables in Algorithm 1 to avoid using the same letters for different things.
>
> 9) Replace "Consequently, we have:" with "We also have:" in the Preliminaries section.
>
> 10) Find two usages of “Lapp” on page 12 and correct them to “LAPP”.